# The hitchhiker's guide to 4d $\mathcal{N} = 2$ superconformal field theories

Mohammad Akhond[1*], Guillermo Arias-Tamargo[2,3†], Alessandro Mininno[4,5‡],
Hao-Yu Sun[6∘], Zhengdi Sun[7§], Yifan Wang[8,9,10¶] and Fengjun Xu[11,12,13‖]

**1** Department of Physics, Swansea University, Singleton Park,
Swansea, SA2 8PP, United Kingdom
**2** Department of Physics, Universidad de Oviedo,
C/ Federico García Lorca 18, 33007 Oviedo, Spain
**3** Instituto Universitario de Ciencias y Tecnologías Espaciales de Asturias (ICTEA),
C/ de la Independencia 13, 33004 Oviedo, Spain
**4** Instituto de Física Teórica IFT-UAM/CSIC, C/ Nicolás Cabrera 13-15,
Campus de Cantoblanco, 28049 Madrid, Spain
**5** II. Institut für Theoretische Physik, Universität Hamburg,
Luruper Chaussee 149, 22607 Hamburg, Germany
**6** Theory Group, Department of Physics, University of Texas, Austin, TX 78712-1192, USA
**7** Department of Physics, University of California, San Diego, CA 92093-0319, USA
**8** Center of Mathematical Sciences and Applications, Harvard University,
Cambridge, MA 02138, USA
**9** Jefferson Physical Laboratory, Harvard University, Cambridge, MA 02138, USA
**10** Center for Cosmology and Particle Physics, New York University,
New York, NY 10003, USA
**11** Institut für Theoretische Physik, Ruprecht-Karls-Universität,
Philosophenweg 19, 69120, Heidelberg, Germany
**12** Yau Mathematical Sciences Center, Tsinghua University, Beijing, 100084, China
**13** Beijing Institute of Mathematical Sciences and Applications (BIMSA),
Huairou District, Beijing, 101408, China

⋆ akhondmohammad@gmail.com , † ariasguillermo@uniovi.es ,
‡ alessandro.mininno@desy.de , ∘ hkdavidsun@utexas.edu , § z5sun@ucsd.edu ,
¶ yifanw@g.harvard.edu , ‖ xufengjun321@gmail.com

## Abstract

Superconformal field theory with $\mathcal{N} = 2$ supersymmetry in four dimensional spacetime provides a prime playground to study strongly coupled phenomena in quantum field theory. Its rigid structure ensures valuable analytic control over non-perturbative effects, yet the theory is still flexible enough to incorporate a large landscape of quantum systems. Here we aim to offer a guidebook to fundamental features of the 4d $\mathcal{N} = 2$ superconformal field theories and basic tools to construct them in string/M-/F-theory. The content is based on a series of lectures at the Quantum Field Theories and Geometry School in July 2020.

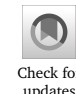

# 1 Introduction

Since its inception about a century ago, quantum field theory (QFT) in four spacetime dimensions has remained an important and enduring theme in theoretical physics. To date, it has produced incredibly precise predictions that present an astoundingly accurate description of our physical world, for example for the particle collisions at the Large Hadron Collider (LHC). Yet, our understanding of QFT at the fundamental level is still rather limited. Conventional approaches to QFT rely on a formulation involving elementary quantum fields together with a Lagrangian that captures the interactions, in which case, physical observables such as correlation functions or scattering amplitudes can be extracted by perturbative Feynman diagram computations. However, it is soon realized that such perturbative methods often either fail or simply do not exist for a general QFT. This happens when the system is strongly coupled and the relevant physical observables do not have (obvious) small expansion parameters. In such a scenario, non-perturbative effects are important and there is no useful Lagrangian procedure. These strong coupling phenomena are particularly common for QFTs in four spacetime dimensions in the infrared limit, thanks to the asymptotic freedom of gauge theories, including the Quantum Chromodynamics (QCD) that describes the strong force that binds the quarks. The obvious challenge is to develop non-perturbative methods that aid and transcend the Lagrangian approach. This has been the focus of many recent research efforts, resulting in especially rich and varied techniques in four dimensions.

A major handle to tame the strongly coupled dynamics in QFTs comes from supersymmetry (SUSY). As with any symmetries, SUSY elucidates the phase diagram of the theory and constrains the observables that preserve (a fraction of) SUSY, doing it so efficiently that it becomes much more tractable to understand the physics in the strong coupling regime. A prime example is the study of $\mathcal{N} = 1$ super-QCD (SQCD) like theories in [1–4]. Here $\mathcal{N}$ counts the number of supersymmetries in the theory, and one naturally expects the constraints from SUSY to become more stringent for larger $\mathcal{N}$. For an interacting QFT in 4d, $\mathcal{N}$ can take values between 1 and 4 [5]. For the extreme case at $\mathcal{N} = 4$, the theory is believed to be uniquely specified by a gauge group (up to discrete topological data), and corresponds to the $\mathcal{N} = 4$ super-Yang-Mills theory.[1] The $\mathcal{N} = 4$ SYM was heavily studied during the late 70's and early 80's, in particular to show that it is ultraviolet finite and conformally invariant [10–13].[2] In fact the $\mathcal{N} = 4$ SYM was the first conformal field theory (CFT) discovered in four dimensions and generalizations to Yang-Mills theories with less supersymmetry were explored shortly after (see similar analysis of finiteness in e.g. [14–19]). The $\mathcal{N} = 2$ case represents a sweet spot between the powerful SUSY constraints and the interesting strongly coupled dynamics in four dimensions, as is evident since the works of [20, 21]. On the one hand, the extra SUSY beyond $\mathcal{N} = 1$ provides the necessary quantitative control over non-perturbative effects, such as instantons and monopoles in gauge theories, for extracting physical observables at strong coupling. On the other hand, $\mathcal{N} = 2$ QFTs share many features, such as emergent conformal symmetry and electric-magnetic duality [22, 23], with more general strongly coupled theories in four dimensions.

Although the 4d $\mathcal{N} = 2$ QFTs were initially constructed and studied based on the Lagrangian approach [15, 16, 20, 21, 24], the picture of what a generic 4d $\mathcal{N} = 2$ QFT is has evolved by leaps and bounds over the years, thanks to the field-theoretic constructions of strongly-coupled theories in [22, 23, 25] and then generalizations from myriad constructions

---

[1]In particular, the $\mathcal{N} = 4$ Superconformal field theories (SCFTs) all have a complex exactly marginal coupling [5, 6], which is identified with the complexified gauge coupling $\tau = \frac{4\pi i}{g_{\text{YM}}^2} + \frac{\theta}{2\pi}$ in the Supersymmetric Yang-Mills (SYM) theory. For a fixed gauge group, the SYM may also have an additional discrete theta angle [7]. We note that a rigorous proof that all $\mathcal{N} = 4$ SCFTs are $\mathcal{N} = 4$ SYMs remains open. Some progress has been made using the superconformal bootstrap [8, 9].

[2]We thank Peter West for explaining to us the early history of 4d SCFTs.

in string/M-/F-theory [26–60]. By consideration of singular geometries in the presence of branes and fluxes that preserve a 4d $\mathcal{N} = 2$ Poincaré supersymmetry, the interesting field theories arise from certain limits (e.g., of the string scale and coupling in string theory) where gravity decouples. While some of the resulting theories have familiar $\mathcal{N} = 2$ Lagrangians, general $\mathcal{N} = 2$ theories that are produced this way are *non-Lagrangian*. Nonetheless, these theories are fully specified by the string/M-/F-theory construction, including all non-perturbative effects, by virtue of string dualities [61]. Consequently, strong coupling phases of the field theory are often directly accessible from the geometry. In particular, $\mathcal{N} = 2$ SCFTs which describe the $\mathcal{N} = 2$ supersymmetric fixed points arise from limits of the geometric setups in string/M-/F-theory that have a *scaling symmetry*, which becomes a part of the full superconformal symmetry in the resulting field theory. Since the SCFTs and their deformations chart the landscape of general 4d $\mathcal{N} = 2$ theories, it behooves us to explore general constructions of these fixed points from the geometric setups in string/M-/F-theory.

Here we come to the main purpose for this set of lecture notes. That is, to provide a brief introduction to a selected collection of geometric tools in string/M-/F-theory that construct 4d $\mathcal{N} = 2$ SCFTs and determine physical observables therein. One approach relies on a generalized version of the Class $\mathcal{S}$ constructions introduced in [33,35], that uses M5-branes (or rather the worldvolume 6d $\mathcal{N} = (2,0)$ theories) compactified on a Riemann surface with punctures and twists, which we will review in Section 4. The other approach deals with type IIB string theory probing an isolated three-fold singularity first introduced in [31], as reviewed in Section 5. We also discuss briefly the F-theory constructions in Section 5.4.2 which generalize the type IIB setup by allowing for a non-trivial axion-dilaton background. Each construction leads to an infinite family of 4d $\mathcal{N} = 2$ SCFTs, and it is not uncommon that the same SCFT can arise from multiple string/M-/F-theory constructions, which often shed complementary light on the SCFT. This selection of topics is made to modestly complement a number of recent reviews on 4d $\mathcal{N} = 2$ CFTs [62–64]. We emphasize that there are other constructions of 4d $\mathcal{N} = 2$ SCFTs which will be outside the scope here, for example from toroidal compactifications of 6d $\mathcal{N} = (1,0)$ or 5d $\mathcal{N} = 1$ SCFTs, see [65–71] for a representative list of references.[3] We end by discussing a number of open questions in Section 6.

To proceed, we start by reviewing the basics of 4d $\mathcal{N} = 2$ supersymmetry and superconformal symmetry in Section 2, paying attention to important physical observables in an $\mathcal{N} = 2$ SCFT such as protected operator spectrum and anomalies. These observables are typically difficult to access directly at the fixed point, due to the lack of a perturbative Lagrangian description. A natural strategy is to study universal deformations of the fixed point theory and to extract SCFT data by extrapolating observables in the deformed theory. A general feature of the $\mathcal{N} = 2$ SCFTs is the presence of a vacuum moduli space that preserves $\mathcal{N} = 2$ SUSY where a useful connection to Lagrangian theories can be made. In particular, we consider the deformation of the SCFT that amounts to moving onto the so-called Coulomb branch (CB) of the moduli space, which is generally expected to exist for interacting $\mathcal{N} = 2$ SCFTs. The far infrared (IR) physics on the Coulomb branch is described by an $\mathcal{N} = 2$ Abelian gauge theory, whose interactions are governed by a *holomorphic* prepotential as a consequence of the $\mathcal{N} = 2$ SUSY which we review in Section 3. Nonetheless, the Coulomb branch effective field theory (EFT) is a very rich object, which is highly sensitive to the spectrum of Bogomol'nyi-Prasad-Sommerfield (BPS) particles supported on the Coulomb branch, that undergo non-trivial wall-crossing and monodromies as we traverse the moduli space [20,21,35,72]. It has interesting interplay with observables that are naturally defined in the SCFT at the origin on the Coulomb branch where the BPS particles become massless, and the interactions are strong, and also in

---

[3]In particular it will be very interesting to understand systematically the relations between the *direct* geometric engineering constructions of 4d $\mathcal{N} = 2$ SCFTs and those from compactifying 6d and 5d SCFTs. The latter also have their own geometric constructions in string/M-/F-theory.

the geometric background in string/M-/F-theory that engineers the theory where the Coulomb branch translates to geometric moduli of the setup, as summarized in Figure 1. This interwoven connection between the SCFT, its EFT and the corresponding string/M-/F-theory geometry enables one to study SCFT observables from both field theoretic and geometric techniques. The dialogues between these techniques not only enhance our physical understanding of the SCFTs but also lead to intriguing relations and identities in mathematics. For example, the Coulomb branch EFT encodes the spectrum of BPS operators at the fixed point which create the BPS states on the moduli space where conformal symmetry is spontaneously broken [72–74]. The EFT also determines various 't Hooft anomalies of the SCFT through the anomaly matching mechanism [75]. More generally, the EFT observables are closely related to supersymmetric partition functions of the SCFT, which can be computed exactly for Lagrangian theories via localization [76–78] (see also the review [79] and references therein). Furthermore, the Coulomb branch EFT provides a beautiful interplay between $\mathcal{N} = 2$ SCFTs and geometry. There is an emergent Riemann surface fibered over the Coulomb branch, giving rise to the Seiberg-Witten (SW) geometry [20,21]. This Riemann surface is known as the Seiberg-Witten curve, whose complex structure varies over the Coulomb branch and encodes the prepotential that determines the EFT. Initially thought of a trick to solve the EFT, the SW geometry (and its generalizations) turns out to appear naturally in the constructions of $\mathcal{N} = 2$ theories from string/M-/F-theory. It keeps track of certain deformation moduli of the geometric backgrounds (in the presence of branes) that preserve $\mathcal{N} = 2$ supersymmetry and survive the field theory limit. A large portion of Section 4 is dedicated to explaining how such a picture arises from the Class $\mathcal{S}$ constructions of 4d $\mathcal{N} = 2$ SCFTs using M5-branes (or rather the 6d (2,0) theories) and in Section 5, we will see a generalized SW geometry that emerges from type IIB string theory probing threefold singularities. In all these cases, the BPS solutions (e.g., instantons and particles) in the field theory correspond to BPS brane (and string) configurations, which can be counted by certain enumerative invariants of the higher dimensional geometry [80–83]. For example, for an $\mathcal{N} = 2$ theory engineered by type IIB string theory probing a singular Calabi-Yau (CY) three-fold, its prepotential is determined by the (refined) topological string partition function of the singular CY, which corresponds to the instanton partition function of the field theory (we refer to [84] for a review on this subject).

Despite the lack of a perturbative description, there have also been steady progress in understanding universal aspects of the $\mathcal{N} = 2$ SCFTs at an abstract level. This is thanks to an axiomatic definition of the SCFT by the spectrum of local operators and the operator-product-expansion (OPE) which obey constraints from associativity, unitarity and superconformal symmetry. A major development in recent years is the (revitalized) conformal bootstrap program (see [85] for a review), which explores these constraints to carve out the space of CFTs on a slice of the infinite dimensional theory space. For general CFTs, the most stringent bounds come from numeric approaches to the bootstrap equations. In the presence of $\mathcal{N} = 2$ superconformal symmetry, it turns out that the 4d SCFT contains a solvable yet rich sub-sector of operators that close under OPE, described by an emergent 2d chiral algebra [86]. There have been a lot of developments in the mini-bootstrap program that attempts to understand the space of chiral algebras that are relevant for $\mathcal{N} = 2$ SCFTs (see, e.g., [87–89] and also [90] for a recent review). Furthermore, knowledge of the chiral algebra in a given SCFT can be fed into the numerical bootstrap program to produce stronger bounds on more general OPE data in the theory. This latter strategy also applies to other SCFT data that can be obtained, for example, from the Coulomb branch EFT and the string/M-/F-theory geometry reviewed above, although not as much explored in $\mathcal{N} = 2$ SCFTs (see [91–94]). It is also natural to contemplate the geometric meaning of the chiral algebra in string/M-/F-theory constructions. It is our hope that a vigorous synergy of the conformal bootstrap, EFT approach, and geometric constructions will greatly enhance our understanding of the 4d $\mathcal{N} = 2$ SCFTs and beyond.

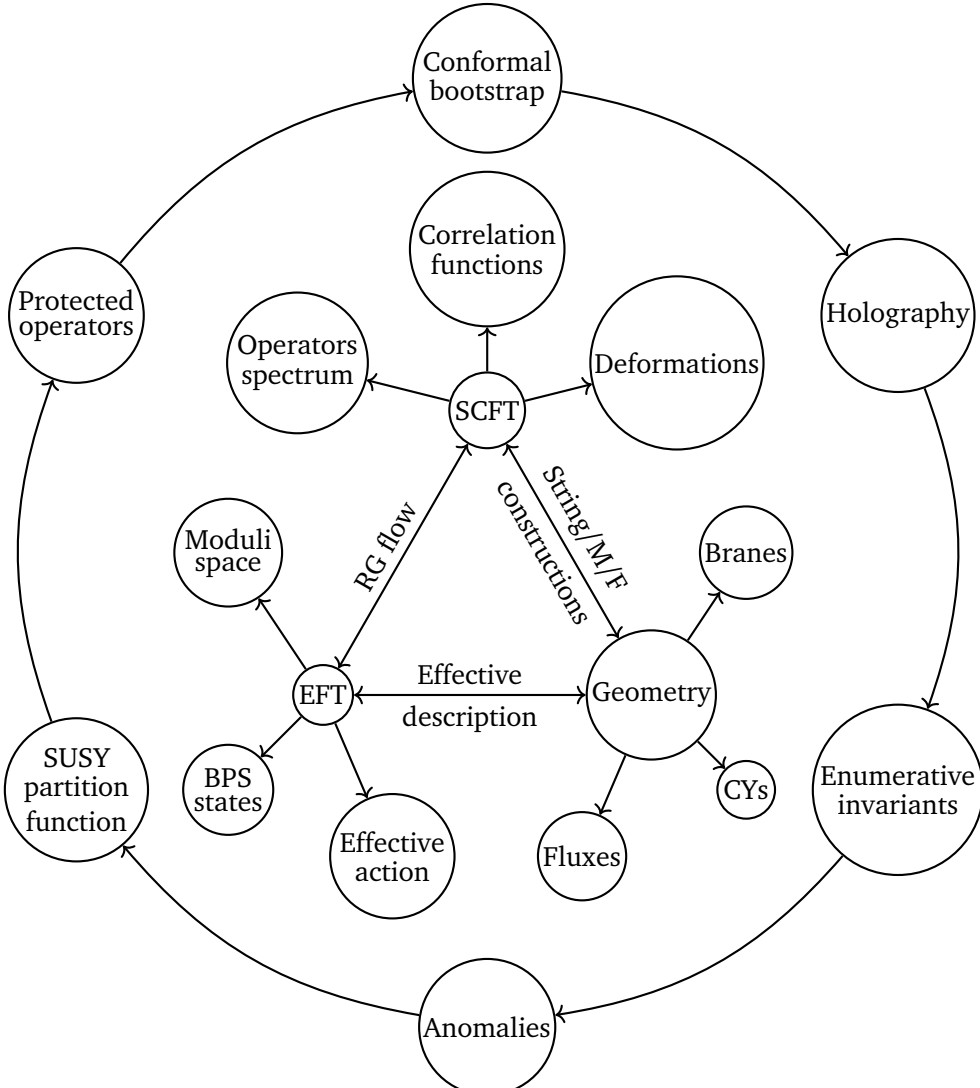

Figure 1: General philosophy behind the approaches to study 4d $\mathcal{N} = 2$ SCFTs. In the center are the three closely related and complementary approaches to SCFTs discussed in these lecture notes, based on bootstrap philosophy, EFT method and geometric engineering. For each of these approaches, the basic building blocks are listed in the nearby circles. They further give rise to observables with natural interpretations in each of these approaches. Some representatives of these observables are listed in the outer ring, which describe various aspects of the same SCFT and for that reason obey nontrivial mathematical relations.

## 2 Basics of 4d $\mathcal{N} = 2$ Supersymmetry

In this section, we review basic concepts of 4d $\mathcal{N} = 2$ supersymmetry in the context of SCFTs. In Section 2.1 we introduce the unitary irreducible representation of $\mathcal{N} = 2$ superconformal algebra. In Section 2.2 we introduce the super-Poincaré algebra with 2 complex Weyl spinor supercharges, give the expressions for the superfields in the $\mathcal{N} = 1$ notation, and write the most general $\mathcal{N} = 2$ Lagrangian for a gauge theory with matter. The supersymmetric vacua of Coulomb Branch and Higgs Branch are defined in Section 2.3. We explain the general non-renormalization theorem for these theories and their behavior under renormalization group flow in Section 2.4. In Section 2.5 we connect these Lagrangian descriptions to the SCFTs.

## 2.1 $\mathcal{N} = 2$ superconformal Symmetry and Representations

We start by giving a brief overview of the 4d $\mathcal{N} = 2$ superconformal algebra and its representations. In 4d flat spacetime $\mathbb{R}^{3,1}$, the conformal algebra $\mathfrak{so}(4, 2)$ is generated by Lorentz transformation $M^{\mu}_{\nu} \in \mathfrak{so}(3, 1)$, translation $P^{\mu}$, dilatation $D$, and special conformal transformation $K_{\mu}$, where $\mu, \nu = 0, 1, 2, 3$ are spacetime indices. Note that the dilatation $D$ generates the Abelian subalgebra $\mathfrak{so}(1, 1)$, and its eigenvalues correspond to the scaling dimension (weight) $\Delta$. In particular, the translation generator $P^{\mu}$ has weight $\Delta = 1$, whereas the special conformal transformation has weight $\Delta = -1$, which follows from their commutation relations with $D$. For more details of the conformal algebra, we refer the readers to [95, 96].

The conformal symmetry can be extended by Poincaré supersymmetry in 4d spacetime.[4] When adding $\mathcal{N}$ supercharges transforming in the Weyl representation of the Lorentz algebra $\mathfrak{so}(3, 1)$, the conformal algebra $\mathfrak{so}(4, 2)$ is enhanced to the superconformal algebra $\mathfrak{su}(2, 2|\mathcal{N})$.[5] The 4d $\mathcal{N} = 2$ superconformal algebra contains the following maximal bosonic subalgebra

$$\mathfrak{su}(2, 2|2) \supset \mathfrak{so}(4, 2) \oplus \mathfrak{su}(2)_R \oplus \mathfrak{u}(1)_r, \tag{2.1}$$

where $\mathfrak{su}(2)_R \oplus \mathfrak{u}(1)_r$ is the R-symmetry algebra whose generators are $R_{(mn)}$ and $r$ respectively where $m, n = 1, 2$ are doublet indices for $\mathfrak{su}(2)_R$. The set of superconformal generators, comparing to the above conformal generators, is now enlarged by the Poincaré supercharges $\mathcal{Q}^m_{\alpha}, \widetilde{\mathcal{Q}}^n_{\dot{\alpha}}$, together with the superconformal partners $\mathcal{S}^m_{\alpha}, \widetilde{\mathcal{S}}^n_{\dot{\alpha}}$, as well as the R-symmetry generators. Here $(\alpha, \dot{\alpha})$ denote the (anti)chiral spinor indices under the Lorentz algebra $\mathfrak{so}(3, 1)$. Note that these supercharges have the following schematic commutation relations

$$\{\mathcal{Q}, \widetilde{\mathcal{Q}}\} \sim P_{\mu}, \quad \{\mathcal{S}, \widetilde{\mathcal{S}}\} \sim K_{\mu}, \quad \{\mathcal{Q}, \mathcal{S}\} \sim D + M_{\mu\nu}\sigma^{\mu\nu}_{\alpha\beta} + R_{mn} + r, \tag{2.2}$$

and similarly for $\{\widetilde{\mathcal{Q}}, \widetilde{\mathcal{S}}\}$. Their scaling dimensions and charges under $\mathfrak{u}(1)_r$ in our convention are listed in (2.3).

|  | $\mathcal{Q}^m_{\alpha}$ | $\widetilde{\mathcal{Q}}^m_{\dot{\alpha}}$ | $\mathcal{S}^m_{\alpha}$ | $\widetilde{\mathcal{S}}^m_{\dot{\alpha}}$ |
|---|---|---|---|---|
| $\mathfrak{u}(1)_r$ | $\frac{1}{2}$ | $-\frac{1}{2}$ | $-\frac{1}{2}$ | $\frac{1}{2}$ |
| $\Delta$ | $\frac{1}{2}$ | $\frac{1}{2}$ | $-\frac{1}{2}$ | $-\frac{1}{2}$ |

$$\tag{2.3}$$

We would like to stress that the fact that R-symmetries belong to superconformal algebras and are thus genuine symmetries of the theory is a hallmark of SCFTs, which, in contrast, are generally not symmetries in non-conformal supersymmetric theories.

The operator content of the SCFT is organized with respect to the superconformal symmetry. The operators with spacetime quantum numbers $j, \bar{j} \in \mathbb{Z}$ with respect to $\mathfrak{so}(3, 1) \sim \mathfrak{su}(2) \oplus \mathfrak{su}(2)$ are further labelled by the eigenvalues of additional Cartan generators of the $\mathcal{N} = 2$ superconformal algebra as $[j, \bar{j}]^{(R;r)}_{\Delta}$, where $\Delta$ is the scaling dimension, $R \in \mathbb{Z}$ is twice the $\mathfrak{su}(2)_R$ spin and $r$ labels the $\mathfrak{u}(1)_r$ charge.

In a unitary SCFT, the local operators form irreducible unitary representations of the superconformal algebra. As for general CFTs, under radial quantization in flat space, these operators are in one-to-one correspondence with states in the Hilbert space on $S^3$, organized into the same representations.[6] Unitarity requires a lowest weight state in each such representation,

---

[4]Note that the maximal spacetime dimension to incorporate Poincaré supersymmetry with conformal symmetry consistently is 6d [97].

[5]When $\mathcal{N} = 4$, the algebra $\mathfrak{su}(2, 2|4)$ is not simple and the physically relevant superconformal algebra is given by its quotient $\mathfrak{psu}(2, 2|4)$. The extra $\mathfrak{u}(1)$ is known as the bonus symmetry in [98, 99].

[6]More explicitly under the state-operator correspondence, every local operator $[j, \bar{j}]^{(R,r)}_{\Delta}$ is identified with a unique state $|[j, \bar{j}]^{(R,r)}_{\Delta}\rangle$, and we will frequently drop out $|\dots\rangle$ for states in this section. We note that this bijection between states and local operators is only possible in CFTs.

known as the **superconformal primary**, which satisfies

$$\mathcal{S}_\alpha^m[j,\bar{j}]_\Delta^{(R;r)} = 0\,,\quad \widetilde{\mathcal{S}}_{\dot\alpha}^m[j,\bar{j}]_\Delta^{(R;r)} = 0\,,\quad K^\mu[j,\bar{j}]_\Delta^{(R;r)} = 0\,, \tag{2.4}$$

where the last equality which defines an ordinary conformal primary follows from the first two and (2.2). All the other states in a given irreducible representation can be obtained by acting with $(\mathcal{Q},\widetilde{\mathcal{Q}})$-supercharges on these superconformal primary states,

$$\mathcal{Q}_\alpha^m \cdots \widetilde{\mathcal{Q}}_{\dot\beta}^n[j,\bar{j}]_\Delta^{(R;r)}\,, \tag{2.5}$$

and hence dubbed (supersymmetric) **descendants** at level $l$, where $l$ counts the number of supercharges that appear above. Note that some of the descendants are also conformal primaries (e.g. the level-one $(\mathcal{Q},\widetilde{\mathcal{Q}})$ descendants due to the commutation relation $[K,\mathcal{Q}] \sim \widetilde{\mathcal{S}}$ and $[K,\widetilde{\mathcal{Q}}] \sim \mathcal{S}$). Furthermore, due to the fermionic nature of the supercharges, the number of conformal primaries in a given superconformal representation is finite. More precisely, in 4d $\mathcal{N} = 2$ SCFT with eight real supercharges, one generically has $2^8 = 64$ conformal primaries in a superconformal representation (multiplet). We will refer to those multiplets as **long multiplets**. Unitarity requires the norms of all states to be non-negative, and thus leads to non-trivial constraints on superconformal representations. In particular, it imposes lower bounds on the scaling dimension $\Delta$ of the superconformal primary $\mathcal{V}$ in terms of its quantum numbers under the maximal bosonic subalgebra, of the form

$$\Delta \geqslant \Delta_\mathcal{V} = f(j,\bar{j},R,r)\,. \tag{2.6}$$

This is known as the **unitary bound**. When the bound is saturated, the superconformal representation becomes reducible, and contains a sub-representation formed by the states with zero norm, known as the **null states**.

---

**Exercise 2.1** Show that the null states in a unitary representation of the (super)conformal algebra form a lowest weight representation by themselves.

---

We can consistently remove the null states by taking the quotient, and the resulting irreducible superconformal multiplet contains fewer operators compared with the long multiplet. Hence, we will refer to those as **short multiplets**. The relations a superconformal primary has to satisfy to saturate the unitarity bound are therefore referred to as **shortening conditions**. Given the complexity of 4d $\mathcal{N} = 2$ superconformal algebra, it is natural to expect there can be several shortening conditions. First, depending on the chirality of the supercharges involved in constructing the null states, there are independent chiral and anti-chiral shortening conditions. The full $\mathcal{N} = 2$ multiplet is thus in general specified by a pair of shortening conditions on the two sides (see Table 1). Below we focus on the chiral shortening conditions, which are further classified into two shortening types, labeled as $A, B$ as in [5] (the anti-chiral shortening types are denoted by $\bar{A}, \bar{B}$). Each shortening type can have several shortening conditions, with the corresponding unitary bounds schematically written as,

$$\begin{aligned}
\Delta_\mathcal{V} &\geq f(j,\bar{j},R,r) + \delta_A\,,\quad j,\bar{j},R,r \text{ unrestricted}\,,\\
\Delta_\mathcal{V} &= f(j,\bar{j},R,r) + \delta_B\,,\quad j,\bar{j},R,r \text{ restricted}\,,
\end{aligned} \tag{2.7}$$

where the function $f(j,\bar{j},R,r)$ is the same for $A, B$ and there are constant offsets $\delta_{A,B}$ satisfying $\delta_A > \delta_B$. By unrestricted, we mean $A$ can happen for any allowed Lorentz representation, while the shortening type $B$ can appear only if the quantum number of $\mathcal{V}$ satisfies certain conditions. Each shortening condition is further distinguished by the chirality of the supercharges involved. Notice that there is an important distinction between type $A$ and $B$ which we come to now.

For type $A$ in (2.7), we get a lower bound on the allowed scaling dimensions. If the bound is saturated, then the superconformal representation will contain null states to be removed. Above the bound, we have a generic long multiplet (which we will denote as $L$). Thus, this type of short representation are referred to as **short multiplets at threshold** and will be denoted as

$$A_l[j,\bar{j}]^{(R;r)}_{\Delta_A}, \quad l = 1, 2, \tag{2.8}$$

where the first null state is of the form $\mathcal{Q}^l[j,\bar{j}]_{\Delta_A}$. It is also important to notice that the null states themselves form a short representation, and such representation would be unitary if its superconformal primary had positive norm.

For type $B$ in (2.7), the constraint on the quantum numbers is a strict equality. This means the type $B$ short multiplets are isolated from the other unitary representations with the same Lorentz and R-symmetry quantum numbers by a finite gap. Such representation will similarly be denoted as

$$B_1[j,\bar{j}]^{(R;r)}_{\Delta_B}, \tag{2.9}$$

with the first null states given by the level 1 descendants with respect to $\mathcal{Q}$ respectively. Unlike type-$A$, the null states removed here still form a sub-representation but cannot be promoted to a separate unitary representation.

This structure leads to the notion of **recombination rules**. We can imagine gradually lowering the scaling dimension $\Delta_\mathcal{V}$ of a generic long multiplet. Eventually it will hit the unitarity bound from above, and fragment into an $A$-type short multiplet plus the short representation $N$ (which may be reducible) containing all the null states:

$$L[j,\bar{j}]^{(R;r)}_{\Delta} \xrightarrow{\Delta \to \Delta_A^+} A_l[j,\bar{j}]^{(R;r)}_{\Delta_A} \oplus N[j_N,\bar{j}_N]^{(R_N;r_N)}_{\Delta_N = \Delta_A + l/2}. \tag{2.10}$$

Such a phenomenon shows the spectrum of short multiplets can change under continuous deformations preserving the superconformal symmetry, which we refer to as the conformal manifold. As we move along the conformal manifold, some long multiplets may become short ones, while some short multiplets may combine to form long multiplets. Hence, the spectrum of the short multiplets is only protected modulo the recombination rules, and such data is captured by the superconformal indices [100]. However, if some short multiplets never enter the RHS of the recombination rules, they are truly protected by superconformal algebra and can be tracked unambiguously along the conformal manifold. Those multiplets will be referred to as **absolutely protected** [5].[7] It is important to notice that some $A$-type operators can be absolutely protected as well. For the case of $\mathcal{N} = 2$ SCFTs, the multiplets are in Table 1 taken from [5] (see also [96] for a more recent review). Note that the full $\mathcal{N} = 2$ superconformal multiplet generally involve a pair of chiral and anti-chiral shortening conditions.

For our later purposes, we single out two important short multiplets:

> ### Definition 2.1: Coulomb branch multiplets
>
> Coulomb branch multiplets are short multiplets of the type $\overline{L}B_1[0,0]^{(0;r)}$ for $r > 1$ and $\overline{A}_2B_1[0,0]^{(0;r)}$ for $r = 1$ (corresponding to a free vector multiplet). The superconformal primaries are scalar operators with $\Delta = r$ and are called Coulomb branch **chiral primaries** which generate the Coulomb branch chiral ring. There are also the anti-chiral primaries, corresponding to the conjugate representations of the chiral primaries.

---

[7]See also earlier works [101–112] on the non-renormalization properties of such chiral operators.

Table 1: Unitary 4d $\mathcal{N}=2$ superconformal multiplets (adapted from [5] with a minor modification on the $U(1)_r$ charges as in (2.3)).

|  | $\bar{L}$ | $\bar{A}_1$ | $\bar{A}_2$ | $\bar{B}_1$ |
|---|---|---|---|---|
| $L$ | $[j,\bar{j}]_\Delta^{(R;r)}$ <br> $\Delta > 2 + R + \max\{j-r,\bar{j}+r\}$ | $[j,\bar{j}\geq 1]_\Delta^{\left(R;r<\frac{\bar{j}-j}{2}\right)}$ <br> $\Delta = 2+R+\bar{j}-r$ | $[j,\bar{j}=0]_\Delta^{\left(R;r<-\frac{j}{2}\right)}$ <br> $\Delta = 2+R-r$ | $[j,\bar{j}=0]_\Delta^{\left(R;r<-\frac{j+2}{2}\right)}$ <br> $\Delta = R-r$ |
| $A_1$ | $[j\geq 1,\bar{j}]_\Delta^{\left(R;r>\frac{\bar{j}-j}{2}\right)}$ <br> $\Delta = 2+R+j+r$ | $[j\geq 1,\bar{j}\geq 1]_\Delta^{\left(R;r=\frac{\bar{j}-j}{2}\right)}$ <br> $\Delta = 2+R+\frac{1}{2}(j+\bar{j})$ | $[j\geq 1,\bar{j}=0]_\Delta^{\left(R;r=-\frac{j}{2}\right)}$ <br> $\Delta = 2+R+\frac{1}{2}j$ | $[j\geq 1,\bar{j}=0]_\Delta^{\left(R;r=-\frac{j+2}{2}\right)}$ <br> $\Delta = 1+R+\frac{1}{2}j$ |
| $A_2$ | $[j=0,\bar{j}]_\Delta^{\left(R;r>\frac{\bar{j}}{2}\right)}$ <br> $\Delta = 2+R+r$ | $[j=0,\bar{j}\geq 1]_\Delta^{\left(R;r=\frac{\bar{j}}{2}\right)}$ <br> $\Delta = 2+R+\frac{1}{2}\bar{j}$ | $[j=0,\bar{j}=0]_\Delta^{(R;r=0)}$ <br> $\Delta = 2+R$ | $[j=0,\bar{j}=0]_\Delta^{(R;r=-1)}$ <br> $\Delta = 1+R$ |
| $B_1$ | $[j=0,\bar{j}]_\Delta^{\left(R;r>\frac{\bar{j}+2}{2}\right)}$ <br> $\Delta = R+r$ | $[j=0,\bar{j}\geq 1]_\Delta^{\left(R;r=\frac{\bar{j}+2}{2}\right)}$ <br> $\Delta = 1+R+\frac{1}{2}\bar{j}$ | $[j=0,\bar{j}=0]_\Delta^{(R;r=1)}$ <br> $\Delta = 1+R$ | $[j=0,\bar{j}=0]_\Delta^{(R;r=0)}$ <br> $\Delta = R$ |

### Definition 2.2: Higgs branch multiplets

Higgs branch multiplets are short multiplets of the type $B_1\overline{B}_1[0,0]^{(R;0)}$. The superconformal primaries are scalars with weight $\Delta = R$ and uncharged under $\mathfrak{u}(1)_r$, but they transform in non-trivial $\mathfrak{su}(2)_R$ representations of dimension $R+1$. They are called Higgs branch **chiral primaries** and generate the Higgs branch chiral ring.

An important feature of these multiplets is that their superconformal primaries can develop continuous VEVs which spontaneously break the conformal symmetry but preserve the $\mathcal{N}=2$ super-Poincaré symmetry, leading to a moduli space of supersymmetric vacua for the SCFT. Depending on which types of superconformal primaries obtain a VEV, the corresponding branch of the moduli space is commonly referred to as the Coulomb, Higgs, and mixed branches, as suggested by the names of the corresponding multiplets.

## 2.2   4d $\mathcal{N}=2$ Multiplets and General $\mathcal{N}=2$ Lagrangians

So far we have focused on general aspects of operators in an $\mathcal{N}=2$ SCFT, which is completely universal but somewhat abstract. It is often educational to look for and study realizations of a CFT and its operator spectrum in terms of its Lagrangian descriptions. To do so, we here review general aspects of $\mathcal{N}=2$ Lagrangian theories (not necessarily conformal).

Let us start by looking at the 4d $\mathcal{N}=2$ super-Poincaré algebra more closely. As alluded before, it arises from an extension of the 4d Poincaré algebra $(P_\mu, M_{\mu\nu})$ by two supercharges $\mathcal{Q}, \tilde{\mathcal{Q}}$, with the corresponding quantum numbers,

$$\left(\mathcal{Q}_\alpha^m, \widetilde{\mathcal{Q}}_{\dot{\alpha}}^m\right) \in \left([1,0]_{\frac{1}{2}}^{\left(1;\frac{1}{2}\right)}, [0,1]_{\frac{1}{2}}^{\left(1;-\frac{1}{2}\right)}\right), \tag{2.11}$$

which have the following non-trivial non-commutative relations (see e.g., [113–116] for the complete set of commutation and anti-commutation relations)

$$\begin{aligned}
\left\{\mathcal{Q}_\alpha^m, \widetilde{\mathcal{Q}}_{\dot{\beta}}^n\right\} &= 2(\sigma^\mu)_{\alpha\dot{\beta}}\,\epsilon^{mn}P_\mu\,, \\
\left\{\mathcal{Q}_\alpha^m, \mathcal{Q}_\beta^n\right\} &= \delta_{\alpha\beta}\epsilon^{mn}\mathbf{Z}\,,
\end{aligned} \tag{2.12}$$

where $\sigma^\mu = \{\mathbb{1}, \sigma^i\}$ are 4d gamma matrices in the chiral representation with $\sigma^i$ being the usual Pauli matrices, $\epsilon^{mn}$ is the Levi-Civita tensor and $P_\mu$ is the generator of the translations in 4d. More interesting is $\mathbf{Z}$, known as the central charge, which represents a central extension

of this anti-commutation relation. The central extension will play an important role in the description of the effective field theory of the 4d $\mathcal{N} = 2$ SCFT in the IR. The last thing to notice is that in our normalization, from the charges under the $\mathfrak{u}(1)_r$ of the supercharges in Eq. (2.3), $Z$ will carry charge $+1$.

To construct a 4d $\mathcal{N} = 2$ Lagrangian, we need to know the irreducible representations of the above supersymmetry algebra, which split into two categories: the **vector multiplet** and the **hypermultiplet**.

- The **vector multiplet** $V_{\mathcal{N}=2}$ contains one complex scalar $\Phi$ and two complex Weyl gaugino $\lambda_\alpha, \tilde{\lambda}_{\dot{\alpha}}$ and one antisymmetric two-form $F_{\mu\nu} := \partial_\mu A_\nu + [A_\mu, A_\nu]$, all in the adjoint representation of the gauge group $G$. The whole multiplet can be constructed from the scalar field $\Phi$, which is further annihilated by the chiral supercharges,

$$\mathcal{Q}^m |\Phi\rangle = 0, \quad m = 1, 2. \tag{2.13}$$

Then one can obtain the other fields as the supersymmetric descendants by acting with the anti-chiral supercharges $\widetilde{\mathcal{Q}}^n, n = 1, 2$ on the state $|\Phi\rangle$: acting once we obtain the two gauginos $\lambda_\alpha, \tilde{\lambda}_{\dot{\alpha}}$ and twice the field strength $F_{\mu\nu}$. Once again, the superscripts $m, n$ represent the $SU(2)_R$ symmetry doublet indices. The scalar field $\Phi$ has charge 1 under the $U(1)_r$ R-symmetry in our notation and from the anti-commutation relations of the supersymmetry algebra, we see that these two fields $\lambda_\alpha^n, A_\mu$ have respectively charge $1/2$ and 0 under the $U(1)_r$ R-symmetry. Regarded as a free 4d $\mathcal{N} = 2$ SCFT, an $\mathcal{N} = 2$ vector multiplet $V_{\mathcal{N}=2}$ realizes the short multiplet $A_2\bar{B}_1[0,0]_1^{(0;1)}$ introduced in Table 1, in which $\Phi$ is the superconformal primary.

- The **hypermultiplet** $H_{\mathcal{N}=2}$ involves two complex scalars $(q, \tilde{q})$ and Weyl fermions $(\psi_\alpha, \tilde{\psi}_{\dot{\alpha}})$. The scalars are annihilated by half of the supercharges, albeit of different chiralities, i.e.,

$$\mathcal{Q}^1 |q\rangle = \widetilde{\mathcal{Q}}^1 |q\rangle = \mathcal{Q}^1 |\tilde{q}\rangle = \widetilde{\mathcal{Q}}^1 |\tilde{q}\rangle = 0, \tag{2.14}$$

where the superscript 1 is the $SU(2)_R$ doublet index with Cartan charge $+\frac{1}{2}$, and the same charge is carried by the scalars $(q, \tilde{q})$. Now we can act with the other supercharges to get the supersymmetric descendant, which are fermions. The scalars are neutral under the $U(1)_r$ R-symmetry, while $SU(2)_R$ acts on $(q, (\tilde{q})^\dagger)$ as a doublet. The fermions on $SU(2)_R$ invariant and carry charges under the $U(1)_r$ symmetry respectively of $\pm 1/2$. The hypermultiplet also has an extra $SU(2)$ flavor symmetry, which commutes with the R-symmetries and under which the scalars and the fermions transform as doublets. In the context of 4d $\mathcal{N} = 2$ superconformal representations, such a hypermultiplet realizes the short multiplet $B_1\bar{B}_1[0,0]_1^{(1;0)}$. In general, if we have $n$ hypermultiplets, the flavor symmetry is enlarged to $USp(2n)$ and the scalars are denoted by $(q^A, \tilde{q}_A)$ where $A$ is the index for the $2n$-dimensional fundamental representation of $USp(2n)$.

Having introduced the 4d $\mathcal{N} = 2$ supersymmetric multiplets, we now can write down the general form of 4d $\mathcal{N} = 2$ Lagrangians. The supersymmetry imposes strong restrictions on the forms of Lagrangians and there is a systematic and efficient way to construct a 4d $\mathcal{N} = 2$ Lagrangian employing the above supermultiplets using the superspace formalism of 4d $\mathcal{N} = 1$. In that formalism, by extending ordinary spacetime to superspace with additional Grassmannian coordinates $(\theta^\alpha, \bar{\theta}_{\dot{\alpha}})$, we can construct various 4d $\mathcal{N} = 1$ superfields. Relevant for us, one of the important $\mathcal{N} = 1$ superfields is known as the real vector superfield $V$, which repackages a Weyl fermion $\lambda_\alpha$ and a vector field $A_\mu$ in a vector multiplet into a compact form as[8]

$$V = -i\theta\sigma^\mu\bar{\theta}A_\mu + i\theta\theta\bar{\theta}\bar{\lambda} - i\bar{\theta}\bar{\theta}\theta\lambda + \frac{1}{2}\theta\theta\bar{\theta}\bar{\theta}D, \tag{2.15}$$

---

[8]Here we choose the Wess-Zumino gauge.

where $D$ is an auxiliary field. Another superfield containing the matter fields in a 4d $\mathcal{N} = 1$ chiral multiplet is known as chiral superfield $Q$

$$Q^A = q^A + i\psi^A\theta + F^A\theta\theta\,, \tag{2.16}$$

which carries $\mathcal{N} = 1$ $U(1)_r$ charge denoted by $r_{\mathcal{N}=1}$. Again, $A$ is the flavor index and $F^A$ is an auxiliary field, and it transforms under the supersymmetry by a total derivative.

Then, an $\mathcal{N} = 2$ vector multiplet can be represented as a combination of an $\mathcal{N} = 1$ vector superfield and an $\mathcal{N} = 1$ chiral superfield:

$$V_{\mathcal{N}=2} \to (V, Q)\,, \tag{2.17}$$

where the chiral superfield $Q$ is in the adjoint representation of the gauge group $G$ with $r_{\mathcal{N}=1} = \frac{2}{3}$ . Here we have used the relations between the $\mathcal{N} = 1$ and $\mathcal{N} = 2$ superconformal R-symmetries,

$$r_{\mathcal{N}=1} = \frac{2}{3}r_{\mathcal{N}=2} + \frac{4}{3}I_3\,, \tag{2.18}$$

where $I_3$ is the Cartan generator of $SU(2)_R$. On the other hand, an $\mathcal{N} = 2$ hypermultiplet can be represented by two $\mathcal{N} = 1$ chiral superfields:

$$H_{\mathcal{N}=2} \to (Q, \widetilde{Q})\,, \tag{2.19}$$

again with $r_{\mathcal{N}=1} = \frac{2}{3}$.

The most general 4d $\mathcal{N} = 1$ supersymmetric Lagrangian can be constructed from a superpotential for the chiral superfields and a Kähler potential that may involve couplings between the chiral superfields and the real vector superfield $V$. By restricting the field content, and for special choices of the superpotential and Kähler potential, the Lagrangian has enhanced $\mathcal{N} = 2$ supersymmetry. Schematically, such a Lagrangian takes the following form

$$\mathcal{L}_{\mathcal{N}=2} = \mathcal{L}_{\mathcal{N}=2}^{\mathrm{SYM}} + \mathcal{L}_{\mathcal{N}=2}^{\mathrm{matter}}\,, \tag{2.20}$$

where the first term contains the contribution from 4d $\mathcal{N} = 2$ vector multiplets and the second one contains the contribution from $\mathcal{N} = 2$ hypermultiplets (possibly coupled to the vector multiplets).

In order to write down a gauge invariant action for the vector multiplet that supersymmetrize the familiar Yang-Mills action $F_{\mu\nu}F^{\mu\nu}$, it is convenient to rewrite the vector multiplet $V$ into the so-called gaugino superfield $W_\alpha$ , as $V$ does not directly contain the gauge field strength $F_{\mu\nu}$, which has the following expansion:

$$W_\alpha = -i\lambda_\alpha + iF_{\mu\nu}(\sigma^{\mu\nu}\theta)_\alpha + D\theta_\alpha + \theta\theta(\sigma^\mu\partial_\mu\bar{\lambda}_\alpha)\,, \tag{2.21}$$

whose lowest component $\lambda_\alpha$ is the Weyl gaugino with $r_{\mathcal{N}=1} = 1$. This is related to $V$ by supercovariant derivatives $D_\alpha \equiv \frac{\partial}{\partial\theta^\alpha} + i\sigma^\mu_{\dot\alpha\beta}\bar\theta^{\dot\beta}\partial_\mu$ as

$$W_\alpha = -\frac{1}{4}\overline{D}^2 D_\alpha V\,. \tag{2.22}$$

Now an $\mathcal{N} = 2$ vector multiplet from (2.17) splits into an $\mathcal{N} = 1$ gaugino superfield $W_\alpha$ and an $\mathcal{N} = 1$ chiral superfield $\Phi$. The corresponding $\mathcal{N} = 2$ super-Yang-Mills Lagrangian takes the form of the following superspace integral

$$\mathcal{L}_{\mathcal{N}=2}^{\mathrm{SYM}} = \frac{1}{8\pi i}\int d^2\theta\,\mathrm{Tr}\left(\tau W_\alpha W^\alpha + \mathrm{c.c.}\right) + \frac{\mathrm{Im}(\tau)}{4\pi}\int d^4\theta\,\mathrm{Tr}\left(\Phi^\dagger e^{\mathrm{adj}(V)}\Phi\right)\,. \tag{2.23}$$

Here the first piece encodes the $\mathcal{N}=1$ SYM kinetic term, while the second term represents the $\mathcal{N}=1$ kinetic term for the chiral superfield in minimal coupling with the vector superfield in the adjoint representation of $G$. The complexified gauge coupling $\tau$ contains both the Yang-Mills coupling and the theta angle,

$$\tau = \frac{\theta_{\text{YM}}}{2\pi} + \frac{4\pi i}{g^2}\,. \tag{2.24}$$

The relative prefactor in (2.23) is fixed by $SU(2)_R$ symmetry, which ensures that the whole Lagrangian preserves $\mathcal{N}=2$ SUSY. The $\mathcal{N}=2$ Lagrangian for the hypermultiplets contains the kinetic terms for $(Q,\widetilde{Q})$ of (2.19) in minimal coupling with the real vector superfield $V$ in some representation $\rho$ of the gauge group $G$, i.e.,[9]

$$\mathcal{L}_{\mathcal{N}=2}^{\text{matter}} = \int d^4\theta \left[ Q^\dagger e^{\rho(V)} Q + \widetilde{Q}^\dagger e^{\overline{\rho}(V)} \widetilde{Q} \right] + \int d^2\theta \, \widetilde{Q} \rho(\Phi) Q + \text{c.c.} \tag{2.25}$$

The second piece, written as an $\mathcal{N}=1$ superpotential,[10] encodes part of the interaction between an $\mathcal{N}=2$ vector multiplet and a hypermultiplet, and the coupling is fixed to 1 by the $SU(2)_R$ symmetry. Note that from the above, one can see that there are no parameters in the general $\mathcal{N}=2$ massless Lagrangian except for the complexified gauge coupling $\tau$.

## 2.3 Supersymmetric Vacuum Moduli Space

We have constructed in the previous section the most general $\mathcal{N}=2$ Lagrangian that involves vector multiplets and hypermultiplets, and one concrete thing we can study via this Lagrangian is the (classical) **moduli space** of vacua. This will come from the scalar potential $V(\Phi,q,\widetilde{q})$ which is obtained from the Lagrangian (2.20) after integrating out the auxiliary fields $D$ in the vector superfield and $F$ in the chiral superfields. The scalar potential $V$ is a sum of squares corresponding to the so-called D-terms and F-terms [117],

$$V(\Phi,q,\widetilde{q}) = \frac{1}{2}\text{Tr}\left(D^2\right) + F\overline{F}\,, \tag{2.26}$$

where the auxiliary fields $D$ and $F$ are determined in terms of the physical scalar fields $\Phi,q,\widetilde{q}$ as we will see below.

The moduli space of supersymmetric vacua is the zero locus of the scalar potential,

$$\mathcal{M}^{classical} = \{\Phi,q,\widetilde{q} \,|\, V(\Phi,q,\widetilde{q}) = 0\}\,, \tag{2.27}$$

and equivalently $D(\Phi,q,\widetilde{q}) = F(\Phi,q,\widetilde{q}) = 0$.[11]

To be concrete, in the following let us consider SQCD with $G = SU(N)$ gauge group and $N_F$ fundamental matter $H_{\mathcal{N}=2}^A, A = 1,\ldots,N_f$, but the equations we write will hold for general $\mathcal{N}=2$ Lagrangian theories with suitable modifications for the indices. The D-term takes the following form,

$$D = \frac{1}{g^2}\left[\Phi,\Phi^\dagger\right] + \left(q_A(q^A)^\dagger - (\widetilde{q}_A)^\dagger\widetilde{q}^A\right)\big|_{\text{traceless}} = 0\,, \tag{2.28}$$

while the F-terms are

$$\begin{aligned}
F_\Phi &= q_A\widetilde{q}^A\big|_{\text{traceless}} = 0\,,\\
F_q &= \Phi q_A = 0\,,\\
F_{\widetilde{q}} &= \widetilde{q}^A\Phi = 0\,.
\end{aligned} \tag{2.29}$$

---

[9]In generic cases, one can also add a mass term for the hypermultiplets $\int d^2\theta\, m\widetilde{Q}Q$. For simplicity, we omit this term for the following discussions.

[10]Recall that the $\mathcal{N}=1$ superfields $Q,\tilde{Q},\Phi$ all have $r_{\mathcal{N}=1} = \frac{2}{3}$.

[11]The vanishing condition on the auxiliary fields also follows from requiring the supersymmetry variation of the fermionic fields to vanish, which is necessary for the configuration to be supersymmetric.

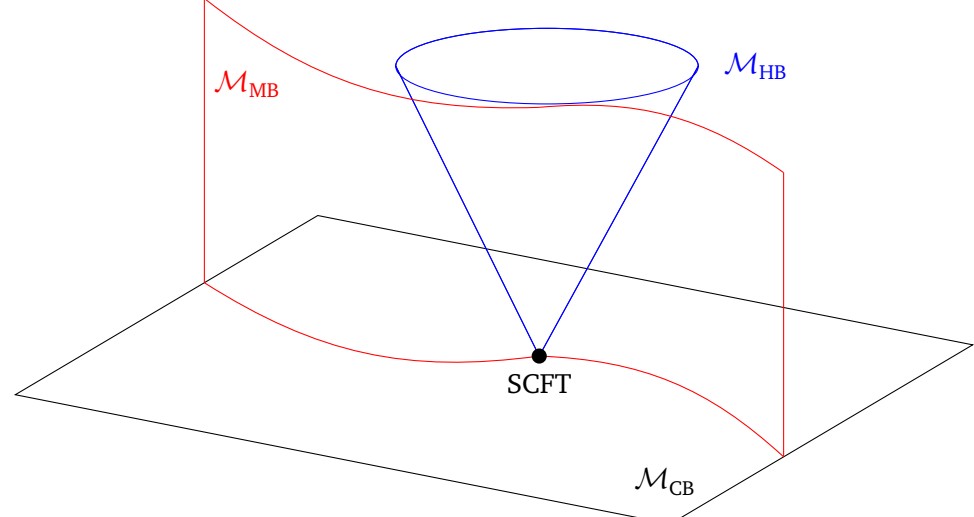

Figure 2: The vacuum moduli space for a generic $\mathcal{N} = 2$ theory.

Here $\Phi$ is in the adjoint representation of the gauge group $SU(N)$ and $q_A$ ($\tilde{q}^A$) denotes the scalar components of $H^A_{\mathcal{N}=2}$ and transform as fundamental (anti-fundamental) representations of the gauge group $SU(N)$ and the flavor group $SU(N_F)$. The relations in Eqs. (2.28) and (2.29) can be simplified to the following sets of equations:

$$\left[\Phi, \Phi^\dagger\right] = 0, \tag{2.30}$$

$$\begin{cases} \left(q_A(q^A)^\dagger - (\tilde{q}_A)^\dagger \tilde{q}^A\right)\big|_{\text{traceless}} &= 0, \\ q_A \tilde{q}^A \big|_{\text{traceless}} &= 0, \end{cases} \tag{2.31}$$

$$\begin{cases} \Phi q_A = \tilde{q}^A \Phi^\dagger &= 0, \\ \Phi^\dagger q_A = \tilde{q}^A \Phi &= 0. \end{cases} \tag{2.32}$$

---

**Exercise 2.2** Bring Eqs. (2.28) and (2.29) in the form of Eqs. (2.30) to (2.32).

---

One should note that Eqs. (2.30) to (2.32) have the following features. They are organized into three $\mathfrak{su}(2)_R$ multiplets, which ensures that the solutions preserve the full $\mathcal{N} = 2$ supersymmetry. Furthermore, the solutions to Eqs. (2.30) to (2.32) have three kinds of branches, schematically depicted in Figure 2, corresponding to Coulomb Branch, Higgs Branch and Mixed Branch, depending on which R-symmetry subgroups are broken.

### Definition 2.3: Coulomb Branch $\mathcal{M}_{\text{CB}}$

The Coulomb Branch (CB) $\mathcal{M}_{\text{CB}}$ is the case when the hypermultiplet scalars $q$ and $\tilde{q}$ have vanishing VEVs. The only non-trivial equation to be imposed is (2.30). This demands the vector multiplet scalar $\Phi$ to take value in the Cartan subalgebra of the gauge algebra $\mathfrak{g}$ up to gauge transformations. Correspondingly, the complex dimension of $\mathcal{M}_{\text{CB}}$ is

$$\dim_{\mathbb{C}}(\mathcal{M}_{\text{CB}}) = \text{rank}(G). \tag{2.33}$$

In terms of gauge invariant operators, $\mathcal{M}_{\text{CB}}$ is parameterized by the VEVs of the CB operators introduced in Def. 2.1.[12] Since these operators carry non-trivial $U(1)_r$ charges but are $SU(2)_R$ singlets, $U(1)_r$ is broken along $\mathcal{M}_{\text{CB}}$ while $SU(2)_R$ is preserved.

---

[12]Note that these operators make sense with or without conformal symmetry. In fact, the CB chiral primaries

For example, when the gauge group is $G = \mathrm{SU}(N)$, the independent chiral CB operators are $\mathrm{Tr}(\Phi^k)$, for $k = 2, \ldots, N$ and the antichiral ones are given by $\mathrm{Tr}(\bar{\Phi}^k)$. Furthermore, the non-trivial VEVs of $\Phi$ break the UV gauge group $G$ to the maximal torus subgroup $\mathrm{U}(1)^r$, where $r$ is the rank of $G$, which indicates that the low-energy effective description on this branch is a $\mathrm{U}(1)^r$ 4d $\mathcal{N} = 2$ supersymmetric gauge theory.

Note that the discussion so-far is classical. In general, one expects quantum effects to modify these solutions and the corresponding low-energy EFTs. Indeed, the classical Kähler potential which produces a flat metric on the CB receives non-trivial quantum corrections which lead to rich physics. Typically, such quantum effects in QFTs, especially from non-perturbative origins, are notoriously hard to study. However, $\mathcal{N} = 2$ supersymmetry put lots of constraints on the quantum CBs, which are required to be certain **special Kähler** manifolds. This is the main focus of Section 3.

**Definition 2.4: Higgs Branch $\mathcal{M}_{\mathrm{HB}}$**

The Higgs branch (HB) of the moduli space is where the vector multiplet scalar $\Phi$ has a zero VEV but the hypermultiplet scalars $q$ and $\tilde{q}$ no longer vanish. The non-trivial equations that must be solved are those in Eq. (2.31)

$$\mathcal{M}_{\mathrm{HB}} = \left\{ \begin{array}{c} (q_a q^{\dagger b} - \tilde{q}_a^\dagger \tilde{q}^b)(T^i)_b^a = 0 \\ (q_a \tilde{q}^b)(T^i)_b^a = 0 \end{array} \right\} \Big/ \{G - \text{gauge transformation}\}, \tag{2.34}$$

which defines the HB as a hyper-Kähler manifold.[13] Here we have explicitly indicated the gauge group indices $a, b$ and suppressed the flavor indices which are pair-wise contracted, and $T^i$ are the generators of the gauge group $G$. The complex dimensions of the HB is

$$\dim_{\mathbb{C}}(\mathcal{M}_{\mathrm{HB}}) = 2(n_H - n_V), \tag{2.35}$$

where $n_H$ and $n_V$ are respectively the number of hypermultiplets and vector multiplets that participate in the Higgsing. The gauge invariant operators whose VEVs parameterize the HB are those in Def. 2.2 which obey non-trivial chiral ring relations in general.[14] The HB operators are neutral under $\mathrm{U}(1)_r$ and charged under $\mathrm{SU}(2)_R$. Correspondingly, the HB preserves the $\mathrm{U}(1)_r$ symmetry and breaks the $\mathrm{SU}(2)_R$ symmetry (also flavor symmetries).

The low-energy effective theory on the Higgs branch is governed by an $\mathcal{N} = 2$ supersymmetric sigma model with the target space $\mathcal{M}_{\mathrm{HB}}$. In contrast to the CB, the hyper-Kähler metric on the HB does not receive quantum corrections, due to the SUSY non-renormalization theorem [117].

**Definition 2.5: Mixed Branch $\mathcal{M}_{\mathrm{MB}}$**

The last possibility is known as the mixed branch, where both vector multiplet and hypermultiplet scalars are turned on, subject to Eqs. (2.30) to (2.32). In terms of the gauge invariant operators, both types of operators in Defs. 2.1 and 2.2 have non-vanishing VEVs. Correspondingly, both $\mathrm{U}(1)_r$ and $\mathrm{SU}(2)_R$ R-symmetries are broken at a generic point on $\mathcal{M}_{\mathrm{MB}}$.

---

are still BPS, satisfying the same shortening conditions, in a non-conformal theory.

[13]For $n_H$ hypermultiplets, this defines a hyper-Kähler quotient $\mathbb{C}^{2n_H} /// G$ with the moment maps specified by the equations in the first bracket in (2.34).

[14]Once again, these operators make sense as BPS operators in a non-conformal setting.

The Mixed branch obeys a similar non-renormalization theorem as for the Higgs branch [117], which says it is locally a metric product of a special-Kähler base and a hyper-Kähler fiber. Thus, the relevant EFT is a hybrid of the CB and HB EFTs described above.

A simple example of $\mathcal{M}_{\text{MB}}$ is given by the $\mathcal{N} = 4$ SYM, which can be seen as the $\mathcal{N} = 2$ SYM coupled to a hypermultiplet in the adjoint representation of the gauge group $G$. The full moduli space of this theory is metrically $\mathbb{C}^{3r}$ and parametrized by $\Phi$, $q$ and $\widetilde{q}$ that lie in the Cartan subalgebra of $G$, and this is a mixed branch (in fact an enhanced Coulomb branch in this case [118]). We refer the readers to [63,118] for more discussions on mixed branches in $\mathcal{N} = 2$ theories.

## 2.4 Holomorphy and $\mathcal{N} = 2$ RG Flows

So far our discussion has mostly remained on the classical level. However, as alluded to above, the effective theory on the Coulomb branch typically receives quantum corrections. Hence, to fully understand the low-energy dynamics, it is imperative to have a handle on such quantum effects. To this end, some basic aspects of renormalization group (RG) flow would be useful. As we will explain, the RG running, combined with 4d $\mathcal{N} = 2$ supersymmetry, leads to incredible constraints on the quantum corrections.

The magic comes from holomorphy. As an example, the $\mathcal{N} = 1$ non-renormalization theorem [119–121], which basically states that the superpotential $W$ is not renormalized (in a particular scheme) in perturbation theory, is largely due to the holomorphy of the superpotential [1]. On the other hand, the $\mathcal{N} = 1$ Kähler potential may receive non-trivial quantum corrections in the form of the wave function renormalizations. However, in the case of $\mathcal{N} = 2$ supersymmetry there is a stronger constraint, because $\mathcal{N} = 2$ supersymmetry (or rather the $SU(2)_R$ R-symmetry) ties together the superpotential and the Kähler potential. Consequently, there is no independent quantum correction to the Kähler potential.

There is still a possible holomorphic renormalization for the holomorphic variables in the 4d $\mathcal{N} = 2$ superpotential. Let us focus on the $\mathcal{N} = 2$ SYM sector in (2.23), with the complexified gauge coupling defined in (2.24). In a general gauge theory, the gauge coupling $g$ receives quantum corrections at each loop order perturbatively, as well as non-perturbative contributions from instantons. In particular, the perturbative corrections follow the renormalization group equation

$$E\frac{dg}{dE} = -\frac{g^3}{16\pi^2}b + \mathcal{O}(g^5).\tag{2.36}$$

Here $E$ is the energy scale where $g$ is measured, and the first term on the RHS represents the 1-loop beta function and the second term comes from higher loop corrections. $b$ is known as the 1-loop beta function coefficient, which can be extracted from the field content in the relevant gauge theory [122,123], namely

$$b = \frac{11}{3}T(\text{adj}) - \frac{2}{3}T(\rho_f) - \frac{1}{3}T(\rho_s),\tag{2.37}$$

where $\rho_f$ and $\rho_s$ denote respectively the $G$ representations of the fermions and the scalars in the gauge theory. $T(\rho)$ is the quadratic Casimir invariant (Dynkin index) in the representation $\rho$ of the gauge group $G$ with the normalization such that $T(\text{adj})$ is equal to the dual Coxeter number of $G$.[15]

The non-normalization property from the 4d $\mathcal{N} = 2$ supersymmetry leads to an enormous simplification to the running of the gauge coupling. It states that the beta function is 1-loop exact, thus we can safely ignore the high-loop corrections $\mathcal{O}(g^5)$. This is a simple consequence

---

[15]For the $G = U(1)$ case, we have $T = \frac{1}{2}q^2$ where $q$ is the charge under the gauge U(1) group.

of holomorphy. Namely, the superpotential, under a certain renormalization scheme, is a holomorphic function of chiral superfields, which include the background chiral superfields whose VEV is viewed as the background complexified coupling $\tau$. Then since perturbative renormalizations cannot depend on $\theta_{\text{YM}}$ as it is associated to a topological term, such contributions at the $n$-loop order should carry a factor $\text{Im}(\tau)^{1-n}$, which is not holomorphic unless $n = 1$. Therefore, we conclude that the renormalization of the gauge coupling $g$ is 1-loop exact perturbatively[16] and correspondingly $\tau$ takes the following form

$$\tau(E) = \tau_{\text{UV}} - \frac{b}{2\pi i} \log\left(\frac{E}{\Lambda_{\text{UV}}}\right) + \dots, \tag{2.38}$$

where only non-perturbative contributions are unspecified.

An important quantity that characterizes a non-trivial RG flow is the **dynamical scale** $\Lambda$ defined as

$$\Lambda^b = E^b e^{2\pi i \tau(E)}, \tag{2.39}$$

which is perturbatively RG-invariant, i.e. $\frac{\partial \Lambda}{\partial E} = 0$. It is also known as the **transmutation scale**, namely the scale where the one-loop coupling diverges, and thus higher-loop and non-perturbative effects need to be taken into account. Equivalently, $\Lambda$ signals the transition between the strong and weak coupling phases of the gauge theory.

Using the dynamical scale $\Lambda$ which can be thought of the scalar component of a background chiral superfield, we can write down the most general expression for the coupling $\tau(E)$ consistent with the one-loop running in (2.38) and holomorphy,

$$\tau(E) = -\frac{b}{2\pi i} \log\left(\frac{E}{\Lambda}\right) + \sum_{n>0} a_n \left(\frac{\Lambda}{E}\right)^{bn}, \tag{2.40}$$

where the sum over $n > 0$ captures non-perturbative contributions from $n$-instantons as $e^{-S_{\text{inst}}} \sim (\frac{\Lambda}{E})^b$.

With $\mathcal{N} = 2$ supersymmetric field content, $b$ simplifies to the following combination

$$b = 2(T(\text{adj}) - T(\rho_{\text{matter}})), \tag{2.41}$$

where $\rho_{\text{matter}}$ denotes collectively the $G$ representation of the hypermultiplets $H_{\mathcal{N}=2}$. The coefficients $a_n$ of the instanton contributions, on the other hand, are notoriously hard to calculate for a generic theory. However, as we will see in Section 3, the Seiberg-Witten theory [20, 21] presents an elegant way to determine these non-perturbative effects with a few physical inputs.

---

**Exercise 2.3** The classical U(1)$_r$ symmetry is anomalous in general $\mathcal{N} = 2$ gauge theories. Identify the U(1)$_r$ anomaly and the non-anomalous residual symmetry.

---

**Exercise 2.4** Argue the terms multiplying $a_n$ in (2.40) are the only ones relevant for the non-perturbative corrections to the gauge coupling $\tau$.

---

## 2.5 Lagrangians for $\mathcal{N} = 2$ SCFTs

After introducing the basics of 4d $\mathcal{N} = 2$ Lagrangians and RG flows, one may wonder how they can help us to study general $\mathcal{N} = 2$ SCFTs, as we have learned that the majorities of

---

[16]The 1-loop beta-function is exact both for $\mathcal{N} = 1$ and $\mathcal{N} = 2$ SCFTs in the holomorphic scheme, but in the former case the corrections to the Kähler potential renormalize the physical coupling (i.e. in the NSVZ scheme) [124–127].

(known) 4d $\mathcal{N} = 2$ SCFT are strongly coupled and have no direct Lagrangian descriptions. Nevertheless, the 4d $\mathcal{N} = 2$ Lagrangians can be helpful in studying 4d $\mathcal{N} = 2$ SCFTs both from the ultraviolet or infrared perspective.

From the IR perspective, it is expected that certain 4d $\mathcal{N} = 2$ Lagrangians appear as effective descriptions of SCFT with $\mathcal{N} = 2$ preserving deformations. These deformations may come from turning on relevant operators, or moving onto the supersymmetric moduli space of vacua. They generally give rise to weakly coupled $\mathcal{N} = 2$ EFTs, and will be described in detail in Section 3.

From the UV perspective, there are two scenarios starting from $\mathcal{N} = 2$ Lagrangians and leading to SCFTs:

**$b = 0$**: In this case, the gauge theory Lagrangian is conformally invariant and defines an SCFT. In particular, the gauge coupling $g$ is exactly marginal and parametrize the conformal manifold of the SCFT. Some simple examples include the $\mathcal{N} = 4$ SYM and the $\mathcal{N} = 2$ conformal SQCD depicted in Figure 3a.

Since $b$ is determined by the matter context in the theory of interest, we can classify conformal Lagrangians by adjusting the matter content in such a way that $b = 0$. In particular, focusing on the case of special unitary gauge groups $\otimes_i \mathrm{SU}(N_i)$, and assuming the matter content is given by bifundamental hypermultiplets, there is a complete classification of conformal quiver Lagrangians that coincides with the classification of Dynkin and affine-Dynkin diagrams. For instance, the $E_6$ affine Dynkin diagram and the corresponding conformal quiver Lagrangian are shown in Figure 3b, where each circle node hosts a gauge group and each edge lives a bifundamental hypermultiplet. The case with the most general gauge groups and matter content was classified in [128].

An obvious merit of the conformal Lagrangians is that they readily enable an array of concrete computations of SCFT observables, for example the supersymmetric sphere partition functions using localization which have non-trivial dependence on the complexified gauge couplings [76–78, 129].

**$b > 0$**: Another context where $\mathcal{N} = 2$ Lagrangians lead to interesting SCFTs in the IR is when the $\mathcal{N} = 2$ theories are asymptotically free. In these cases, the superconformal symmetry is not manifest but only emergent in the IR. The way in which such emergent SCFTs have been found is by tuning some $\mathcal{N} = 2$ preserving parameters in the Lagrangian, such as potential mass parameters, the VEV of some protected operators, or even the dynamical scale $\Lambda$ itself. As an example, depicted in Figure 3c, SU(2) SYM theory coupled with one fundamental hypermultiplet and the pure SU(3) SYM theory have, on the CB, a superconformal fixed point described by the $(A_1, A_2)$ Argyres-Douglas (AD) theory [22].[17] To relate such asymptotic free theories to emergent IR SCFTs may seem like a guessing game, but there are systematic tools using the IR EFT of an asymptotic Lagrangian. The strategy is to look for patches on the CB (that corresponds to tuning $\mathcal{N} = 2$ preserving parameters) with emergent scale invariance. This is how the original AD theory was discovered in [22, 23].

It is possible, but much harder, to extract SCFT observables from these asymptotic free UV Lagrangians. In addition to tuning the parameters required for identifying the SCFT point in the IR, one must be careful in dealing with decoupled sectors that arise from the flow. In specific theories, some progress has been made, see for examples [130–132].

Recently it was discovered that the emergence of 4d $\mathcal{N} = 2$ SCFTs is not limited to $\mathcal{N} = 2$ supersymmetric RG flows, but can also come from certain flows preserving only $\mathcal{N} = 1$ supersymmetry at the intermediate scale and experiencing non-trivial SUSY enhancements in the

---

[17]We will discuss how to realize these theories from a top-down approach either in Class $\mathcal{S}$ in Section 4 or in type IIB in Section 5

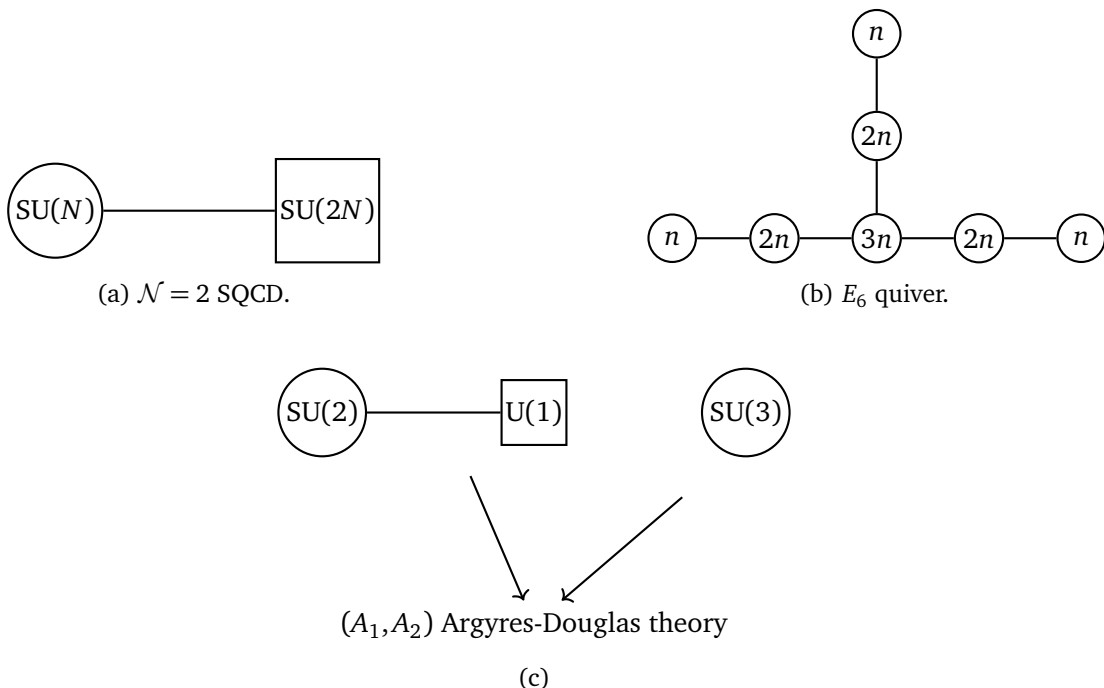

(a) $\mathcal{N}=2$ SQCD.

(b) $E_6$ quiver.

$(A_1, A_2)$ Argyres-Douglas theory

(c)

Figure 3: Examples of manifest (top two) or emergent (bottom one) SCFTs.

IR [133, 134]. The $\mathcal{N}=1$ gauge theories in this scenario are typically related to nilpotent mass deformations of some $\mathcal{N}=2$ SCFT.

## 2.6 Anomalies of $\mathcal{N}=2$ SCFTs

We recall that a quantum field theory with a classical symmetry $G$ (e.g. symmetry of the Lagrangian), whether it is gauge or global, can possibly develop an anomaly due to quantum effects which can be detected by non-invariance of the path integral measure. We refer to the reviews [135, 136] for more details of the basics of anomaly in QFTs.

Whereas a gauge anomaly indicates that the theory is not consistent, an anomaly associated with global symmetry does not invalidate the theory, but instead, it is a powerful tool to extract robust information about the theory, especially when it is strongly coupled and without Lagrangian descriptions like many 4d $\mathcal{N}=2$ SCFTs. We refer to such an anomaly as a 't Hooft anomaly: it does not change along the RG flow as argued by 't Hooft [137], and hence can be reliably calculated in the UV when the UV theory admits certain weakly coupled descriptions. The notion of symmetries and 't Hooft anomalies has been generalized and reformulated in recent years, which is an active area of research. In what follows, we will mainly consider perturbative anomalies for continuous 0-form global symmetry (and spacetime symmetries), and comment on non-perturbative anomalies for discrete and generalized global symmetries in 4d $\mathcal{N}=2$ SCFTs in the conclusion.

The possible perturbative anomalies for Lorentz and global symmetries in a general $d$-dimensional QFT is classified by a degree $d+2$ anomaly polynomial $\mathcal{I}_{d+2}$ which is a polynomial in the characteristic classes for the background gauge fields coupling to the symmetries. For $d=4$, the possible anomalies are the mixed gravitational-Abelian anomaly, the pure non-Abelian anomaly, the mixed Abelian-non-Abelian anomaly and the pure Abelian anomaly corresponding to the four terms in $\mathcal{I}_6$ below,

$$\mathcal{I}_6 = \alpha c_1(F)p_1(T) + \beta c_3(F) + \gamma c_2(F)c_1(F) + \delta c_1(F)^3, \qquad (2.42)$$

where $c_i(F)$ denotes the $i$-th Chern class for the background gauge field with curvature $F$ and $p_1(T)$ is the Pontryagin class for the tangent bundle to the spacetime manifold.

Any 4d SCFT has a distinguished global symmetry given by the U(1)$_r$ R-symmetry and the relevant (mixed) anomalies are governed by the two coefficients $k_{rrr}$ and $k_r$ below,

$$\mathcal{I}_6 \supset -\frac{k_r}{24}c_1(F)p_1(T) + \frac{k_{rrr}}{6}c_1(F)^3, \tag{2.43}$$

where $F$ is restricted to the U(1)$_r$ background above. These anomaly coefficients are determined by the parity-odd structure of the three-point-functions involving the U(1)$_r$ current and the stress-energy tensor as Figure 4.[18] Here we list the contributions of 4d $\mathcal{N} = 1$ vector

$$k_{rrr} = \left( \begin{array}{c} J_r \\[-10pt] \\ J_r \end{array} \right)_{\text{odd}} \qquad k_r = \left( \begin{array}{c} T \\[-10pt] \\ T \end{array} \right)_{\text{odd}}$$

Figure 4: The anomaly coefficients $k_{rrr}$ and $k_r$ from three-point-functions of the U(1)$_r$ current $J_r$ and stress-energy tensor $T$. In Lagrangian theories, they are determined by the massless Weyl fermions running in the 1-loop triangles.

multiplet $V$ and chiral multiplet $Q$ to $(k_{rrr}, k_r)$:

$$\begin{array}{llll} V: & k_{rrr} := \operatorname{Tr} r_{\mathcal{N}=1}^3 = \operatorname{rank} G, & k_r := \operatorname{Tr} r_{\mathcal{N}=1} = \operatorname{rank} G, \\ Q: & k_{rrr} := \operatorname{Tr} r_{\mathcal{N}=1}^3 = (r(q)-1)^3 \dim(Q), & k_r := \operatorname{Tr} r_{\mathcal{N}=1} = (r(q)-1)\dim(Q), \end{array} \tag{2.44}$$

where the trace Tr is over all Weyl fermions, and $r(q)$ denotes the U(1)$_r$ R-charge of the scalar component of the 4d $\mathcal{N} = 1$ chiral multiplet $Q$.

In addition to the 't Hooft anomalies, conformal field theories in even dimensions also have Weyl anomalies, which modify the traceless condition for the stress tensor by curvature invariants of the spacetime manifold and thus are also known as trace anomalies. In 4d the trace anomaly takes the following form[19]

$$\left\langle T_\mu^\mu \right\rangle = \frac{c}{16\pi^2}(\text{Weyl})^2 - \frac{a}{16\pi^2}(\text{Euler}), \tag{2.45}$$

where

$$(\text{Weyl})^2 = R_{\mu\nu\rho\sigma}^2 - 2R_{\mu\nu}^2 + \frac{1}{3}R^2, \ (\text{Euler}) = R_{\mu\nu\rho\sigma}^2 - 4R_{\mu\nu}^2 + R^2. \tag{2.46}$$

The constant coefficients $(a, c)$ are known as the conformal central charges for 4d CFTs, which are determined by $\mathcal{N} = 1$ supersymmetry in terms of the above U(1)$_r$ anomalies as [138, 139]

$$a = \frac{3}{32}\left[3k_{rrr} - k_r\right], \quad c = \frac{1}{32}\left[9k_{rrr} - 5k_r\right]. \tag{2.47}$$

---

[18]As an aside, the U(1)$_r$ R-symmetry in a general non-conformal 4d $\mathcal{N} = 2$ gauge theory has an Adler-Bell-Jackiw (ABJ) anomaly due to the **dynamical** gauge field in the theory, whose anomaly coefficient is $2b$ where $b$ is the one loop beta function coefficient defined in (2.41). Consequently, for nonzero $b$, this R-symmetry is explicitly broken to $\mathbb{Z}_{2b}$. However, if such a gauge theory is conformal, i.e., $b = 0$, the U(1)$_r$ is free of the ABJ anomaly and a bona fide symmetry, as expected for general 4d $\mathcal{N} = 2$ SCFTs.

[19]The trace anomalies of general even-dimensional CFTs, take a similar form. In particular, the $a$-anomaly is universal to all dimensions whereas the Weyl part depends on the specific dimension. For example, in $d = 2$, there is no Weyl part whereas in $d = 6$ the Weyl part has three independent terms.

Such a non-trivial connection exists because the $\mathcal{N} = 1$ supersymmetry relates the R-symmetry current and the stress tensor.

Our interest here is on 4d $\mathcal{N} = 2$ SCFTs whose R-symmetry $U(1)_{r_{\mathcal{N}=2}} \times SU(2)_R$ have (mixed) anomalies. As before, by $\mathcal{N} = 2$ supersymmetry, they are determined in terms of the Weyl anomalies $a$ and $c$ [75],

$$\mathcal{I}_6 \supset (c-a)c_1(F_{U(1)_{r_{\mathcal{N}=2}}})p_1(T) + (a-c)c_1(F_{U(1)_{r_{\mathcal{N}=2}}}))^3 + 2(2a-c)c_1(F_{U(1)_{r_{\mathcal{N}=2}}})c_2(F_{SU(2)_R}). \tag{2.48}$$

One simple check of the above is to compare with (2.43) and (2.47), noting the relation between the $\mathcal{N} = 1$ and $\mathcal{N} = 2$ $U(1)_r$ symmetries in (2.18). We collect the anomalies for $\mathcal{N} = 2$ free fields in Table 2.

Table 2: The anomalies of 4d $\mathcal{N} = 2$ free fields.

| | $a$ | $c$ | $\mathcal{I}_6$ |
|---|---|---|---|
| Vector multiplet | $\frac{5}{24}$ | $\frac{1}{6}$ | $-\frac{1}{24}c_1(F_{U(1)_{r_{\mathcal{N}=2}}})p_1(T) + \frac{1}{24}c_1(F_{U(1)_{r_{\mathcal{N}=2}}})^3 + \frac{1}{2}c_1(F_{U(1)_{r_{\mathcal{N}=2}}})c_2(F_{SU(2)_R})$ |
| Hypermultiplet | $\frac{1}{24}$ | $\frac{1}{12}$ | $\frac{1}{24}c_1(F_{U(1)_{r_{\mathcal{N}=2}}})p_1(T) - \frac{1}{24}c_1(F_{U(1)_{r_{\mathcal{N}=2}}})^3$ |

When the SCFT has a conformal Lagrangian description, i.e., $b = 0$ case in 2.5, the 't Hooft anomalies can be read-off from Table 2 and the conformal central charges are simply given by

$$a = \frac{5n_v + n_h}{24}, \, c = \frac{2n_v + n_h}{12}, \tag{2.49}$$

where $n_v$ and $n_h$ are the total numbers of $\mathcal{N} = 2$ vector multiplets and hypermultiplets in the theory.

For non-Lagrangian SCFTs, it is generally much harder to determine their anomalies. Fortunately, we have the powerful tool of anomaly matching, which exploits the invariance property of anomalies under deformations. If the theory admits a symmetric deformation to a weakly coupled description, the anomalies can be recovered from there. In the context of $\mathcal{N} = 2$ SCFTs, the useful deformations involve going onto the vacuum moduli space of the theory. For example, on the CB of the SCFT, the $U(1)_r$ symmetry and conformal symmetry are spontaneously broken, whereas the $SU(2)_R$ is preserved. As usual, this means the mixed anomalies involving $U(1)_r$ and $SU(2)_R$ will receive contributions from both the field content in the Coulomb branch EFT and Wess-Zumino terms that involve the Goldstone boson. Using the relation between $(a, c)$ and the 't Hooft anomalies given by (2.48), this leads to the following formulas for the conformal central charges at the fixed point [75],

$$a = \frac{5r}{24} + \frac{h}{24} + \frac{R(A)}{4} + \frac{R(B)}{6}, \quad c = \frac{r}{6} + \frac{h}{12} + \frac{R(B)}{3}, \tag{2.50}$$

where $r$ and $h$ count the number of free vector multiplets and free hypermultiplets at a generic point of the Coulomb branch of the 4d SCFT respectively, and $R(A)$ and $R(B)$ capture contributions from the Wess-Zumino terms [140–144]. $R(A)$ and $R(B)$ are accessible from the CB EFT coupled to general background fields and it turns out to they have simple expressions in terms of basic CB data such as the spectrum of scaling dimensions for the CB chiral primaries [75] (see also Eq. (1.1a)-(1.1c) of [145]). We will see explicitly how to extract $R(A)$ and $R(B)$ for a large zoo of SCFTs in Section 5.1.

Finally, if the $\mathcal{N} = 2$ SCFT has a global symmetry $G_{flavor}$ (which we assume to be simple for simplicity), it is possible to introduce another central charge $k_{flavor}$ that captures the mixed $U(1)_r$-$G_{flavor}$ anomaly,

$$\mathcal{I}_6 \supset -\frac{1}{2}k_{flavor}c_1(F_r)c_2(F_{flavor}). \tag{2.51}$$

As for the other perturbative anomalies, it is determined by parity-odd structure in the three-point-function of the $U(1)_r$ and $G_{\text{flavor}}$ currents. With $\mathcal{N} = 2$ supersymmetry, this anomaly is further related to the OPE of two $G_{\text{flavor}}$-currents [75, 146],

$$\langle J_\mu^a(x) J_\nu^b(0) \rangle = \frac{3 k_{\text{flavor}}}{4\pi^4} \delta^{ab} \frac{x^2 g_{\mu\nu} - 2 x_\mu x_\nu}{x^8} + \frac{2}{\pi^2} f^{abc} \frac{x_\mu x_\nu x \cdot J^c(0)}{x^6} + \dots . \tag{2.52}$$

Note that here the currents $J_\mu^a$ are normalized by the second term on the RHS and the structure constants $f^{abc}$ satisfy

$$f^{aed} f^{bde} = 2 h^\vee \delta^{ab} , \tag{2.53}$$

where $h^\vee$ is the dual Coxeter number for $G_{\text{flavor}}$. In this convention, $n_H$ free hypermultiplets have a $G_{\text{flavor}} = \text{USp}(2n_H)$ flavor symmetry with anomaly coefficient $k_{\text{flavor}} = 1$. The computation of the flavor central charge $k_{\text{flavor}}$ in non-trivial SCFTs can be found for instance in [146, 147].

# 3 Coulomb Branch Effective Theory and Argyres-Douglas Points

In this section, we discuss the Coulomb branch (CB) effective field theory (EFT) that governs the low-energy dynamics on the CB moduli space of a general 4d $\mathcal{N} = 2$ theory. We will show that such an EFT provides an indispensable tool to study the 4d $\mathcal{N} = 2$ SCFT that lives at the origin of the CB. For completeness, it is useful to start by reviewing the seminal work of Seiberg-Witten (SW) [20, 21] and give an introduction to the SW geometry. As we will show, many properties and dynamics of the CB EFT of a 4d $\mathcal{N} = 2$ theory can be determined from an auxiliary geometric object, known as the **Seiberg-Witten curve**.

The basic degrees of freedom on the CB, of complex dimension $\dim_\mathbb{C} \mathcal{M}_{\text{CB}} = r$, are $r$ $\mathcal{N} = 2$ Abelian vector multiplets including the complex scalar fields that parametrize $\mathcal{M}_{\text{CB}}$, and $U(1)^r$ gauge fields governed by a Maxwell action, as well as their fermionic partners. If the theory has a gauge theory description in the UV, as discussed in Section 2.3, then these Abelian vector multiplets correspond to the Cartan generators of the gauge group $G$ of rank $r$, which are preserved along the CB of vacuum moduli space where the hypermultiplet scalars vanish $\langle q \rangle = \langle \tilde{q} \rangle = 0$ while the non-Abelian vector multiplet scalar develops a VEV $\langle \Phi \rangle$. For $G = \text{SU}(r+1)$, up to a gauge transformation, the VEV takes the form

$$\langle \Phi \rangle = \text{diag}\left( a_1, a_2, \dots, a_{r+1} = -\sum_i a_i \right) . \tag{3.1}$$

The nonzero $a_i$ give rise to massive W-bosons as usual with $m_W \sim |a_i - a_j|$, massive hypermultiplets through the superpotential $W = \tilde{Q}\Phi Q$ with $m_H \sim |a_i|$, and less obviously massive monopoles with $m_M \sim \frac{1}{g_i^2}|a_i|$ where $g_i$ denotes the Maxwell couplings. We are interested in the Wilsonian EFT for the $U(1)^r$ massless vector multiplets that are obtained from integrating out all these massive fields.[20] In the following, slightly abusing the notation, we will denote the Abelian vector multiplets by $a_i$ with $i = 1, 2, \dots, r$ and the bottom scalar components also by $a_i$, when there is no room for confusion from the context.

The EFT is governed by a local Lagrangian $\mathcal{L}_{\text{EFT}}$ at a generic point on $\mathcal{M}_{\text{CB}}$. Despite the triviality of conventional pure Abelian gauge theories described by the Maxwell action with constant couplings, this CB EFT has rich dynamics due to the extra scalar fields over which

---

[20]We emphasize that this Wilsonian EFT is well-defined regardless of the existence of a Lagrangian UV description. However, a first-time reader may find it useful to think about the Lagrangian examples.

the Maxwell couplings vary in a non-trivial fashion. In general, the Wilsonian effective action is a very complicated object. However, $\mathcal{N} = 2$ supersymmetry provides stringent constraints on the effective Lagrangian $\mathcal{L}_{\text{EFT}}$ living on the CB,[21] and it takes the following form in $\mathcal{N} = 1$ superspace [1, 121, 148]

$$\mathcal{L}_{\text{EFT}} \supset \frac{1}{4\pi} \text{Im} \left( \int d^4\theta \frac{\partial \mathcal{F}}{\partial \Phi_i} \bar{\Phi}_i + \int d^2\theta \frac{1}{2} \frac{\partial^2 \mathcal{F}}{\partial \Phi_i \partial \Phi_j} W_\alpha^i W^{j\alpha} \right), \quad i, j = 1, ..., r, \quad (3.2)$$

where $\Phi_i$ and $W_\alpha^i$ respectively denote the chiral and gaugino superfields in the $\mathcal{N} = 2$ U(1)$^r$ vector multiplets $a_i$. $\mathcal{F}$ is a holomorphic function (locally), known as the prepotential, which determines the whole effective action (up to second derivatives) as a consequence of supersymmetry. Such constraint intimately relates the special Kähler geometries to the dynamics of 4d $\mathcal{N} = 2$ CB theories. To see that, recall that the first term on the RHS of (3.2) is known as the Kähler potential $K(\Phi_i)$ and with the VEVs (3.1) now promoted to slowly varying moduli fields, the scalar part reduces to

$$\text{Im} \left( \frac{\partial^2 \mathcal{F}}{\partial a_i \partial a_j} da^i d\bar{a}^j \right), \quad (3.3)$$

where $\bar{a}$ denotes the conjugate of $a$. In terms of a non-linear sigma model, it defines a metric

$$g_{i\bar{j}} = \partial_i \partial_{\bar{j}} K = \text{Im} \left( \frac{\partial^2 \mathcal{F}}{\partial a_i \partial a_j} \right), \quad (3.4)$$

on the space of inequivalent vacuum configurations, i.e., the moduli space $\mathcal{M}_{\text{CB}}$, which is further an $r$ complex dimensional rigid special Kähler manifold, as the metric is determined by the prepotential $\mathcal{F}$ [149]. Unitarity requires the scalar kinetic term to be positive, thus the prepotential $\mathcal{F}$ is constrained such that the sigma model metric (3.4) is positive-definite.

The second term on the RHS of (3.2) contains the Maxwell Lagrangian for the U(1)$^r$ gauge fields,

$$\mathcal{L}_{EM} = \frac{1}{16\pi} \left[ \text{Im} \left( \frac{\partial^2 \mathcal{F}}{\partial a_i \partial a_j} \right) F^i \wedge \star F^j + \text{Re} \left( \frac{\partial^2 \mathcal{F}}{\partial a_i \partial a_j} \right) F^i \wedge F^j \right]. \quad (3.5)$$

Consequently the prepotential $\mathcal{F}$ also determines the field-dependent complexified gauge coupling (matrix) $\tau^{\text{IR}} = \frac{\theta}{2\pi} + \frac{4\pi i}{g^2}$ by

$$\tau_{ij}^{\text{IR}} = \frac{\partial^2 \mathcal{F}}{\partial a_i \partial a_j}, \quad (3.6)$$

whose imaginary part coincides with the sigma model metric (3.4) and its positivity ensures that the gauge kinetic terms are well-defined.

One remark is that we have seen two sets of coordinates for the CB moduli space $\mathcal{M}_{\text{CB}}$ so far, given by the vector multiplet scalars $a_i$ discussed above and the Coulomb branch operators

$$u_i := \text{Tr}(\Phi^{i+1}), \quad i = 1, ..., r, \quad (3.7)$$

as indicated in Def. 2.3. The coordinates $\{u_i\}$ defined in terms of gauge invariant operators are physical and unambiguous. In contrast, the coordinates $\{a_i\}$ are not gauge invariant (they transform under the Weyl group $S_N$ of SU($N$)) and contain a further ambiguity due to the electromagnetic duality on the CB which we will come to shortly. Consequently, the coordinates $\{a_i\}$ are only defined locally on the CB and are subject to identifications by the gauge and duality transformations from patch to patch. Nonetheless, the local coordinates $\{a_i\}$ are

---

[21]In the cases with $\mathcal{N} = 2$ supersymmetry, the Wilsonian action on the CB coincides with the 1PI effective action at the two-derivative level. See more discussions in [84] and references therein.

what make possible the EFT description on the $\mathcal{N} = 2$ CB and encode an elegant emergent geometry, with the redundancies in the coordinates leading to constraints on it.

In the UV, the prepotential $\mathcal{F}_{\text{UV}}$ can be read off from the classical Lagrangian term (2.23) of an $\mathcal{N} = 2$ super Yang-Mills theory, which has the form

$$\mathcal{F}_{\text{UV}} \sim \tau_{\text{UV}} \text{Tr} \left( \mathbf{V}^2_{\mathcal{N}=2} \right), \tag{3.8}$$

where $\mathbf{V}_{\mathcal{N}=2}$ denotes the $\mathcal{N} = 2$ vector multiplet in the $\mathcal{N} = 2$ superspace formalism[22] which packages the $\mathcal{N} = 1$ chiral multiplets $Q$ and $\mathcal{N} = 1$ gaugino multiplet $W_\alpha$ in a compact form as $\mathbf{V}_{\mathcal{N}=2} := Q + W_\alpha \widetilde{\theta}^\alpha$. In the IR, the prepotential $\mathcal{F}$ would be more complicated, and it is a holomorphic function of the vector multiplet scalars $a_i$ with various corrections that could depend on the dynamical scale $\Lambda$ and mass parameters,

$$\mathcal{F}(a_i) \sim \tau^{\text{IR}}_{ij}(a) a^i a^j + \dots, \tag{3.10}$$

where $\tau^{\text{IR}}_{ij}(a)$ has an expansion of the form (2.40) when the energy scale $E$ is set to $a$. The main goal of the SW theory is to determine the IR prepotential from the corresponding one in the UV, and as Seiberg and Witten pointed out [20,21], with some physical input, it can be completely determined by the associated SW geometry.[23]

The rest of this section is organized as follows. In Section 3.1 we review electric-magnetic duality in 4d $\mathcal{N} = 2$ theories. In Section 3.2 we introduce the prepotential for 4d $\mathcal{N} = 2$ SQFTs and its role in describing the special geometry on the CB. In Section 3.3 we explain how to obtain the quantum prepotential from the SW curve. We introduce Argyres-Douglas (AD) theories and their scale-invariant SW solutions in Section 3.4. Finally, in Section 3.5 we describe the CFT data that can be extracted from the CB EFT near an AD point.

## 3.1 Electric-Magnetic Duality

We have seen that the EFT on the CB is governed locally by the Lagrangian (3.2) for the $U(1)^r$ vector multiplets. It turns out that this Lagrangian description is not unique. Instead there are multiple (but equivalent) Lagrangians with different sets of fundamental variables that are related to one another by a supersymmetric version of the electric-magnetic duality. Around a generic point on $\mathcal{M}_{\text{CB}}$, they are all equally good local descriptions of the low energy physics. Globally the CB EFT is built from such local patches by gluing maps involving non-trivial duality transformations that lead to monodromies. Such duality monodromies necessitates singularities on the CB, which is the reason behind the interesting CB dynamics. In this subsection, we are going to lay down some fundamental aspects of the electric-magnetic duality.

Let us start by recalling how electric-magnetic duality works in the pure Maxwell theory in 4 dimensions. Consider the partition function for Maxwell theory in the Euclidean signature

$$\int \left[ \mathcal{D}A_\mu \right] \exp \int d^4x \left( -\frac{1}{4g^2} F_{\mu\nu} F^{\mu\nu} + \frac{i\theta}{32\pi^2} \epsilon_{\mu\nu\rho\sigma} F^{\mu\nu} F^{\rho\sigma} \right). \tag{3.11}$$

The duality transformation is implemented by changing the integration variable from the gauge field $A$ to the field strength $F$. However, at this point we are actually losing information:

---

[22]Here we introduce another set of supercoordinates $\widetilde{\theta}$ for the $\mathcal{N} = 2$ superspace and the classic Lagrangian (2.23) can be simply put as

$$\mathcal{L}_{\text{EFT}} = \int d^2\theta d^2\widetilde{\theta} \mathcal{F}(a_i) + \text{h.c.} \tag{3.9}$$

More details of this formalism can be seen in, e.g., [150].

[23]On the other hand, given a general SW geometry (which encodes the low energy EFT of a putative SCFT), it can be difficult to extract information of the corresponding SCFT. See [151] for a recent attempt using the mixed Hodge structure of the fiber of the SW geometry.

Maxwell theory is not just a theory of 2-forms, but rather a theory of locally exact 2-forms. This is usually encoded in the Bianchi identity $dF = 0$. In order to recover this information in the new dual formulation where we are path integrating over the space of field strengths, we introduce a new Lagrange multiplier 1-form $A^D$ and write

$$
\int \left[ \mathcal{D}F_{\mu\nu} \right] \left[ \mathcal{D}A^D_\lambda \right] \exp \int d^4x \left( -\frac{1}{4g^2} F_{\mu\nu} F^{\mu\nu} + \frac{i\theta}{32\pi^2} \epsilon^{\mu\nu\rho\sigma} F_{\mu\nu} F_{\rho\sigma} + \frac{i}{8\pi} \epsilon^{\mu\nu\rho\sigma} \partial_\mu A^D_\nu F_{\rho\sigma} \right) .
\tag{3.12}
$$

To pass to the dual description, we now have to perform the integral over $F$. Since this is a Gaussian theory, we can do this at the classical level, i.e., eliminate $F$ via its equations of motion. This manipulation is more transparent if we first rewrite the Lagrangian in terms of the self-dual and anti-self-dual field strengths $F^\pm_{\mu\nu} := \frac{1}{2} \left( F_{\mu\nu} \pm \frac{1}{2} \epsilon_{\mu\nu\rho\sigma} F^{\rho\sigma} \right)$ (and similarly for the field strength of $A^D$) as

$$
\frac{i}{8\pi} \int d^4x \left( \bar{\tau}(F^+)^2 - \tau(F^-)^2 \right) + \frac{i}{4\pi} \int d^4x \left( F^+_D F^+ - F^-_D F^- \right) .
\tag{3.13}
$$

The reader is encouraged to confirm that upon replacing $F^\pm$ by their equations of motion, one arrives at the following path integral,

$$
\int \left[ \mathcal{D}A^D_\mu \right] \exp \left( -\frac{i}{8\pi} \int d^4x \left[ \frac{-1}{\bar{\tau}} \left( F^+_D \right)^2 - \frac{-1}{\tau} \left( F^-_D \right)^2 \right] \right) ,
\tag{3.14}
$$

which indeed describes the same Maxwell theory but with a dual magnetic variable $A_D$ and dual coupling $\tau_D := -\frac{1}{\tau}$.

Comparing (3.14) with (3.11), we conclude the Maxwell theory is invariant[24] under the following transformation:

$$
A \to A_D \,, \; \tau \to -\frac{1}{\tau} \,.
\tag{3.15}
$$

Furthermore, by noting that the Maxwell theory is also invariant under the shift $\theta \to \theta + 2\pi n$ with $n \in \mathbb{Z}$, the whole duality group is enhanced to $\mathrm{SL}(2,\mathbb{Z})$.

> **Exercise 3.1** Substitute the EOMs for $F^\pm$ inside Eq. (3.13) and show that the resulting theory is Eq. (3.14).

So far we have focused on the EM duality in the pure Maxwell theory. With $\mathcal{N} = 2$ supersymmetry, one naturally expects that the scalar field $a_i$, which sits in the same supermultiplet as the gauge field $A_i$, should also enjoy the same duality property. To this end, let us come back to the effective theories on the CB. The crucial insight is that $\mathrm{Im}\left( \tau^{\mathrm{IR}}_{ij} \right)$ as defined in (3.4) and (3.6) is both harmonic and positive-definite, and thus cannot be globally defined over the entire moduli space $\mathcal{M}_{\mathrm{CB}}$ unless it is a constant, in which case the theory is free. Nonetheless, $\mathrm{Im}\left( \tau^{\mathrm{IR}}_{ij} \right)$ is locally well-defined, for example in a semi-classical (weak-coupling) region on the CB. The obstruction to extending this to the entire CB is caused by genuine quantum effects, which modify the structure of the classical moduli space, which a priori can be globally described by $a_i$. For simplicity, we are going to focus on discussing rank-1 cases in the rest of the section, so the subscript $_{ij}$ can be dropped out.

If we define another coordinate

$$
a^D = \frac{\partial \mathcal{F}}{\partial a} \,,
\tag{3.16}
$$

---

[24]As an aside, we would like to stress that if the 4d space-time is curved, then the Maxwell theory could possibly have certain anomaly [152, 153] under a finite subgroup of $\mathrm{SL}(2,\mathbb{Z})$ which becomes a symmetry of the theory at special values of $\tau$, dubbed duality anomaly in [154, 155].

then the metric on the CB can be written as

$$ds^2 = \operatorname{Im} da_D \, d\overline{a} = -\frac{i}{2}(da_D \, d\overline{a} - da \, d\overline{a}_D). \tag{3.17}$$

Here $a$ and $a_D$ are (multivalued) holomorphic functions of $u \equiv \operatorname{Tr}\left(\Phi^2\right)$ on $\mathcal{M}_{\mathrm{CB}}$. The $\mathcal{N} = 2$ supersymmetric extension of the SL$(2,\mathbb{Z})$ duality in the pure Maxwell theory acts on $(a, a_D)$ as

$$\begin{pmatrix} a^D \\ a \end{pmatrix} \rightarrow \begin{pmatrix} f & g \\ h & l \end{pmatrix} \begin{pmatrix} a^D \\ a \end{pmatrix}, \tag{3.18}$$

with $f, g, h, l \in \mathbb{Z}$ and the determinant of the matrix equal to one. The above transformation clearly leaves the CB metric (3.17) invariant[25] and furthermore it acts on the effective coupling $\tau^{\mathrm{IR}}$,

$$\tau^{\mathrm{IR}} = \frac{\partial a^D}{\partial a}, \tag{3.19}$$

as

$$\tau^{\mathrm{IR}} \rightarrow \frac{f \tau^{\mathrm{IR}} + g}{h \tau^{\mathrm{IR}} + l}, \tag{3.20}$$

which is precisely how SL$(2,\mathbb{Z})$ duality acts on the Maxwell coupling.

The particle states also transform under the above duality, such as electrons, monopoles and dyons. Among these objects, we can identify the BPS particles [156, 157] that saturate certain BPS conditions [158]. Indeed, recalling the definition of central charges $Z$ in Eq. (2.12), for any $\mathcal{N} = 2$ particle with mass $M$, it is a consequence of the supersymmetry algebra that

$$M \geq |Z|, \tag{3.21}$$

and the saturation of the inequality is required for BPS particles. Note that such BPS saturated states are protected by the $\mathcal{N} = 2$ supersymmetry from wandering off the bound, due to either perturbative or non-perturbative corrections.[26] The central charges in 4d $\mathcal{N} = 2$ theories are determined in a semi-classical (weakly-coupled) regime by

$$Z = na + ma^D + \sum_A f_A \mu_A, \tag{3.22}$$

where now $a$ and $a^D$ are the bottom components of the corresponding (dual) $\mathcal{N} = 2$ vector multiplets, and $\mu_A$ denotes background mass parameters for flavor symmetries. Finally, $n$, $m$ and $f_A$ are respectively electric, magnetic and flavor charges for a BPS particle.

Mathematically, in different patches of $\mathcal{M}_{\mathrm{CB}}$, the doublet

$$\left(a, a^D\right) \tag{3.23}$$

defines **special coordinates** which, as we will show later, are tied with the special geometry on $\mathcal{M}_{\mathrm{CB}}$. We can treat the two coordinates on the equal footing. Namely, as the coordinate $a$ denotes the scalar component in an $\mathcal{N} = 2$ vector multiplet, we can view the other coordinate $a^D$ as a magnetic dual of $a$, so that $a^D$ belongs to an $\mathcal{N} = 2$ vector multiplet containing the dual gauge field $A^D$. The dual description, in terms of $a^D$, can be determined from $a$ and

---

[25]The special Kähler metric on the CB is in fact invariant under the larger SL$(2,\mathbb{R})$ group (and Sp$(2r,\mathbb{R})$ for the higher rank case) which is reduced to SL$(2,\mathbb{Z})$ (and Sp$(2r,\mathbb{Z})$ in general) due to the quantization of the electro-magnetic charges.

[26]More precisely, the BPS particles are stable (due to the BPS bound) at a generic point on the CB. At certain real codimension one loci, there are walls of marginal stability where the BPS particles may decay, leading to non-trivial jumps in the BPS spectrum that are known as the wall-crossing phenomena. See [20, 21] for a description of such phenomena in SU(2) gauge theories.

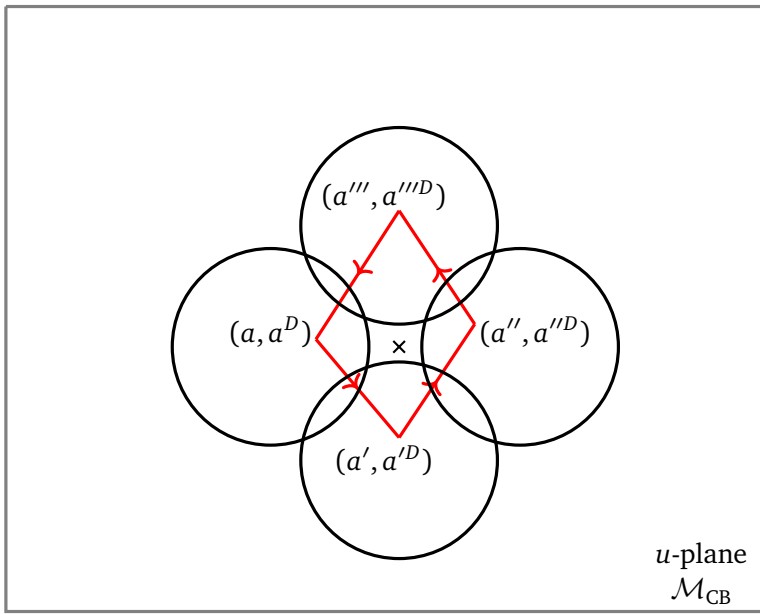

Figure 5: Different choice of special coordinates $(a, a_D)$ on the CB and the duality monodromy around a singularity.

the prepotential $\mathcal{F}(a)$ using (3.16), so both descriptions carry the same information which is captured by $\mathcal{F}$. Going from one patch $\mathcal{U}$ with the doublet $(a(u), a^D(u))$ to another patch $\mathcal{U}'$ with another doublet $(a'(u), a'^D(u))$ involves an SL(2, $\mathbb{Z}$) duality transformation. Indeed, the doublet $(a, a^D)$ can be viewed as the holomorphic section of an SL(2, $\mathbb{Z}$) bundle over the moduli space $\mathcal{M}_{\text{CB}}$ away from the singularities and undergoes nontrivial $SL(2, \mathbb{Z})$ monodromies around the singularities on $\mathcal{M}_{\text{CB}}$ (see Section 3.2 for more details). This is essentially the gist of how the EM duality is encoded on the CB. A schematic picture to keep in mind is depicted in Figure 5.

The similar analysis can be generalized to a higher rank-$r$ theory, where $\mathcal{M}_{\text{CB}}$ is now an $r$ complex dimensional space. The duality group becomes Sp(2$r$, $\mathbb{Z}$) and each of the special coordinates $\mathbf{a}$ and $\mathbf{a_D}$ now can be viewed as an $r$-dimensional vector which transforms as

$$\begin{pmatrix} \mathbf{a^D} \\ \mathbf{a} \end{pmatrix} \rightarrow \begin{pmatrix} A & B \\ C & D \end{pmatrix} \begin{pmatrix} \mathbf{a^D} \\ \mathbf{a} \end{pmatrix}, \tag{3.24}$$

where $A, B, C, D$ are $r$-by-$r$ matrices, and together they parametrize the Sp(2$r$, $\mathbb{Z}$) group.

## 3.2 Prepotential and Special Geometry on Coulomb Branch

As explained in the previous section, the CB EFT is built from local descriptions on patches of the CB that are glued together by EM duality transformations and the low energy physics is completely encoded in the holomorphic prepotential $\mathcal{F}(a)$. In this section, we give a brief introduction to the general strategy to solve for $\mathcal{F}(a)$. The determination of the prepotential allows for finding the special coordinates in Eq. (3.23) which describe the Coulomb branch of the effective field theory.

First, there are several general constraints on the prepotential:

- It must be locally a holomorphic function of $a$,

- It must respect the symmetries present in the UV theory, e.g., U(1)$_r$ R-symmetry,

- Finally, in the large VEV limit $a \to \infty$, the physics is weakly coupled on the CB, so we can trust the computations from directly using the UV Lagrangian. This means that the prepotential, in those regimes, must be compatible with the results that can be obtained from the UV theory.

Under these general constraints, the general expression for the prepotential is given by [148]

$$\mathcal{F}(a) = \frac{1}{2}\tau_{\text{UV}}a^2 + \frac{ib}{8\pi}a^2\ln\left(\frac{a^2}{\Lambda^2}\right) + \sum_{k=1}^{\infty}F_k\left(\frac{\Lambda}{a}\right)^{bk}a^2, \tag{3.25}$$

which is an expansion valid for the weak coupling region $|a| > \Lambda$ where $\Lambda$ is the dynamically generated scale. The second term comes from the perturbative 1-loop correction in the weak-coupling limit of the CB, i.e., the 1-loop running of Eq. (2.40), and the coefficient $b$ in this contribution has been defined in Eqs. (2.37) and (2.41). The third term arises from the non-perturbative corrections due to possible instanton corrections. The coefficients $F_k$ can be determined in different ways:

1. One way is to directly compute the instanton effects weighted by $e^{2\pi ni\tau}$. This technique has been developed for SU($N$) (and U($N$)) gauge groups in [76,77] and then generalized to other classical gauge groups such as SO($N$) and Sp($N$) [159,160].[27]

2. Alternatively, it is possible to use the fact that $F(a)$ is holomorphic and determine it by its behaviors around various singularities on $\mathcal{M}_{\text{CB}}$, which is the main spirit of the story developed in [20,21].

Let us spell out more details of this second approach. The singularities,[28] in $\mathcal{M}_{\text{CB}}$ are places where the special coordinates $(a, a_D)$ are not single-valued, so that the Coulomb branch EFT breaks down.[29] One simple example can be found at the origin $u = 0$ in the SU(2) SCFT [21], where $a = \sqrt{u/2}$. This is exactly the reason $a$ does not define a global coordinate on $\mathcal{M}_{\text{CB}}$. Physically, the appearance of these singularities on the CB is due to the fact that certain charged particles become massless at these singular points, which leads to the singularities in the EFT if one integrates them out [20,21].

The singularities have an interpretation in terms of monodromy in the special geometry. When we go around a singular point $s$ along a loop $\gamma$ on the $u$-plane, the special coordinates $(a^D, a)$ pick up a certain monodromy $\mathcal{M}_\gamma$ such that

$$\begin{pmatrix} a^D \\ a \end{pmatrix} \to \mathcal{M}_\gamma \begin{pmatrix} a^D \\ a \end{pmatrix}. \tag{3.26}$$

Because $a^D$ is a function of $a$, if there are no singularities, then there are no non-trivial monodromies. The monodromy characterizes the singularity on the CB. Furthermore, these monodromies are constrained by

$$\prod_{\{\gamma\}}\mathcal{M}_\gamma = \mathcal{M}_\infty, \tag{3.27}$$

meaning that the product of all the monodromy matrices associated to singularities on the CB must equal the monodromy matrix given by a path surrounding all the singularities at infinity in the $u$-plane. This is precisely the region where instanton corrections are negligible,

---

[27]In our convention Sp(1) $\simeq$ SU(2).

[28]Following [63] such a singularity refers to the one where the Kähler metric $g_{ij}$ develops a singularity, rather than the complex structure carried by $\mathcal{M}_{\text{CB}}$ developing a singularity. The latter has different physical implications, as studied in [161,162].

[29]The effective gauge coupling at the singularity takes a finite value except for the cusp type (see for example [163]).

so $\mathcal{M}_\infty$ can be determined from a one-loop computation in the UV theory. Eq. (3.27) then constrains the possible structure of the singularities on the CB, and it turns out to put remarkably powerful constraints on the coefficients $F_k$ in Eq. (3.25) [20,21] such that all $F_k$'s can be determined together under few additional assumptions, without doing any explicit instanton computations. We refer the readers to [20,21] for further details of the argument (or see for example reviews [84,164]).

> **Example 3.1:**
>
> We give an example of how monodromies characterize singularities on the CB. Let us consider the case of a U(1) vector multiplet coupled to a hypermultiplet of charge $\sqrt{p}$. The one-loop running gives
>
> $$\tau(a) = \frac{p}{2\pi i} \log\left(\frac{a}{\Lambda_{\mathrm{UV}}}\right), \tag{3.28}$$
>
> where $a$ is the vector multiplet scalar that defines the mass of the charged hypermultiplet. In particular, if we move on the CB close to $a \sim 0$, where the hypermultiplet become massless, and after going around $a = 0$, we see that the coupling in (3.28) picks up a shift
>
> $$\tau \rightarrow \tau + p. \tag{3.29}$$
>
> And from Eq. (3.20) we know that the corresponding SL$(2,\mathbb{Z})$ monodromy is
>
> $$M = \begin{pmatrix} 1 & p \\ 0 & 1 \end{pmatrix}. \tag{3.30}$$

More generally, we can have singularities where other particles become massless, not only the electrons but more generally, the dyons with electric and magnetic charges $(p,q)$. Each of their monodromies will be in the same conjugacy class as that for an electron in the hypermultiplet, but represented by a different SL$(2,\mathbb{Z})$ element instead. The associated monodromy turns out to be

$$M^{p,q} = \begin{pmatrix} 1+pq & p^2 \\ -q^2 & 1-pq \end{pmatrix}. \tag{3.31}$$

Another possibility is that multiple BPS particles become massless simultaneously at the singularity and this gives rise to more general monodromies, which are classified by the Kodaira classification. More information can be found in [163]. Among them, the important ones for our latter discussions are those where two BPS particles with charges $(p,q)$ and $(p',q')$ who are **mutually non-local**, meaning $(pq' - p'q) \neq 0$. Such a singularity gives rise to a strongly coupled SCFT, the so-called Argyres-Douglas theory, to which we will come back later.

## 3.3 Seiberg-Witten Solution to Quantum Prepotential

The electromagnetic monodromy tells us how the special coordinates $(a, a_D)$ behave around a singularity in $\mathcal{M}_{\mathrm{CB}}$. However, to determine $F_k$ and therefore the prepotential $\mathcal{F}$, one needs to know the exact expression of $(a, a_D)$. The question now is to identify multi-valued functions $(a, a_D)$ that display the required monodromies $M$ around each singularity in $\mathcal{M}_{\mathrm{CB}}$. This is known as a Riemann-Hilbert problem, and it has a unique solution up to multiplication by an entire function. It turns out such a solution $(a, a_D)$ has a nice geometric picture.

The main starting point is that the SL$(2,\mathbb{Z})$ invariance of the effective theory motivated Seiberg and Witten to resort to an extra geometric object: torus $T^2$, for which SL$(2,\mathbb{Z})$ acts

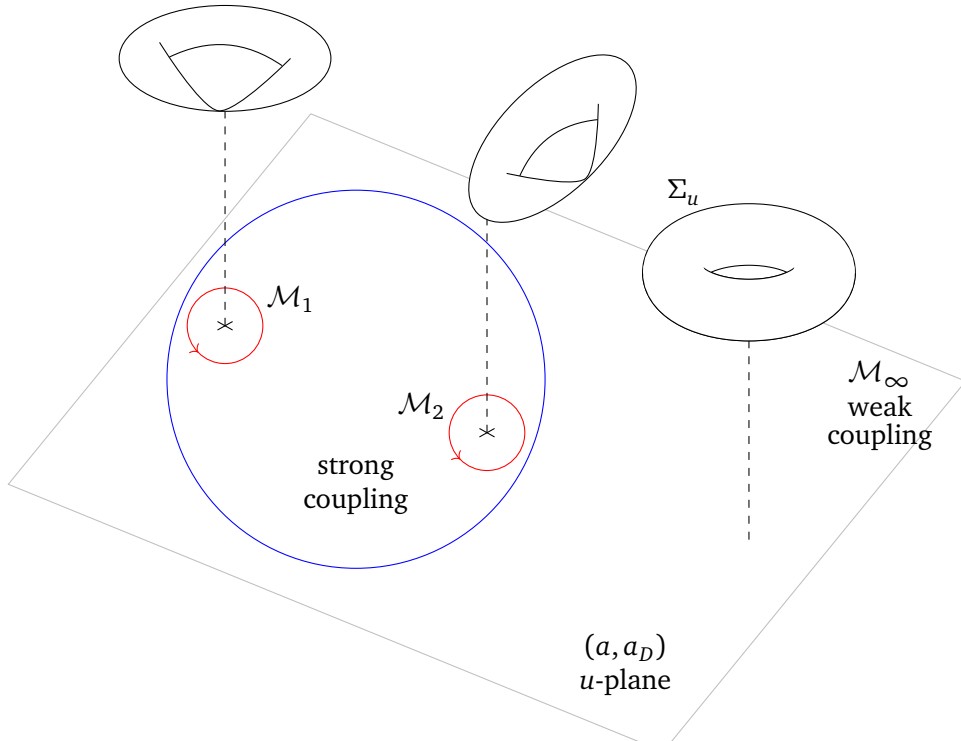

Figure 6: Emergent elliptic fibration on the $u$-plane.

as the modular symmetry group.[30] More precisely, by looking at the CB for an $\mathcal{N} = 2$ theory, schematically depicted in Figure 6, it is natural to interpret the effective coupling $\tau^{\text{IR}}$ on the CB as the complex structure of an emergent torus. An algebraic torus with a holomorphic section, dubbed as an elliptic curve and denoted as $\Sigma_u$, can be described algebraically as a Weierstrass model, which is defined by a hypersurface in $\mathbb{P}^2$,

$$y^2 = x^3 + f(u,m)x + g(u,m). \tag{3.32}$$

The complex structure $\tau^{\text{IR}}$ depends on $u$, a global coordinate on $\mathcal{M}_{\text{CB}}$, and the dependence is encoded in the coefficients $f$ and $g$ of Eq. (3.32). The parameter $m$ represents other possible mass deformations that can be turned on. And such a curve $\Sigma_u$ is known as a Seiberg-Witten curve.

In terms of this Seiberg-Witten curve, there is a canonical basis $\{A, B\} \in H_1(\Sigma_u, \mathbb{Z})$ of 1-cycles, and we can identify the special coordinates (3.23) on the CB as the period integrals,

$$a = \oint_A \lambda \,, \quad a^D = \oint_B \lambda \,, \tag{3.33}$$

where $\lambda$ is a certain 1-form differential called **SW differential**. It is a meromorphic 1-form on (3.32), subject to the Special Kähler constraint, i.e.,

$$\frac{\partial \lambda}{\partial u} = \Omega + d\phi = \frac{dx}{y} + d\phi \,, \tag{3.34}$$

where $\Omega = \frac{dx}{y}$ is a holomorphic non-vanishing 1-form on the SW curve and $d\phi$ refers to an arbitrary exact 1-form. Moreover, the residues of $\lambda$ depend on the potential mass deformations

---

[30]Here we are focusing on the rank-1 case. For a general rank-$r$ gauge group, the torus will be replaced by a genus-$r$ Riemann surface, see Section 4.

in the EFT. The gauge coupling $\tau^{\mathrm{IR}}$ is identified with the complex structure of $\Sigma_u$, and is given by the ratio of the following periods

$$\tau^{\mathrm{IR}} := \frac{\partial a_D}{\partial a} = \frac{\oint_A \Omega}{\oint_B \Omega}. \tag{3.35}$$

The mass of a BPS particle is given by the integral of $\lambda$ over a non-trivial 1-cycle on $\Sigma_u$. We have seen in the previous sections that the singularities on the CB can be interpreted as some points where certain BPS particles become massless. In terms of the cycles on $\Sigma_u$, this means that the singularities are points on the CB where the corresponding cycles degenerate. And the monodromy associated with such a singularity has a nice geometrical interpretation, as the Picard-Lefshetz transformation. Namely, near a singularity where an $A$-cycle vanishes, then the other cycle $B$ transforms according to the Picard-Lefshetz formula,

$$B \to B - (B \cdot A)A, \tag{3.36}$$

where $A \cdot B$ denotes the algebraic intersection number between these two cycles.

The main gist is that we can translate the problem of solving the CB EFT, i.e., $(a, a_D)$, into a problem of identifying the correct SW description in terms of the SW curve and the differential, $(\Sigma_u, \lambda)$, which is known in mathematics to define a special Kähler geometry. This allows us to geometrize the non-perturbative physics of $\mathcal{N} = 2$ theories in a controlled way.

What is still missing is the physical meaning of the emergent torus and its differential. The answer to this question comes naturally from 6d $\mathcal{N} = (2, 0)$ theories, and it will be the main topic of Section 4.

## 3.4 Scale-invariant Seiberg-Witten Solutions and Argyres-Douglas Theories

We have briefly introduced some salient aspects of Seiberg-Witten geometry $(\Sigma_u, \lambda)$ of an $\mathcal{N} = 2$ theory and now we would like to show how it sheds light on studying SCFTs at its IR RG fixed points. As alluded to, a nontrivial superconformal fixed point arises at a point in $\mathcal{M}_{\mathrm{CB}}$ where two mutually non-local charged particles become massless, but how is this reflected from the SW geometry $(\Sigma_u, \lambda)$? Historically, what people did to identify an SCFT on a CB for a 4d $\mathcal{N} = 2$ asymptotically free theory, was to first extract the SW curve and SW differential $(\Sigma_u, \lambda)$ on its CB, and on some patches in $\mathcal{M}_{\mathrm{CB}}$, the SW geometry $(\Sigma_u, \lambda)$ has certain emergent scale invariance, which can be used to identify the superconformal fixed points. This is exactly what Argyres and Douglas did in [22]. We can view this way as a top-down approach in the sense that the SW geometry $(\Sigma_u, \lambda)$ is derived from a UV Lagrangian.

More recently, a more direct approach has been developed to look for SCFTs, using a bottom-up approach [163, 165, 166]. Instead of requiring a UV Lagrangian in the first place, they looked directly at the IR object $(\Sigma_u, \lambda)$ and demanded manifest scale invariance of the geometry to encode the conformal invariance of the theory of interest. Recall that given a 4d $\mathcal{N} = 2$ SCFT, the CB is parameterized by the VEV of the chiral primary operators listed in Def. 2.1, hence admits a $\mathbb{C}^*$ action which descends from the $U(1)_r \times R^+$ symmetry at the fixed point, where $R^+$ denotes the dilatation symmetry which is spontaneously broken by the scalar VEVs. Correspondingly it is expected that the SW geometry $(\Sigma_u, \lambda)$ would possess such a $\mathbb{C}^*$ action (locally). Note that in this bottom-up approach, the pair $(\Sigma_u, \lambda)$ is ad hoc and *a priori* not necessarily related to any $\mathcal{N} = 2$ theories. Nevertheless, it turns out that this bottom-up approach provides a powerful way to identify new SCFTs [163, 165, 166].

Let us take the rank-1 case as an example to illustrate the main idea of this approach. As alluded, a Seiberg-Witten curve can be written in the Weierstrass form

$$\Sigma_u \,:\, y^2 = x^3 + f(u)x + g(u). \tag{3.37}$$

We here have arranged ourselves in a parametrization such that at $u = 0$ the theory is scale invariant and thus superconformal. The strategy is to impose $\mathbb{C}^*$ actions on the Weierstrass description of the curve $\Sigma_u$ to realize the scale invariance explicitly. As one can see, this puts constraints on the possible SW geometry. The immediate consequence is that it requires the SW curve $\Sigma_u$ to possess only one metric singularity, as having two singularities naturally introduces a scale that breaks the scale invariance. The parameter $u$ naturally supports a $\mathbb{C}^*$ charge, since $u$ is supposed to be the VEV of a CB chiral primary operator charged under the $U(1)_r$ R-symmetry. Under a $\mathbb{C}^*$ action, $u$ behaves as

$$u \to \xi^r u, \tag{3.38}$$

where $\xi \in \mathbb{C}^*$, and $r$ is the $U(1)_r$ charge of $u$ and also the conformal dimension $\Delta$ of the CB chiral primary operator. By demanding $\mathbb{C}^*$ symmetry at the level of $\Sigma_u$, we obtain that $f$ and $g$ must be monomials in $u$. As the last ingredient, we use the relation (3.21) between the central charge and the mass of BPS particles with the normalization that the masses have charge 1 under the R-symmetry $U(1)_r$. Meanwhile, as in Eq. (3.22), the central charge can be written in terms of the special coordinates which are related to the SW differential through Eq. (3.33). We hence conclude that the SW differential $\lambda$ has the $U(1)_r$ charge as

$$r[\lambda] = 1. \tag{3.39}$$

Meanwhile it follows from Eq. (3.34) that

$$\frac{\partial \lambda}{\partial u} \sim \frac{dx}{y}. \tag{3.40}$$

Using Eqs. (3.39) and (3.40), we can obtain the scaling dimension of the chiral primary operator $u$ on the $\mathcal{M}_{CB}$:

$$\Delta[u] = r[u] = \frac{6}{6-n} \text{ or } \frac{4}{4-n} \geqslant 1, \tag{3.41}$$

with $n \in \mathbb{Z}_+$, where the requirement that they are not smaller than 1 comes from unitarity constraints on 4d conformal field theories[31] [167]. It turns out that the only possible nontrivial rank-1 CB are those in Table 3, where we omitted the trivial case $\Delta(u) = 1$. All these possibilities are realized by non-trivial theories, in particular by the Argyres-Douglas theories [22], whose definition states they have fractional scaling dimension $\Delta(u)$. From the Class $\mathcal{S}$ perspective, they are constructed by irregular punctures on the Gaiotto curve, which we will introduce in Section 4.8.2.

Higher rank versions of these strongly coupled AD theories have also excited interests to study them from the gravitational point of view via the AdS/CFT correspondence. Recently the holographic dual of the large central charge cases of all $(A, A)$ AD theories (that we will introduce properly in Sections 4.8.2 and 5) have been proposed in [168, 169].[32]

---

[31]To be more specific, the unitarity of a $d$ dimension CFT requires all the scalars to have the conformal dimension $\Delta \geqslant \frac{d-2}{2}$ or $\Delta = 0$. Nevertheless, in a certain SCFT when the coordinate ring of the CB is not freely generated, such a unitarity constraint can be violated. This is due to non-trivial relations between CB operators such that the coordinates $u$'s in CB are not generically the VEVs of primary operators in the SCFT hence their scaling dimensions $\Delta(u)$ can be less than 1, see more discussions in [161].

[32]Another interesting result has been proposed in [170]. The authors considered type IIB string theory compactified on a K3 surface wrapped by $n$ D7-branes. The low-energy effective action in 4d is pure $SU(n)$ $\mathcal{N} = 2$ SYM on whose CB the $(A_1, A_{n-1})$ AD theory is realized. They argued that each D7-brane splits into a pair of exotic branes, and when $n$ exotic branes of the same kind collide at the same point, the low-energy worldvolume dynamics of the stack of such exotic branes is given by the $(A_1, A_{n-1})$ AD theory. We refer to the original paper for details of the nature of those exotic branes.

**Exercise 3.2** Using Eqs. (3.39) and (3.40), deduce the conformal dimensions of $u$, i.e., $\Delta[u] > 1$, for the rank-1 theories in Table 3 (see [63] for hints).

Table 3: Conformal dimensions of chiral primary operator $u$ in various rank-1 SCFTs. The theory $H_0$ is the simplest AD theory, i.e., $(A_1, A_2)$, with a trivial global symmetry. The names of the theories denote the singularities probed by D3-branes in F-theory (which is $A_n$ for the $H_n$-type singularities) [171–180]. Other details of these theories will be given in Section 5.4.2.

| Theory | Flavor group | $\Delta[u]$ |
|--------|--------------|-------------|
| $H_0$ | — | 6/5 |
| $H_1$ | SU(2) | 4/3 |
| $H_2$ | SU(3) | 3/2 |
| $D_4$ | SO(8) | 2 |
| $E_6$ | $E_6$ | 3 |
| $E_7$ | $E_7$ | 4 |
| $E_8$ | $E_8$ | 6 |

## 3.5 SCFT data at the Argyres-Douglas point from a Coulomb Branch EFT

We have briefly shown how SW geometry $(\Sigma_u, \lambda)$ helps us identify its corresponding SCFT at its superconformal fixed point by imposing scale invariance. However, there are some caveats throughout this bottom-up approach. For instance, the SW geometry does not have sufficient information to fully classify 4d $\mathcal{N} = 2$ SCFTs [163, 165]. In particular, given a singular SW curve with its SW differential, it may correspond to multiple distinct SCFTs. The prototypical example is the $\mathcal{N} = 4$ SYM (viewed as an $\mathcal{N} = 2$ theory with one adjoint hypermultiplet) and the $\mathcal{N} = 2$ theory coupled to four fundamental hypermultiplets. They have the same SW geometry $(\Sigma_u, \lambda)$ on their CBs [21], but they are completely different 4d theories.

Indeed, it is necessary to study general deformations of SW geometry, such as mass deformations, in a bid to extract much more information about an SCFT from its CB EFT. In particular, since the CB is parameterized by the VEVs of the chiral primaries listed in Def. 2.1, it is possible to extract the spectrum of those protected operators just from the EFT (as we have reviewed for the rank-one case in the last section). The couplings and the mass deformations also manifest themselves from the CB EFT, and they can be learned from the fully deformed SW curve.

Another possible piece of data of an SCFT that can be extracted from the CB EFT is its conformal and flavor central charges. The supersymmetry Ward identities relate these CFT observables to the 't Hooft anomalies involving the U(1)×SU(2) R-symmetry, Lorentz symmetry and flavor symmetries. The 't Hooft anomalies can in turn be determined from the CB EFT by anomaly matching [75]. Indeed, the (conformal and flavor) central charges of the SCFT receive a contribution from the free massless fields on the CB, but there are also Wess-Zumino (WZ) contributions to the relevant anomalies. Schematically,

$$(a, c, k_G)_{\text{SCFT}} = (a, c, k_G)_{\text{CB}}^{\text{free}} + \text{ WZ contributions} , \qquad (3.42)$$

where the WZ contributions exactly reproduce $R(A), R(B)$ in (2.50) [75, 145].

# 4 Class $\mathcal{S}$ Constructions from Five-branes

In this section, we explore the constructions of 4d $\mathcal{N} = 2$ supersymmetric theories from the dimensional reduction of a 6d theory.[33] In particular, we focus on Class $\mathcal{S}$ theories (where "$\mathcal{S}$" stands for "Six"). The starting point for a Class $\mathcal{S}$ theory is a 6d $\mathcal{N} = (2, 0)$ SCFT $\mathcal{T}_{\mathfrak{g}}$, labelled by a simply-laced Lie algebra $\mathfrak{g}$. We then compactify the theory on a Riemann surface $\mathcal{C}$, which is called the **UV curve**. Such a curve can have punctures, for which we need to specify boundary conditions for certain fields, to be explained in the bulk of this section. The collection of the Lie algebra, the UV curve and the data coming from the punctures $p$, will define a 4d $\mathcal{N} = 2$ Class $\mathcal{S}$ theory, which we denote as $\mathcal{T}(\mathfrak{g}, \mathcal{C}, p)$.

We will start by reviewing $\mathcal{N} = (2, 0)$ theories in 6d in Section 4.1. Section 4.2 describes how to obtain the Seiberg-Witten solution for an Abelian 6d $\mathcal{N} = (2, 0)$ theory, and we will introduce the UV curve $\mathcal{C}$ and the precise Class $\mathcal{S}$ construction in Section 4.3. Section 4.4 is devoted to the quantum prepotential obtained by solving the classical Hitchin system, while in Section 4.5 we introduce the concept of punctures for $\mathcal{C}$. We reserve Section 4.6 for examples, and in Section 4.7 we quickly review the AGT correspondence. Finally, further generalizations are briefly mentioned in Section 4.8. In particular, while in the preceding sections we focus on $A_{N-1}$ type theories, we list possible extensions to other structure groups and UV curves in Section 4.8.1. More general punctures are introduced in Section 4.8.2, and finally in Section 4.8.3 we describe the twisting procedure with an outer-automorphism of the Lie algebra $\mathfrak{g}$ of the theory.

## 4.1 A Lightning Review of $\mathcal{N} = (2, 0)$ Theories

A 6d $\mathcal{N} = (2, 0)$ SCFT $\mathcal{T}_{\mathfrak{g}}$ is the *maximal* superconformal theory that may exist. On the one hand, 6 is the largest number of dimensions where supersymmetry and conformal symmetry are compatible [97]. On the other hand, it has the largest possible superconformal algebra in 6d, whose real form is given by $\mathfrak{osp}(8^*|4)$ [183, 184]. This superalgebra contains $\mathfrak{so}^*(8) \oplus \mathfrak{usp}(4)_R = \mathfrak{so}(2, 6) \oplus \mathfrak{so}(5)_R$, the bosonic subalgebras for the conformal and the R-symmetries respectively. Such free $\mathcal{N} = (2, 0)$ theories consist of Abelian tensor multiplets each containing a real self-dual 2-form gauge field $B_{\alpha\beta}$, spinors $\lambda$, and five scalars $\Phi_I$. With respect to the R-symmetry, they transform respectively as a singlet, a **4** and a **5** irreducible representations of $\mathfrak{so}(5)$.

For a very long time, it was believed that interacting field theories in 6d could not exist, because interaction terms in the Lagrangian are non-renormalizable in $d \geq 5$ dimensions. However, people have realized that interacting 6d $\mathcal{N} = (2, 0)$ theories can be engineered from String/M-theory, for instance by considering a stack of M5-branes in M-theory, with the decoupled center of mass degrees of freedom removed. In particular, $N$ coincident M5-branes realize a $\mathcal{T}_{\mathfrak{su}(N)}$ SCFT this way, with no Lagrangian description.

Another possible way of constructing an interacting 6d $\mathcal{N} = (2, 0)$ SCFT is to place type IIB string theory on the singular geometry $\mathbb{R}^{1,5} \times \mathbb{C}^2/\Gamma$, where $\Gamma \subset SU(2)$ is a finite subgroup of $SU(2)$ according to the ADE classification [185]. This construction includes the aforementioned $\mathcal{T}_{\mathfrak{su}(N)}$ SCFTs as the special case $\Gamma = \mathbb{Z}_N$. It also facilitates the study of the moduli space of these theories, since we can move along it by blowing up the singularity at the origin of $\mathbb{C}^2/\Gamma$ into a collection of finite-size 2-cycles and the resolved manifold is a hyperkähler *ALE* manifold. The number of such 2-cycles is equal to the rank of the algebra $\mathfrak{g}$, and for each of these exceptional 2-cycles, one can associate VEVs of the above five scalar $\Phi_I$'s to the integral of the NS-NS two-form field, the R-R two form field and the triplet of symplectic forms. Hence,

---

[33]A detailed discussion of 6d SCFTs is beyond the scope of the current review. The reader may wish to consult [181, 182] as an entry point to the literature on this vast subject.

the vacua of $\mathcal{T}_{\mathfrak{g}}$ are parameterized by said VEVs, giving

$$\mathcal{M}_{\mathfrak{g}} = \mathbb{R}^{5r_{\mathfrak{g}}}/\mathcal{W}_{\mathfrak{g}}, \tag{4.1}$$

where $r_{\mathfrak{g}}$ is the rank of $\mathfrak{g}$, and $\mathcal{W}_{\mathfrak{g}}$ is its Weyl group.

A useful description of such theories is obtained by compactifying them on a circle $S^1_R$ of size $R$. The fields in the Abelian 6d theory $(B, \lambda, \Phi)$ are reduced to $(A, \lambda, \Phi)$, where now $A$ is a 1-form gauge field in 5d coming from the reduction of the $B$ field, while $\lambda$ and $\Phi$ remain respectively as fermions and scalars, but now in 5d. The resulting theory is the 5d $\mathcal{N} = 2$ Abelian SYM. More generally, due to the maximal supersymmetry it is expected that the $S^1_R$ compactifcation of a general interacting 6d $\mathcal{N} = (2, 0)$ SCFT is described, below the Kaluza-Klein (KK) scale, by a 5d $\mathcal{N} = 2$ non-Abelian SYM with gauge coupling $g^2_{\text{YM}} \sim R$. In fact, by matching certain BPS states on the 5d Coulomb branch and those on the 6d tensor branch, one recovers the ADE classification of the 6d $\mathcal{N} = (2, 0)$ SCFTs [186]. Importantly the 5d SYM secretly remembers the 6d circle through its instanton particles, which are charged under the topological current $J = \star \text{tr}(F \wedge F)$ and have mass $m_I \sim \frac{1}{g^2_{\text{YM}}}$. They are naturally identified with the KK modes of the $S^1_R$ compactification [187, 188]. This feature has made possible the determination of protected observables in the 6d $(2, 0)$ SCFT from the 5d SYM by keeping track of the KK tower (such as the 6d superconformal index in [189–191]).[34]

## 4.2 Seiberg-Witten Solution from the Abelian $\mathcal{N} = (2, 0)$ Theory

It is also possible to go one step further, and compactify the theory on another circle (in general we will compactify on a Riemann surface) and naturally we expect to obtain a 4d QFT in the low-energy limit. Before moving on to the Class $\mathcal{S}$ construction, let us see how this is related to the Seiberg-Witten story.

Recall from Section 3 that the EFT for a general 4d $\mathcal{N} = 2$ SCFT is encoded by its SW geometry. Focusing on a theory of rank 1, this geometry consists of an elliptic curve fibered over the $u$-plane (which is parametrized by the VEV of the scalar in the 4d vector multiplet in the EFT).

The effective action can be described by said curve $\Sigma_u$, and the SW differential $\lambda$. The pair $(\Sigma_u, \lambda)$ determines the prepotential that contains the information about the dynamics on the Coulomb branch at long wavelengths. This geometric structure has a physical meaning in terms of the 6d theory. We can look at the 6d $\mathfrak{u}(1)$ $\mathcal{N} = (2, 0)$ theory reduced on the SW curve $\Sigma_u$, and from this we obtain the 4d $\mathcal{N} = 2$ EFT. In particular, the 4d EFT has a gauge field and its dual $(A_\mu, A^D_\mu)$, and they come from the reduction on the canonical basis of $H_1(\Sigma_u)$ (and its Hodge dual) of the self-dual 2-form $B$ in the 6d $\mathcal{N} = (2, 0)$ theory. Moreover, we know from Section 3.3 that the masses of BPS particles are given by the integrals of the Seiberg-Witten differential $\lambda$ over 1-cycles. From the 6d perspective, such BPS particles are coming from BPS strings wrapping those 1-cycles.

This can be nicely understood from the M-theory point of view, where we have a single M5-brane wrapped around the SW curve. The BPS strings are coming from certain supersymmetric M2-branes, and the infinitesimal tensions of the M2-branes, which extend on one other transverse direction, naturally give rise to the Seiberg-Witten differential $\lambda$. If the 4d theory has rank $r > 1$, so that at a generic point of the CB the gauge group is broken to $U(1)^r$, the only difference is that the SW curve wrapped by the single M5-brane is no longer elliptic, rather it has genus equal to the rank [28].

---

[34]Incidentally, here we can see why there is no obvious Lagrangian description of the $(2, 0)$-theory: a naive dimensional reduction of a 6d action leads to a 5d action directly, not inversely, proportional to $R$. For more reasons against the existence of a Lagrangian, see [192, 193].

We have now lifted the effective 4d Coulomb Branch physics to a 6d free theory compactified on a Riemann surface. The next step will be to show that such a 6d theory has an interacting UV completion using the Class $\mathcal{S}$ construction.

## 4.3 The UV Curve $\mathcal{C}$ and Class $\mathcal{S}$ Construction

We have shown in the previous section that the effective theory in 4d on the Coulomb branch can be uplifted to a 6d $\mathcal{N} = (2,0)$ Abelian theory on $\Sigma_u$ – this theory corresponds to a single M5-brane wrapping the SW curve. The idea of this section is to find the interacting UV theory in 6d that flows to such Abelian theory. This UV theory will be the interacting $\mathcal{N} = (2,0)$ SCFT defined on $\mathbb{R}^4 \times \mathcal{C}$, where $\mathcal{C}$ is a punctured Riemann surface called the **UV curve** or **Gaiotto curve** (in the SU($N$) case it will correspond to a stack of M5-branes wrapping $\mathcal{C}$).

The first step in the Class $\mathcal{S}$ construction is to identify the 4d $\mathcal{N} = 2$ supersymmetry from the 6d parent. In the 6d theory we have 16 supercharges in total, and a generic surface $\mathcal{C}$ may not preserve the 8 supercharges we want in 4d. It is necessary to perform a **partial topological twist** that mixes some R-symmetries into some rotational symmetries. Let us consider the sub-algebras $\mathfrak{so}(2)_r \oplus \mathfrak{so}(3)_R \subset \mathfrak{so}(5)_R$ of R-symmetry and $\mathfrak{so}(1,3)_{4d} \oplus \mathfrak{so}(2)_{\mathcal{C}} \subset \mathfrak{so}(2,6)$ of Poincaré symmetry. The twisted rotation symmetry $\mathfrak{so}(2)_{\text{twist}}$ on $\mathcal{C}$ is defined to be the diagonal part of $\mathfrak{so}(2)_{\mathcal{C}} \oplus \mathfrak{so}(2)_r$, and the residual bosonic symmetries, including Lorentz and R-symmetries are

$$\mathfrak{so}(1,3)_{4d} \oplus \mathfrak{so}(2)_{\text{twist}} \oplus \mathfrak{so}(3)_R. \tag{4.2}$$

Compactifying the theory on $\mathcal{C}$ and performing the partial topological twist, we obtain a system preserving $\mathfrak{so}(3)_R = \mathfrak{su}(2)_R$ R-symmetry, and two Weyl supercharges transforming in the doublet, which exactly provides the amount of supersymmetry we need to obtain a 4d $\mathcal{N} = 2$ Poincaré supersymmetry algebra.

The vacua of the 6d theory are parameterized by the VEV of the scalar fields $\Phi_I$ that take values in the Cartan subalgebra of the gauge Lie algebra $\mathfrak{g}$, with $I = 1, \ldots, 5$. Initially, these five fields could be democratically rotated into each other by means of the $\mathfrak{so}(5)_R$ symmetry. However, this is no longer the case after the partial topological twist. Three of the fields, say $\Phi_3$, $\Phi_4$, and $\Phi_5$, are charged under the untouched R-symmetry $\mathfrak{so}(3)_R = \mathfrak{su}(2)_R$. Thus these are the fields that parametrize the Higgs branch of the 4d theory. The remaining two fields, $\Phi_1$ and $\Phi_2$ are charged under the $\mathfrak{so}(2)_r = \mathfrak{u}(1)_r$ R-symmetry, so they parametrize the Coulomb branch. Note that after the partial topological twist, this R-symmetry is mixed with the rotational symmetry in $\mathcal{C}$, namely these fields are no longer scalars. It is convenient to write

$$\Phi_z \equiv \Phi_1 + i\Phi_2, \tag{4.3}$$

so that $\Phi_z dz$ is a (1,0)-form on $\mathcal{C}$ with holomorphic coordinate $z$, called the **Hitchin field** (or **Higgs field**[35]). As we will see, the 4d Coulomb branch is determined by BPS configurations of the Hitchin field on $\mathcal{C}$ modulo gauge transformations.

The next step is to relate the Seiberg-Witten curve that describes the low-energy dynamics of our theory to the ingredients of the UV construction: the Gaiotto curve $\mathcal{C}$, the Hitchin field $\Phi_z$, and the gauge algebra which we take $\mathfrak{g} = A_n$ (we postpone comments on the other possible choices of Lie algebra to Section 4.8.1) . Following [33] (see also [64]), let us consider $T^*\mathcal{C}$, i.e., the canonical line bundle over $\mathcal{C}$. We define coordinates $(z, x)$ for $T^*\mathcal{C}$ with $x \in \mathbb{C}$ parametrizing the fiber direction. It can be shown (and we shall elaborate further in Section

---

[35]The name "Higgs field" comes from the mathematical literature regarding Higgs bundles and Hitchin integrable systems. As it has nothing to do with the usual Higgs field in the physics sense (and moreover we just saw that it parametrizes the Coulomb branch of the moduli space), in the physics literature it is more often referred to as "Hitchin field".

4.4) that the Seiberg-Witten curve $\Sigma \subset T^*\mathcal{C}$ is a $n$-sheeted cover of $\mathcal{C}$, given by the equation

$$\langle \det(x - \Phi_z) \rangle = \det(x - \varphi_z) = 0. \tag{4.4}$$

From the 4d point of view, this equation specifies a genus-$n$ curve fibered over the Coulomb branch, as expected. The second ingredient we need to extract the IR dynamics is the SW differential. With our choice of coordinates on $T^*\mathcal{C}$, it is simply given by

$$\lambda = x\,dz. \tag{4.5}$$

Once again, this construction has a nice interpretation in M-theory [28]. In the UV, we have a stack of $n+1$ M5-branes wrapping $\mathcal{C}$. The eleven dimensions split as $4+4+3$: the first 4 are flat and the branes are extended on them, the 4d low-energy theory lives there; the next 4 correspond to $\mathcal{C}$ and its transverse directions making up $T^*\mathcal{C}$; and the last 3 are also flat but transverse to the M5-branes. The scalar fields $\Phi_I$ describe the displacement of the M5-branes in those transverse directions: $\Phi_3$, $\Phi_4$ and $\Phi_5$ are translations in the last three directions, and $\Phi_z, \Phi_{\bar{z}}$ in the two directions transverse to $\mathcal{C}$ inside $T^*\mathcal{C}$. The idea is that this stack of M5-branes, when going to the infrared, merges into one single M5-brane (this is how we get our low-energy Abelian 6d $\mathcal{N} = (2, 0)$ theory) that takes a complicated shape with possibly a non-trivial genus (for rank higher than 1). Specifically, the shape of the M5-brane, roughly speaking governed by the coordinate $x$, is given by (4.4). The meaning of this equation is that, for a given configuration of displacement of the M5-branes at UV (namely a given configuration of the Hitchin field $\Phi_z$), the possible values of $x$ are the eigenvalues of $\Phi_z$. Then, if we consider a different configuration of $\Phi_z$, we will be moving on the Coulomb branch, and in doing so we will recover our expected fibration (4.4).

The last ingredient of the construction, which greatly enriches the possible theories we can build in Class $\mathcal{S}$, are **punctures**. They can be most easily motivated from the M-theory perspective, or rather the dual type IIA brane diagram.[36] In this framework, the 4d theory is obtained from a Hanany-Witten (HW) setup with D4-branes hanging from NS5-branes. The D4-branes come from M5-branes which wrap the M-theory circle, while the NS5-branes come from M5-branes which have fixed positions in said circle. Having semi-infinite D4-branes on the right and the left of the brane diagram leads to a non-dynamical flavor symmetry node in the corresponding quiver, since these D4-branes are much heavier than those hanging between two NS5-branes. Now we can bring this configuration to a more familiar form by lifting it to M-theory and compactifying the internal directions of the M5 branes (coming from the direction longitudinal to the D4s and transverse to the NS5s in the IIA brane diagram plus the M-theory circle). We perform this compactification by adding a finite number of points corresponding to infinity, one for each semi-infinite NS5- or D4-brane. In this way, we will end up with dynamical M5-branes wrapped around a Riemann surface with some special points $p_j$, corresponding to the heavy M5-branes: these will be the punctures.

Motivated by this, we can now introduce punctures directly in the M-theory setup. We implement this by considering a Riemann surface $\mathcal{C}$ with several marked points $\{p_i\}_i$, and adding defect M5-branes spanning $\mathbb{R}^{1,3} \times T^*_{p_i}\mathcal{C}$ to our previous setup where the original stack of M5-branes span $\mathbb{R}^{1,3} \times \mathcal{C}$. More abstractly, we can also understand punctures in the 6d $\mathcal{N} = (2,0)$ SCFT in general as codimension-2 defects with prescribed boundary conditions. They will correspond to poles in the Hitchin field at points $p_i \in \mathcal{C}$ which carry information about the global symmetry of the 4d theory. The rough idea is that, via (4.4), they will become poles of $x(z)$, and when integrating the Seiberg-Witten differential around the $p_i$ we will pick up

---

[36]There is an interesting holographic dual to the class $\mathcal{S}$ construction in the string/M-theory picture. The M-theory background corresponding to holographic duals of class $\mathcal{S}$ theories were first penned down in [34]. The holographic duals to the T-dual type IIA were explored in [194–196]

their residue, which is thus identified with the mass parameters for the BPS particles. There can be several types of punctures corresponding to different flavor symmetries, but we postpone a more detailed discussion on this until Section 4.5.

## 4.4 Quantum Geometry From Classical Hitchin System

In the meantime, we would like to make more precise how we can recover the effective theory useful for the 4d description from the 6d $\mathcal{N} = (2,0)$ theory compactified on the UV curve $\mathcal{C}$. This is done using the Hitchin integrable system [197], whereby we can obtain the exact Coulomb branch geometry by solving some classical equations. In order to explain this relation, we make use of the chains of compactifications, shown in Figure 7, from a 6d $\mathcal{N} = (2,0)$ theory to a 3d $\mathcal{N} = 4$ theory.

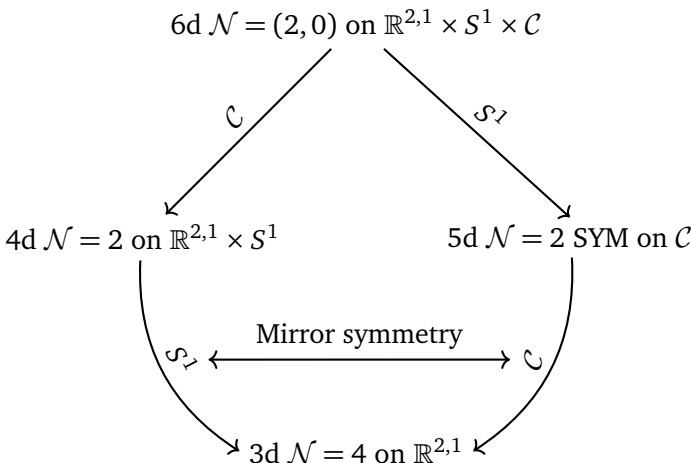

Figure 7: The relations between 6d $\mathcal{N} = (2,0)$ SCFT, 5d $\mathcal{N} = 2$ SYM, 4d $\mathcal{N} = 2$ Class $\mathcal{S}$ theory and its 3d $\mathcal{N} = 4$ cousin from successive compactifications. The mirror symmetry relates two UV descriptions of the same 3d $\mathcal{N} = 4$ SCFT in the IR.

The arrows can be understood as follows. Compactifying a 6d $\mathcal{N} = (2,0)$ theory on a Riemann surface $\mathcal{C}$ leads to a rich class of strongly coupled 4d $\mathcal{N} = 2$ theories that we have introduced in the previous sections. To extract physical information of the resulting 4d theory from the 6d setup, it turns out to be useful to compactify further on a circle $S^1$ of radius $R$. The 3d $\mathcal{N} = 4$ theory is generally a nontrivial SCFT which encodes physics of the 4d parent. In particular, the Coulomb branch of the 4d theory gets enhanced by the VEVs of the BPS line operators on $S^1$ in the 3d limit and the full 3d CB EFT is described by an $\mathcal{N} = 4$ sigma model with a hyperkähler (HK) target space $\mathcal{M}$ of complex dimension $2r$ (twice the dimension of the 4d CB) [35]. From the 3d perspective, $\mathcal{M}$ is the branch of the vacuum moduli space preserving the $\mathfrak{so}(3)_R$ symmetry in (4.2).

This description of the HK manifold in terms of 4d line operators may be a bit abstract. In fact the structure of the HK manifold $\mathcal{M}$ can be made more transparent if we reverse the order of the compactifications.[37] As mentioned before, the 6d $\mathcal{N} = (2,0)$ theory is non-Lagrangian, but its compactification on a circle $S^1$ of radius $R$ is described by a 5d maximally supersymmetric Yang-Mills theory at low energy. If we further compactify such theory on $\mathcal{C}$, then the moduli space of the 3d theory is governed by the set of solutions to the BPS equations for the 5d fields

---

[37]Note that the order of the compactification from six to three dimensions is irrelevant for the moduli space $\mathcal{M}$ [35]. This is tied with the fact that due to the topological twisting, the BPS-protected quantities do not care about the relative scales between $\mathcal{C}$ and $S^1$.

on $\mathcal{C}$, which turn out to be a Hitchin system of equations on $\mathcal{C}$ [198]. As we explain below, the HK manifold $\mathcal{M}$ coincides with the moduli space of solutions to the Hitchin equations on $\mathcal{C}$.

To be more explicit, let us describe the moduli space of the 3d $\mathcal{N} = 4$ theory obtained from a 6d $\mathcal{N} = (2, 0)$ theory first compactified on a circle $S^1$ and then on $\mathcal{C}$. If we denote the scalars of the 5d SYM theory as $\Phi_I$, $I = 1, \ldots 5$, the Hitchin field as the combination in (4.3) and the gauge field as

$$A = A_z dz + A_{\bar{z}} d\bar{z},\tag{4.6}$$

then the Hitchin equations are the following:

$$
\begin{aligned}
F + \left[\Phi_z, \overline{\Phi}_z\right] &= 0,\\
\bar{\partial}_A \Phi_z \equiv d\bar{z}\left(\partial_{\bar{z}} \Phi_z + [A_{\bar{z}}, \Phi_z]\right) &= 0,\\
\partial_A \overline{\Phi}_z \equiv dz\left(\partial_z \overline{\Phi}_z + [A_z, \overline{\Phi}_z]\right) &= 0.
\end{aligned}
\tag{4.7}
$$

The space of solutions of these equations on $\mathcal{C}$ describes a branch of the moduli space of the 3d theory where the $\mathfrak{so}(3)_R$ symmetry in the twist compactification is preserved (see (4.2)) and this is to be identified with the aforementioned HK manifold $\mathcal{M}$ which is also known as the Hitchin moduli space.

At this point we know that the resulting 3d theory should be the same for both the two ways of compactifying the 6d theory. However, there is a non-trivial map between the two RG flows (see Figure 7). They are related by the 3d mirror symmetry [199] as explained in [29] (see also [200]).[38] If we want to learn about the Coulomb branch of the 4d theory, we can look at the Coulomb branch of the 3d theory, which inherits the former with enhancements due to line operators wrapping the circle. We have also just shown that the moduli space of the 3d theory preserving the $\mathfrak{so}(3)_R$ symmetry can be described by the Hitchin system coming from the compactification of the 5d SYM theory on a Riemann surface [29, 198, 200] whose solutions parameterize the Higgs branch of the 5d theory. By 3d mirror symmetry, the Coulomb branch in 3d coming from the compactification on a circle $S^1$ is the same as the Higgs branch of the 5d theory compactified on $\mathcal{C}$, which is nothing but the HK manifold $\mathcal{M}$.

Finally, if we suppress the coordinates from the 4d line operators wrapping the compactification circle, which parametrize fiber directions of $\mathcal{M}$, the 4d Coulomb branch is identified with the base $B$ of the Hitchin integrable system. As an example, if $\mathfrak{g} = A_{k-1}$, then the base $B$ is

$$B = \bigoplus_{r=2}^{k} H^0(\mathcal{C}, K_{\mathcal{C}}^{\otimes r}),\tag{4.8}$$

which are holomorphic sections of gauge-invariant monomials of the Hitchin field of the form $\text{Tr}\left(\Phi_z^r\right)$. Then the Seiberg-Witten curve is given by the spectral curve of the integrable system (4.7), which is precisely (4.4).

Let us stress for the last time that we have obtained the quantum Coulomb branch EFT of a generally strongly-coupled 4d $\mathcal{N} = 2$ theory from the classical solutions to BPS equations that describe the moduli space of the 5d $\mathcal{N} = 2$ SYM. This is made possible by mirror symmetry, which relates the former to the Higgs branch of the mirror theory, and is therefore protected from the quantum corrections [201].

## 4.5 Codimension-2 Defects and Punctures on $\mathcal{C}$

As we have seen in the previous section, the integrable system for the 4d $\mathcal{N} = 2$ theory has been associated to a Hitchin system on $\mathcal{C}$ with gauge group SU($N$) [197]. We consider $\Phi_z$ as

---

[38]Here is one quick way to see this. The scalar $\Phi_z$ in (4.3) which is charged under $\mathfrak{u}(1)_r$ and thus relevant for describing the 4d $\mathcal{N} = 2$ CB is one of two complex scalars in the hypermultiplet of the 5d $\mathcal{N} = 2$ SYM regarded as an $\mathcal{N} = 1$ theory. Consequently, $\Phi_z$ naturally parametrize the CB of the 4d $\mathcal{N} = 2$ theory reduced on $S^1$ and the HB of the 5d $\mathcal{N} = 2$ SYM reduced on the Riemann surface $\mathcal{C}$.

the Higgs field for the Hitchin system. It is a holomorphic 1-form in the adjoint representation of SU($N$). The SW curve $\Sigma$ of this system is given by Eq. (4.4). By computing the determinant, we can rewrite the SW curve as

$$x^n = \sum_{i=2}^{n} \phi_i(z) x^{n-i} \,, \tag{4.9}$$

where $\phi_i(z)$ are holomorphic functions of degree $i$ defined on the punctured Riemann surface. For the case of the gauge group SU($N$), $\phi_j$ are polynomials of the gauge invariant combination of the Higgs field, i.e., Tr($\Phi_z^j$). In terms of the Hitchin system, a point-like defect on $\mathcal{C}$ corresponds to a pole of $\Phi_z$ (i.e. a corresponding boundary condition). If we pick local coordinates $z$ such that a puncture is at $z = 0$, the Higgs field behaves locally as

$$\Phi_z = \frac{A}{z} + \dots \,, \tag{4.10}$$

where $A$ is an element of $\mathfrak{su}(N)$ that specifies the nature of the puncture and affects the spectrum of BPS particles obtained by wrapping the 6d self-dual string along a cycle surrounding the pole. Since $\Phi_z$ is not gauge invariant, the defects are characterized by the conjugacy class of $A$, and equivalently (co)adjoint orbits in $\mathfrak{su}(N)$ (see [41] for a review). There are two classes of such orbits, corresponding to the Jordan form of the matrix $A$: the semisimple matrix gives rise to the so called **semisimple** orbits, and the nilpotent matrix to the **nilpotent** orbits. In the first case we are introducing additional scales, the eigenvalues of $A$, which are charged under the $\mathfrak{u}(1)_r$ R-symmetry. These eigenvalues are picked up by the integral of the SW differential around the puncture and encode the mass of the BPS particles in the low-energy EFT. Thus, we refer to the eigenvalues of $A$ as the mass parameters of the puncture, which naturally break conformal invariance. On the other hand, the nilpotent $A$ introduces no physical scale, because any gauge invariant combination built from powers of $A$, which would introduce such a scale, vanishes.[39] Punctures labelled by nilpotent orbits are therefore useful to construct theories which manifestly preserve conformal symmetry.

**Example 4.1:**

Let us illustrate using an example for the $\mathfrak{su}(2)$ gauge theory. In this case we have just one Casimir operator

$$\phi_2 = \frac{1}{2} \text{Tr}\left(\Phi_z^2\right) \,, \tag{4.11}$$

where

$$\Phi_z \sim \frac{\begin{pmatrix} m & 0 \\ 0 & -m \end{pmatrix}}{z} + \text{ regular terms.} \tag{4.12}$$

The corresponding Casimir then takes the form

$$\phi_2(z) = \frac{m^2}{z^2} + \dots \,, \tag{4.13}$$

where $m$ is a *mass parameter*. Such parameter represents the mass of BPS particles appearing integrating the Seiberg-Witten differential around $z = 0$. The subleading terms encode the VEVs of the CB operators.

---

[39]Equivalently, a scale transformation of the nilpotent puncture can be undone by a complexified gauge transformation.

In the massless limit the pole remains first order,

$$\Phi_z \sim \frac{\begin{pmatrix} 0 & 1 \\ 0 & 0 \end{pmatrix}}{z} + \text{ regular terms}, \tag{4.14}$$

and similarly for $\phi_2$.

---

**Exercise 4.1** Show that it is possible to put $\begin{pmatrix} m & 0 \\ 0 & -m \end{pmatrix}$ in the form $\begin{pmatrix} m & a \\ 0 & -m \end{pmatrix}$ with $a \in \mathbb{C}$, using conjugation with a matrix in the complexified gauge group $\text{SL}(2,\mathbb{C})$.

---

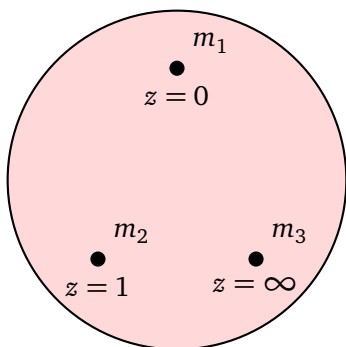

Figure 8: $T_2$ theory.

To be more concrete, let us consider a UV curve $\mathcal{C}$ defined by a sphere $S^2$ with 3 punctures as in Eq. (4.12). This is sometimes called the $T_2$ theory, and it is shown in Figure 8. We have used $m_1$, $m_2$ and $m_3$ to denote the mass parameters for the punctures respectively at $z = 0, 1, \infty$. The Casimir $\phi_2(z)$ can be written as

$$\phi_2(z) = \frac{f(z, m_i)}{z^2(z-1)^2}, \tag{4.15}$$

where $f(z, m_i)$ is a degree 2 polynomial, so that the Hitchin field behaves as in Eq. (4.12) near the punctures at 0, 1 and $\infty$. The Coulomb branch of this theory is trivial and the EFT does not contain any other parameter except for the masses $m_i$ and it is for this reason sometimes called "rigid". The $T_2$ theory, indeed, describes a free $\mathcal{N} = 2$ half-hypermultiplet in the trifundamental representation of $\text{SU}(2)^3$.

Here we focus on the **regular** (or **tame**) punctures for which the Hitchin field has at the leading order a simple pole like in Eq. (4.12) at the position of the puncture. Whenever a puncture is regular, for the $A_N$ type Class $\mathcal{S}$ theory, there is a Young tableau corresponding to the pole structure of each puncture,[40] which encodes a partition of the integer number $N$. Then the global symmetries can be identified directly from said Young tableau.

Consider the SW curve in Eq. (4.9). We can define the **pole structure** of the puncture using a set of positive integers $\{p_j\} = \{p_2, \ldots, p_N\}$. They are the order of the pole that $\phi_j$ can admit at the puncture. From the structure of the poles we find the flavor symmetry group associated to the puncture as follows [33, 36].

First, for the $A$-type puncture, we associate a Young diagram to the pole structure $\{p_k\}$:

---

[40]In this situation the puncture is called "regular" also in the notation introduced in [36]. Let us stress that the two notions of regular punctures do not coincide and in this note we refer to "regular" punctures as those for which the Hitchin field in that point behaves like in Eq. (4.12).

1. Start with a Young diagram with two boxes in a row.

2. For each $k = 3, \ldots, N$:

    - If $p_k - p_{k-1} = 1$, add a box to the current row.

    - If $p_k - p_{k-1} = 0$, start a new row below, with one box.

Then, the global flavor symmetry group is

$$G = S\left(\prod_h U(n_h)\right), \tag{4.16}$$

where $n_h$ is the number of columns of height $h$ in the Young tableaux associated to the puncture.[41]

**Example 4.2:**

As an example, consider $\{p_k\} = \{1, 2, 3, 3, 4, 5, 5\}$. Then we start with a Young diagram as $\square\square$. Since, $p_3 - p_2 = 1$, we add a box on the same row. The same is done for $p_4 - p_3 = 1$, but for $p_5 - p_4$, we start a new row. Completing the procedure, we end up with the following Young tableaux: $\begin{array}{|c|c|c|c|}\hline 0&1&2&3\\\hline 3&4&5\\\cline{1-3} 5\\\cline{1-1}\end{array}$ . There is 1 column of height three, 2 columns of height two, and 1 column of height one. Therefore the global symmetry is $S(U(1) \times U(2) \times U(1))$.

The above procedure can also be reversed, to reconstruct the pole structure $\{p_k\}$ of the differentials $\phi_k$ from a Young diagram (see for example [36]):

1. Label the $N$ boxes of the Young diagram with integers, starting from the longest row, and assigning to its leftmost box the label 0.

2. Increase the label by one as we move to the right along a row.

3. Move on the row below it, and assign to its leftmost box the same label as the rightmost box in the previous row. Repeat the labeling procedure on this row.

4. The resulting sequence of $N$ labels are $\{p_1, \ldots p_N\}$, with $p_1 = 0$ and $p_2$ necessarily equal to 1.

Moreover, if a regular puncture has $p_k = k - 1$ for all $k$, then it is called **maximal** or **full** puncture, and its Young tableaux is a horizontal row of boxes. An $SU(N)$ Young diagram can have at most $N - 1$ rows, and the puncture associated to it is called **minimal** or **simple** puncture.

**Example 4.3:**

As an example, we can start with the Young tableau of Example 4.2, but without numbers on the boxes: $\begin{array}{cccc}\square&\square&\square&\square\\\square&\square&\square\\\square\end{array}$ . We label the Young tableau using the prescription just given. The result is (obviously) the following Young tableaux: $\begin{array}{|c|c|c|c|}\hline 0&1&2&3\\\hline 3&4&5\\\cline{1-3} 5\\\cline{1-1}\end{array}$ . The pole structure is $\{p_k\} = \{1, 2, 3, 3, 4, 5, 5\}$. This can be related to the explicit form of the Hitchin field in Eq. (4.4). The rows and the columns of the Young tableaux $\begin{array}{cccc}\square&\square&\square&\square\\\square&\square&\square\\\square\end{array}$ are respectively

---

[41]There can be enhancements of the global flavor symmetry. In order to find the correct global symmetry group it is helpful, for instance, to compute the Superconformal Index or the Hilbert Series of the theory.

$s_i = \{4, 3, 1\}$ and $t_i = \{3, 2, 2, 1\}$.

When the hypermultiplets are massless, the Hitchin field $\Phi_z$ at the puncture has a residue of the form (see also [62])

$$\operatorname{Res}\Phi_z \sim J_{s_1} \oplus J_{s_2} \oplus J_{s_3} = \begin{pmatrix} J_{s_1} & 0 & 0 \\ 0 & J_{s_2} & 0 \\ 0 & 0 & J_{s_3} \end{pmatrix}, \tag{4.17}$$

where

$$J_4 = \begin{pmatrix} 0 & 1 & 0 & 0 \\ 0 & 0 & 1 & 0 \\ 0 & 0 & 0 & 1 \\ 0 & 0 & 0 & 0 \end{pmatrix}, \quad J_3 = \begin{pmatrix} 0 & 1 & 0 \\ 0 & 0 & 1 \\ 0 & 0 & 0 \end{pmatrix}, \quad J_1 = 0, \tag{4.18}$$

and in general $J_s$ is an $s \times s$ Jordan block matrix. It is then easy to show that by plugging Eq. (4.17) into Eq. (4.4) and comparing with Eq. (4.9), the order of the pole for $\phi_k(z)$ is exactly given by $\{p_k\}$.

In the case in which the hypermultiplets are massive, the matrix-valued 1-form should have a residue that is conjugate to a matrix

$$\operatorname{diag}\left(m_1, m_1, m_1, m_2, m_2, m_3, m_3, m_4\right), \tag{4.19}$$

subject to the traceless condition. More generally, for a Young tableau with column heights $\{t_i\}$, the residue of the Hitchin field is conjugate to the diagonal matrix

$$\operatorname{diag}\big(\underbrace{m_1, \ldots, m_1}_{t_1}, \underbrace{m_2, \ldots, m_2}_{t_2}, \ldots\big), \tag{4.20}$$

such that

$$\sum_i t_i m_i = 0. \tag{4.21}$$

Such residues can be associated to the mass parameters for the flavor symmetry in Eq. (4.16). In the example at hand, the flavor symmetry is

$$\mathrm{S}\left(\mathrm{U}(1) \times \mathrm{U}(2) \times \mathrm{U}(1)\right). \tag{4.22}$$

There are then one mass parameter $m_1$ associated to the first U(1), two mass parameters $m_2$ and $m_3$ for U(2) and another mass parameter $m_4$ for the second U(1), with one linear relation between them, in agreement with (4.19). The generalization to an arbitrary puncture is straightforward.

---

**Exercise 4.2** Prove that plugging Eq. (4.17) into Eq. (4.4) for Example 4.3 and comparing with Eq. (4.9), the order of the pole for $\phi_k(z)$ is exactly given by $\{p_k\}$.

---

As usual, it is instructive to think in terms of string theory if we want to gain some intuition about the punctures and their relation to the global symmetry of the low-energy theory. As we explained in Section 4.3, the punctures originate from M-theory as defect M5-branes transverse to the Gaiotto curve at the special points $p_i$. These branes become semi-infinite D4- or NS5-branes after the reduction to type IIA string theory [28]. A stack of $N$ D4-branes has a low-energy SU($N$) gauge theory living on them. In fact, the gauge group would be U($N$), except a U(1) factor, corresponding to the movement as a whole of the stack of D4-branes on the vertical direction along the NS5-brane, at the same time on its left and on its right, is decoupled from the spectrum (this point will become relevant presently). The D4-branes being semi-

infinite make the kinetic term for these gauge fields vanish in comparison to the similar term corresponding to finite D4-branes hanging between two NS5-branes. Consequently, a stack of $N$ semi-infinite D4-branes will give rise to an SU($N$) flavor symmetry in the low-energy theory. The type IIA NS5-branes do not have such a low-energy gauge theory description, as the worldvolume dynamics of a single NS5-brane is governed by the two-form $B_{\mu\nu}$ and its supersymmetric partners, and having several of them coincide generates a 6d theory on the origin of the tensor branch featuring tensionless strings (which in the low energy limit defines an $\mathcal{N} = (2,0)$ SCFT). Still, after the compactification to 4 dimensions, the reduction of the $B$ field leads to an emergent U(1) gauge field for each five-brane. In our situation, where the NS5-branes are infinite but not coincident, this means we will have a U(1) global symmetry for each one of them. All in all, we conclude that both the full and the simple punctures of Class $\mathcal{S}$ have simple interpretations in the type IIA brane diagram, as the semi-infinite D4-branes at the left and right, and the vertical NS5-branes respectively.

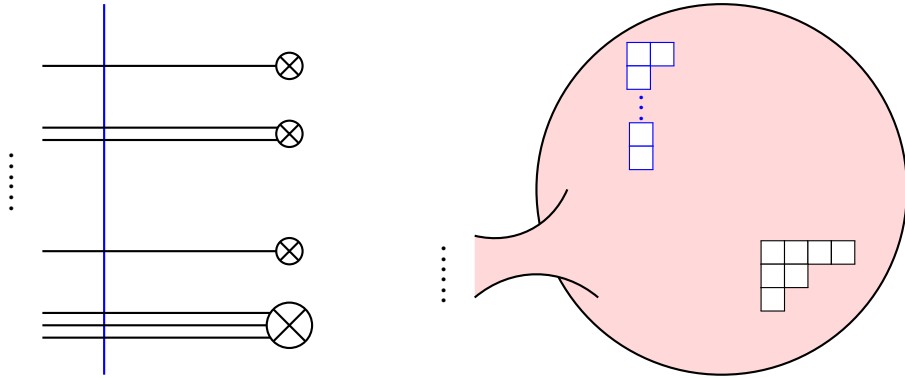

Figure 9: Dictionary between the regular punctures in Class $\mathcal{S}$ and the type IIA brane diagram. On the left, we show the rightmost part of a HW setup with D4-NS5-D6-branes. On the right, we show the corresponding punctures on the Riemann surface. The NS5-brane and its corresponding minimal puncture are drawn in blue.

Also other regular punctures, associated with more complicated Young diagrams, can be easily understood in type IIA, provided we add D6-branes to the picture. The way to do this is, for a given puncture, first add as many D6-branes as there are columns in the Young tableau labeling the puncture, and then make the D4-branes end on the D6-branes, distributing them as indicated by the number of boxes on each column (see Figure 9 for an example). Now that we have added D6-branes, the way to read the global symmetry of the HW setup is different than in the previous case with just semi-infinite D4-branes. This is because a D4-brane hanging between an NS5-brane and a D6-brane has no low-energy gauge theory degrees of freedom; in fact, all its massless degrees of freedom are frozen by the boundary conditions imposed by the NS5 and the D6 brane. Instead, the fields charged under the global symmetry come from fundamental strings hanging between the D4- and D6-branes, and background gauge fields for these global symmetries come from the worldvolume of the D6-branes. It is convenient to make a number of Hanany-Witten moves to bring the D6-branes to the interior of the brane diagram and make the 'frozen' D4-branes disappear. In the example of Figure 9, we have two D6-branes with one D4-brane each: we can bring them into the first interval between NS5-branes and annihilate these two D4-branes; the two coincident D6-branes produce a U(2) global symmetry factor. Likewise, for the D6-brane in which two (resp. three) D4-branes end, we will need to make two (resp. three) Hanany-Witten moves bringing it to the second (resp. third) interval between NS5-branes: in either case, they lead to U(1) global symmetry factors. Note that this is precisely the global symmetry that we obtain in (4.16), once we remove the overall U(1) factor as explained in the previous paragraph.

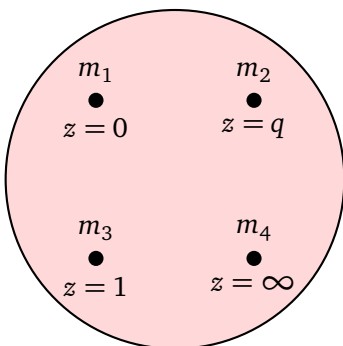

Figure 10: Riemann sphere with 4 punctures.

Let us now go back to 6 dimensions, and consider more interesting theories that include more parameters besides the mass of each puncture. One way to do it is to add one extra puncture to the sphere: holomorphic coordinate redefinitions on $\mathcal{C}$ allows us to fix the position of only three of the punctures to $z = 0, 1, \infty$. The position of the fourth one given by $q$ parametrizes the complex structure of the punctured sphere, and is a new parameter associated to which interesting physics can happen. In particular $q$ is dimensionless and defines an exactly marginal parameter of the 4d theory. More generally, the complex structure moduli of the UV curve $\mathcal{C}$ define a conformal (sub)manifold for the resulting 4d $\mathcal{N} = 2$ SCFT in the Class $\mathcal{S}$ construction.

To be more concrete, let us consider the $\mathfrak{su}(2)$ 6d $\mathcal{N} = (2,0)$ theory compactified on the sphere in Figure 10, where each of the 4 punctures is a pole of the Hitchin field as in equation (4.12).

---

**Exercise 4.3** Compute the Seiberg-Witten curve for the theory in Figure 10, using the same procedure described for the theory of Figure 8. The final result will be given in Example 4.4 in the next section.

---

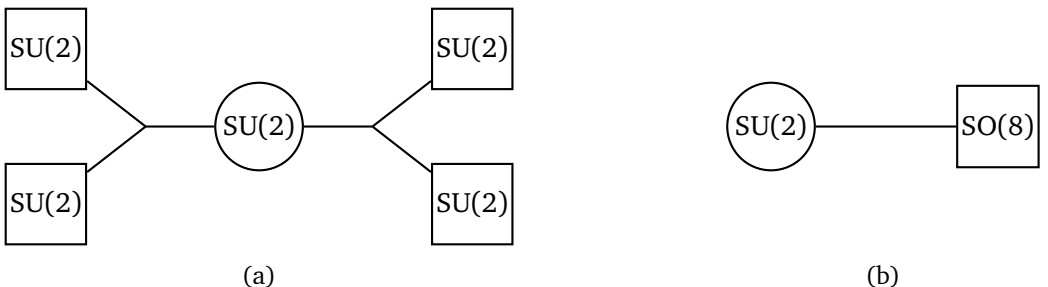

Figure 11: The SCFT described by $\mathcal{N} = 2$ SU(2) SQCD with four fundamental flavors. Each SU(2) flavor node in the left diagram corresponds to one puncture in Figure 10. The right diagram makes manifest the full SO(8) flavor symmetry of the SCFT.

The Seiberg-Witten curve will correspond to that of a quiver diagram of SU(2) with 4 flavors, as shown in Figure 11. The reason why we give two quivers for the same SCFT is related to the complex structure moduli $q$ in Figure 10, and in general different cusp limits of the complex structure moduli of the UV curve produce different quiver representation of the same SCFT from the Class $\mathcal{S}$ construction.[42] In Figure 11b we are showing the usual quiver for

---

[42]Note that the Class $\mathcal{S}$ construction using the 6d $A_1$ theory with the UV curve given in Figure 10 only makes manifest the SU(2)$^4$ subgroup of the SO(8) flavor symmetry. There is an alternative Class $\mathcal{S}$ construction using the 6d $D_4$ theory compactified on a sphere with one regular full puncture and an irregular puncture (see Section 4.8.2)

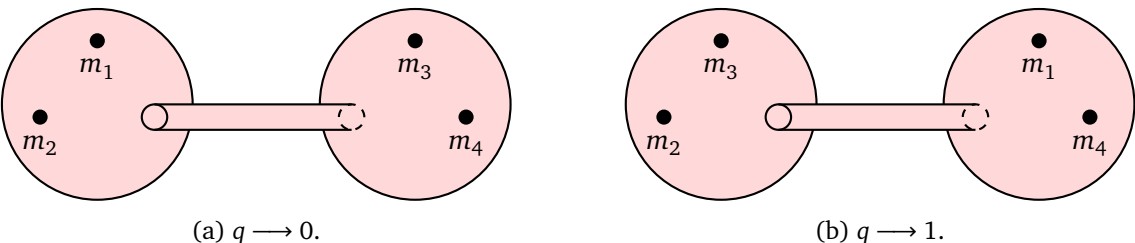

(a) $q \longrightarrow 0$.             (b) $q \longrightarrow 1$.

Figure 12: Factorization of the Riemann sphere with 4 punctures.

SU(2) gauge theory with 4 fundamental flavor hypermultiplets. However, different values of $q$ on the Riemann sphere may give different factorization of the Riemann surface. In Figure 12a we show the factorization of the sphere in the limit when $q \to 0$. As we have seen before, each puncture of the sphere represents an SU(2) hypermultiplet, and the tube joining a pair of punctures from the two spheres represents an SU(2) vector multiplet, and $q$ is related to the complexified gauge coupling as $q = e^{2\pi i \tau}$ (thus $q \to 0$ corresponds to the weak coupling limit of the conformal gauge theory). This theory then corresponds to the quiver in Figure 11a. If we instead tune $q \to 1$ (such that the original gauge theory becomes strongly coupled), we get the factorization in Figure 12b, which is another gauge theory description, S-dual to the previous one. At the level of the quivers, the duality exchanges the mass parameter corresponding to different SU(2) subgroups of the SO(8) global symmetry. Note that this duality, which seems somewhat obvious from the 6d perspective, is highly non-trivial from the point of view of the 4d quiver in Figure 11b. We are saying that when going to the strong coupling limit of the SU(2) SQCD with four fundamental flavors, the theory looks again as a weakly coupled version of the same SQCD up to swapping some masses. This phenomena can then be generalized to other theories in Class $\mathcal{S}$, by studying cusps in the complex structure moduli space of the punctured Riemann surface $\mathcal{C}$ which will produce dualities in the resulting 4d theories [33].

## 4.6 Examples of $\mathcal{N} = 2$ SCFTs in Class $\mathcal{S}[A_n]$

We now give a number of selected examples of $\mathcal{N} = 2$ SCFTs in Class $\mathcal{S}$ for illustration.

**Example 4.4:**

  As a first example, we consider (possibly) the simplest SCFT in 4d, SU(2) SQCD with four fundamental flavors. This is the prototypical example for Class $\mathcal{S}$ theories, and can also be found in other reviews and textbooks e.g. [62, 64]. An industrious reader who has completed Exercise 4.3 may jump ahead to the next example.

  The starting point is the $\mathcal{N} = (2, 0)$ $\mathfrak{su}(2)$ SCFT in 6d, compactified on a sphere with four punctures as shown in Figure 10. To each of these punctures we associate mass parameters $m_1, \ldots, m_4$. This means that locally at each puncture the Hitchin field looks like

$$\Phi_z \sim \frac{\begin{pmatrix} m_i & 0 \\ 0 & -m_i \end{pmatrix}}{z} + \text{ regular terms.} \tag{4.23}$$

  The fact that we are at rank one makes our life easy: expanding the determinant (4.4) into the form (4.9) leads to

$$x^2 = \phi_2(z). \tag{4.24}$$

---

that makes the full SO(8) symmetry transparent.

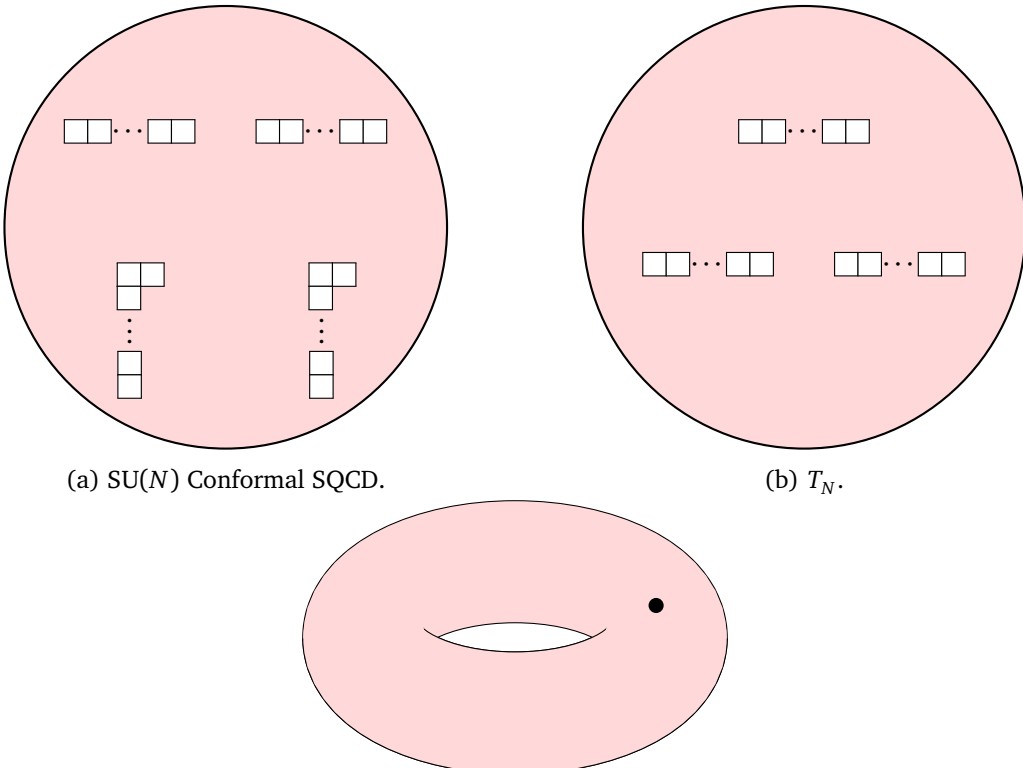

(a) SU($N$) Conformal SQCD.

(b) $T_N$.

(c) $A_1$ $\mathcal{N} = (2,0)$ theory on a torus with a regular puncture.

Figure 13: Examples of Riemann surfaces and shapes of the punctures.

Imposing the condition (4.23) at each puncture $z = 0, 1, q, \infty$ (in order to study the last puncture one should go to the other patch of the sphere $z \to 1/w$) leads to a one parameter family of curves,

$$x^2 = \frac{1}{z(z-1)(z-q)}\left(\frac{q}{z}m_1^2 + \frac{q(q-1)}{z-1}m_2^2 + \frac{z-q}{z-1}m_3^2 + zm_4^2 - u\right). \qquad (4.25)$$

This is the Seiberg-Witten curve for the conformal SU(2) SQCD, with $u \in \mathbb{C}$ the usual parameter of the Coulomb branch.

**Example 4.5:**

The second example is shown in Figure 13a. This is SU($N$) conformal SQCD in terms of the Class $\mathcal{S}$ description: the Riemann surface $\mathcal{C}$ is a sphere with 4 punctures. The two upper punctures are maximal punctures (i.e., the Young tableau has just one row), and they carry the full SU($N$) flavor symmetries, while the two lower punctures are minimal (i.e., the Young tableau has one column with $N-1$ boxes and one with 1 box), and the flavor symmetry is just U(1).

**Example 4.6:**

More generally, many SCFTs from the Class $\mathcal{S}$ construction do not have a Lagrangian description. An example is given by the $T_N$ theories in Figure 13b. The Riemann surface is again a sphere but this time we put three maximal punctures. There is no Lagrangian description[43] for this interacting SCFT, but it has manifestly SU($N$) × SU($N$) × SU($N$) global symmetry. A necessary condition to have an $\mathcal{N} = 2$ conformal Lagrangian description is

to have some complex structure moduli on the Riemann surface that we can tune (which would correspond to the complexified gauge coupling). This is the case of a sphere with 4 punctures as in Figure 10, but it is not the case for the $T_N$ theories with just 3 punctures.

**Example 4.7:**

The Riemann surface $\mathcal{C}$ can in general be higher genus. For example, when $\mathcal{C}$ is a once-punctured torus as in Figure 13c, the 4d theory realized by $A_1$ $\mathcal{N} = (2,0)$ on such torus with a regular minimal puncture corresponds to the $\mathcal{N} = 4$ SU(2) SYM (whose $\mathcal{N} = 2$ preserving mass deformed theory is also known as the $\mathcal{N} = 2^*$ theory). The structure of the pole is given by Eq. (4.12).

Moreover, the same 4d SCFT may have multiple Class $\mathcal{S}$ constructions where the corresponding UV curves differ. Coming back to the $\mathcal{N} = 4$ SU(2) SYM example again, as we will see in more detail in Section 4.8.3, there is an alternative Class $\mathcal{S}$ construction in terms of the $A_2$ $\mathcal{N} = (2,0)$ theory on a sphere with one twisted regular and one twisted irregular puncture (as in Figure 15). This second construction makes manifest the Witten's anomaly [212] for the SU(2) symmetry of the $\mathcal{N} = 4$ SYM [56, 213].

In general, multiple Class $\mathcal{S}$ constructions can provide complementary descriptions to the same SCFT, and in particular, the constructions using irregular punctures are useful in making explicit the symmetry enhancement and details of the symmetries such as their 't Hooft anomalies. We will introduce properly irregular punctures in Section 4.8.2 and their twisted versions in Section 4.8.3.

## 4.7 A Glimpse of the AGT Correspondence

The Alday-Gaiotto-Tachikawa (AGT) correspondence is a conjecture introduced in [214] that relates the Nekrasov partition function [76, 77] (see also [215, 216]) of the 4d $\mathcal{T}(\mathfrak{g}, \mathcal{C}, p)$ Class $\mathcal{S}$ theory obtained from compactification of a 6d $\mathcal{N} = (2,0)$ theory of type $\mathfrak{g}$, with the $\mathfrak{g}$-Toda theory conformal blocks [217]. Below we focus on the $A_1$ Class $\mathcal{S}$ construction in which case the Toda CFT is the well-known Liouville theory (see [218–220] for recent reviews on this subject).

To be more precise the $\mathcal{T}(A_1, \mathcal{C}, p)$ theory is placed on a squashed sphere $S_b^4$ of size $R$ with squashing parameter $b$ defined by

$$b^2(x_1^2 + x_2^2) + \frac{1}{b^2}(x_3^2 + x_4^2) + x_5^2 = R^2 \,, \tag{4.26}$$

with $x_i \in \mathbb{R}$. The supersymmetric partition function on $S_b^4$ can be evaluated explicitly by the localization method [78, 221] (see also in [79, 222] for exhaustive reviews), whenever an $\mathcal{N} = 2$ Lagrangian description for the 4d theory is available. The supersymmetric localization reduces the path integral to a finite dimensional matrix model, which yields a ordinary integral over the real parts of the vector multiplet scalars, weighted by the exponentiated classical action and the one-loop determinants from the various fields in the theory, and furthermore dressed by contributions from point-like instantons at the two poles of $S_b^4$. The full partition function takes the form

$$Z_{S_b^4}(q, \bar{q}) = \int da Z_{\text{cl}}(a, q, \bar{q}) Z_{1\text{-loop}}(a) Z_{\text{inst}}(a, q) Z_{\text{inst}}(a, \bar{q}) \,. \tag{4.27}$$

---

[43]We mean that there is no Lagrangian that can be written in $\mathcal{N} = 2$ language. It is possible that there exist $\mathcal{N} = 1$ Lagrangians that flow to an interacting $\mathcal{N} = 2$ theory in 4d [133]. Supersymmetry is broken explicitly along the flow, but restored in the IR. There are many examples of these flows also applied to $(G, G')$ theories (that we introduce in Section 5.2), see for instance [134, 202–211].

The contribution $Z_{cl}$ is weighting the localizing configurations by the classical action, while the one loop fluctuations are accounted by $Z_{1\text{-loop}}$. Finally, $Z_{inst}$ denotes the Nekrasov instanton partition function.[44]

Such a partition function is conjectured to be equal to the correlator of certain vertex operators in the Liouville CFT (or its generalization Toda CFT in the higher rank case). Such vertex operators are inserted at each puncture $z_i$ and depend on the puncture types in the Class $\mathcal{S}$ construction,

$$Z_{S_b^4}(\mathcal{T}(A_1, \mathcal{C}, p)) = \langle V_{p_1}(z_1) \dots V_{p_n}(z_n) \rangle_{\overline{\mathcal{C}}}^{\text{Liouville}}, \tag{4.28}$$

where $\overline{\mathcal{C}}$ is the compact Riemann surface where the points $z_1, \dots, z_n$ have been removed. We are not going to enter into the details of this correspondence. Instead we list the dictionary between observables in the Liouville CFT and rank-one SCFTs from Class $\mathcal{S}$ construction in Table 4 from [214]. A quick review of the relevant ingredients in the Liouville CFT can be found in [218]. For details and extensions of the AGT correspondence, we also refer to the extensive review [64] and its detailed list of references.

## 4.8 Further Generalizations

Before wrapping up our discussion of Class $\mathcal{S}$ constructions, we proceed to briefly mention several further ingredients that we have at our disposal to construct 4d $\mathcal{N} = 2$ SCFTs and which we have not yet covered, namely different choices of the $\mathcal{N} = (2,0)$ theory in 6d, different singular behaviors of the Hitchin field near each puncture, and the possibility of including a monodromy twist when going around a puncture.

### 4.8.1 General $\mathcal{N} = (2,0)$ Theory

In the previous sections, we have mainly focused on the Class $\mathcal{S}$ constructions that use a stack of $N$ M5-branes, which describes the 6d $\mathcal{N} = (2,0)$ SCFT of the $A_{N-1}$ type. There are other kinds of 6d $\mathcal{N} = (2,0)$ theories, and they are of the types $D_N$, $E_6$, $E_7$ and $E_8$. The $D_N$ type theories can be constructed in M-theory from a stack of $N$ M5-branes on top of an M-theory orientifold OM5 [223–225]. More generally, they can be engineered by type IIB string theory probing ADE singularities as reviewed in Section 4.1.

Recall in the $A_{N-1}$ type Class $\mathcal{S}$ construction, important 4d Coulomb branch physics (e.g. spectrum of chiral primaries, chiral couplings and flavor symmetries) is contained in the $\mathfrak{su}(N)$ invariant differentials $\phi_k$ for $k = 2, \dots, N$ that are made out of the Higgs field $\Phi_z$. At the abstract level, these differentials correspond to half-BPS operators in the 6d $\mathcal{N} = (2,0)$ SCFT [5] and for general ADE type, they are in one-to-one correspondence with the Casimirs for the Lie algebra $\mathfrak{g}$. For the $D_N$ type Class $\mathcal{S}$ construction, the independent differentials are $\phi_k$ with $k = 2, 4, \dots, 2N-2$ and $\widetilde{\phi}_N$ corresponding to the Pfaffian invariant of $\mathfrak{so}(2N)$. For $E_6$, the differentials are $\phi_k$ with $k = 2, 5, 6, 8, 9, 12$. For $E_7$, it is $k = 2, 6, 8, 10, 12, 14, 18$, and for $E_8$, $k = 2, 8, 12, 14, 18, 20, 24, 30$.

A large class of 4d $\mathcal{N} = 2$ theories can be constructed from a 6d $\mathcal{N} = (2,0)$ SCFT labelled by an ADE Lie algebra, twist compactified on a Gaiotto curve $\mathcal{C}$ with a collection of punctures and focusing onto the low-energy limit. For the 4d theory to be an SCFT, there are the following options for $\mathcal{C}$,

---

[44]General rank-one 4d $\mathcal{N} = 2$ SCFTs are not Lagrangian so there maybe no known localization formula for the sphere partition function. This happens for the Argyres-Douglas theories constructed by irregular punctures. Nonetheless we still expect its partition function to have the form of an integral over a real slice of its Coulomb branch as in (4.27) where the building blocks have natural interpretations in the Liouville (and also Toda) CFT as in Table 4 (see for example [130]).

Table 4: Dictionary between the 2d and 4d observables in the rank-one AGT correspondence [214].

| $\mathcal{N}=2$ **theories** $\mathcal{T}(A_1,\mathcal{C},p)$ | **Liouville theory on** $\overline{\mathcal{C}}$ |
| --- | --- |
| Deformation parameters $\epsilon_1$ and $\epsilon_2$ | Liouville parameters $\epsilon_1 = b$ and $\epsilon_2 = b^{-1}$ $c = 1 + 6Q^2$ and $Q = b + b^{-1}$ |
| Four free hypermultiplets | 3-punctured sphere |
| Mass parameter $m$ associated to an SU(2) flavor | Insertion of a Liouville vertex operator $e^{2m\phi}$ |
| SU(2) gauge group with UV coupling $\tau$ | OPE channel with gluing parameter $q = e^{2\pi i \tau}$ |
| VEV $a$ of an SU(2) gauge group | Primary $e^{2\alpha\phi}$ for the intermediate channel, $\alpha = Q/2 + a$ |
| $Z_{\text{inst}}(a,Q)$ | Virasoro conformal blocks |
| $Z_{\text{1-loop}}(a)$ | Product of OPE coefficients (given by the DOZZ formula) |
| $Z_{S^4_b}(\mathcal{T}(A_1,\mathcal{C},p))$ | Liouville correlator |

- A sphere with three or more regular punctures.[45]

- A sphere with one regular puncture and one irregular punctures.[46]

- A sphere with one irregular puncture.

- Higher-genus $\mathcal{C}$ with regular punctures.

---

[45]The SCFTs constructed from a sphere with three regular punctures are called **tinkertoys** and have been studied extensively in [36, 37, 41, 44–46, 49, 53, 226].

[46]The irregular punctures will be introduced in Section 4.8.2.

### 4.8.2 Irregular (wild) Punctures

The original Class $\mathcal{S}$ construction in [33] is vastly generalized by including more general punctures known as **irregular** (**wild**) singularities. In fact, they are crucial in providing Class $\mathcal{S}$ constructions of Argyres-Douglas type theories which are otherwise impossible with just regular punctures.[47]

While the Hitchin field has simple poles for regular punctures, there are also punctures for which the Hitchin field has a stronger singularity than that in Eq. (4.12).[48] Moreover, the Hitchin field can also have poles with certain fractional orders that are compatible with the structure group $G$ [35]. As a consequence, the orders of the poles for the $\phi_k$ differentials can be larger than $k$. We call these punctures **irregular** (or **wild**) and they have been studied in [35, 40, 42, 47, 56, 228, 229].[49] While regular punctures have a simple classification by nilpotent orbits for the Lie algebra $\mathfrak{g}$ labelling the 6d $\mathcal{N} = (2,0)$ SCFT (e.g. by Young tableaux for classical Lie algebras and by Bala-Carter labels more generally [41]), the classification of irregular punctures is much richer and there is a systematic way to read-off the physical information such as (Cartan generators of) flavor symmetries, exactly marginal couplings, and CB chiral primaries from the Hitchin pole (see for example [42, 47]).

> **Example 4.8:**
>
> The simplest example of AD SCFTs that can be realized in Class $\mathcal{S}$ is shown in Figure 14, usually denoted $(A_1, A_2)$. The theory is rank 1 with a Coulomb Branch operator of fractional dimension $\Delta[u] = \frac{6}{5}$. Its Class $\mathcal{S}$ construction involves the $A_1$ $\mathcal{N} = (2,0)$ theory on $\mathcal{C}$ which is a sphere with a single irregular puncture of the type,
>
> $$\Phi_z \sim \frac{\begin{pmatrix} 1 & 0 \\ 0 & -1 \end{pmatrix}}{z^{2+3/2}} dz + \dots . \tag{4.29}$$
>
> Using the same prescription introduced before, the singular (scaling-symmetric) SW curve is given by, after a coordinate transformation,
>
> $$x^2 + z^3 = 0, \tag{4.30}$$
>
> with SW differential $\lambda = x\,dz$, from which we read-off the scaling dimensions $\Delta[z] = \frac{2}{5}$ and $\Delta[x] = \frac{3}{5}$. The normalization is such that $\Delta[\lambda] = 1$ since its integrals along cycles on the SW curve give CB central charges. The full SW curve with deformations is
>
> $$x^2 + \phi_2(z) = 0, \tag{4.31}$$
>
> with Casimir $\phi_2(z) = z^3 + \tau_u z + u$ from which we recover the Coulomb branch operator in the $(A_1, A_2)$ SCFT with $\Delta[u] = \frac{6}{5}$ and $\tau_u$ is the corresponding chiral coupling.

In most cases of Class $\mathcal{S}$ constructions with irregular punctures, the U(1)$_r$ symmetry of the superconformal algebra is emergent. These general punctures give UV definitions of codimension-2 defects in the $\mathcal{N} = (2,0)$ theory that flow to superconformal defects in the IR preserving the 4d $\mathcal{N} = 2$ superconformal symmetry. They are used to realizing AD type

---

[47]A characteristic of the Class $\mathcal{S}$ construction with untwisted regular punctures is that the resulting SCFT has a CB chiral ring whose spectrum of scaling dimensions are entirely integral. This feature excludes most Argyres-Douglas type theories which typically have fractional dimensions in the CB operator spectrum. However, Class $\mathcal{S}$ constructions with twisted regular punctures may give rise to CB operators with fractional dimensions [227].

[48]They define more general codimension-2 defects in the 6d $\mathcal{N} = (2,0)$ SCFT.

[49]Remember that the notation of "irregular" puncture that we are using here is associated to the behavior of the Hitchin field and not to the definition introduced [36].

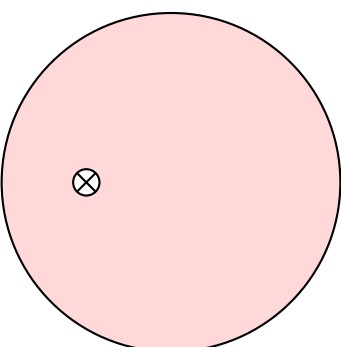

Figure 14: $A_1$ $\mathcal{N} = (2,0)$ theory on a sphere $\mathcal{C}$ with a single irregular puncture.

SCFTs with fractional scaling dimensions for the Coulomb Branch operators in the Class $\mathcal{S}$ set up. Such theories do not admit a Lagrangian description, and at the present they do not yet have a complete classification.

Unlike the Coulomb branch which is encoded in the Hitchin system associated to the Class $\mathcal{S}$ setup, less obvious is the Higgs branch of these AD SCFTs. One useful way to proceed is to consider the 3d $\mathcal{N} = 4$ SCFT from the circle compactification of the Class $\mathcal{S}$ theory as in Figure 7. Because of the non-renormalization property of the Higgs branch, the 4d Higgs branch is identical to the one in the 3d limit [201]. Although the general 3d $\mathcal{N} = 4$ theories that arise this way are strongly coupled and do not have known weakly coupled descriptions, a large class of them do have UV descriptions by $\mathcal{N} = 4$ quiver gauge theories in the mirror frame [199], and thus commonly referred to as the "3d mirrors" for the corresponding 4d SCFTs. For Class $\mathcal{S}$ constructions with regular punctures, the 3d mirrors were identified in [230, 231], and the generalizations to cases with irregular punctures can be found in [42, 232–240]. Whenever a 3d mirror Lagrangian description is available, we not only obtain the 4d Coulomb branch from the Hitchin base of the Higgs branch for the 3d mirror as previously explained, we also recover the 4d Higgs branch from the 3d (quantum) Coulomb branch. The latter crucially receives contributions from not only the 3d Cartan vector multiplet scalars but also the monopole operators [241–246], which play an important role in understanding symmetry enhancement on the Coulomb branch of the 3d mirror and consequently the corresponding 4d Higgs branch.

Another complementary approach to understanding the Higgs branch of an AD SCFT is through identifying its chiral algebra sector [86] and the associated variety [247] which is conjectured to coincide with the 4d Higgs branch [248]. The chiral algebra sector is a subset of the local operators in the 4d SCFT that closes under OPE restricted to a 2d plane (known as the chiral algebra plane) [8]. A lot of progress has been made to identify the chiral algebra sectors of AD SCFTs and the corresponding associated varieties [73, 74, 232, 248–258] including for Class $\mathcal{S}$ theories with twisted irregular punctures [56], which we will come to shortly. For AD SCFTs where both the 3d mirror and the chiral algebra (and its associated variety) are available, the two approaches discussed here have been demonstrated to produce the same Higgs branch, providing nontrivial evidence for the conjectured relation between associated varieties of 2d chiral algebras and 4d Higgs branches (see for example [232, 248]).

### 4.8.3 Outer-Automorphism Twist

The last generalization we will briefly discuss involves decorating the UV curve $\mathcal{C}$ with twist lines connecting twisted punctures. So far we have focused on non-twisted defects, which are genuine half-BPS codimension-2 defects in the 6d $(2,0)$ SCFT labelled by ADE Lie algebra $\mathfrak{g}$. For regular punctures, such a defect is defined by a specific embedding in $\mathfrak{su}(2)$ on $\mathfrak{g}$ up to conjugacy which specifies the Hitchin pole for the Higgs field $\Phi_z$ [41] and suitably generalized

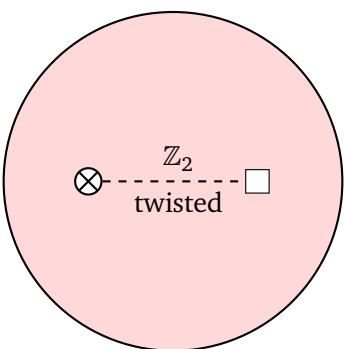

Figure 15: Class $\mathcal{S}$ construction of SCFTs by a pair of twisted regular and irregular punctures in the 6d $\mathcal{N} = (2,0)$ SCFT. In the text we specifically consider the $\mathbb{Z}_2$ twist in the $A_2$ $\mathcal{N} = (2,0)$ SCFT that engineers the $\mathcal{N} = 4$ SU(2) SYM.

for irregular punctures [47]. The 6d SCFT has a discrete global ordinary (0-form) symmetry given by the outer-automorphism group $\text{Out}(\mathfrak{g})$,

$$\text{Out}(A_N) = \text{Out}(D_{N \neq 4}) = \text{Out}(E_6) = \mathbb{Z}_2, \quad \text{Out}(D_4) = S_3. \tag{4.32}$$

It is a general fact that in any QFT with a 0-form symmetry $G$, one can define monodromy or twist codimension-2 defects where the charged operators undergo a monodromy transformation by an element $g \in G$ when going around the defect. Projecting to the two dimensions transverse to the codimension-2 defect, the nontrivial monodromy indicates the existence of a topological line implementing the $g$ transformation that ends at the defect insertion and is simply the projection of the codimension-1 topological symmetry defect labelled by $g$ in the full theory.[50]

Here in the 6d $\mathcal{N} = (2,0)$ SCFT, the codimension-2 **twisted** defect (also called **monodromy defects**) introduces a monodromy by an element of the outer-automorphism group $o \in \text{Out}(\mathfrak{g})$ when we go around the defect [41,47,260,261]. The Hitchin pole for a twisted regular puncture is now specified by the embedding $\rho : \mathfrak{su}(2) \longrightarrow \mathfrak{g}_0$, where $\mathfrak{g}_0 \subset \mathfrak{g}$ is the $o$-invariant subalgebra [41], and a suitable generalization is needed to account for twisted irregular punctures [56]. We collect the possible twists $o$ in Class $\mathcal{S}$ constructions and the corresponding invariant subalgebra $\mathfrak{g}_0$ together with its Langlands dual $\mathfrak{g}_0^\vee$ in Table 5. The Langlands dual $\mathfrak{g}_0^\vee$ accounts for the flavor symmetry of the twisted regular punctures of the maximal (full) type [41].

**Example 4.9:**

Here we illustrate Class $\mathcal{S}$ constructions with twisted irregular punctures by a simple example. In the same example, we will also see how different Class $\mathcal{S}$ constructions of the same SCFT give more insights on the theory itself. We consider half-BPS codimension-2 defects in the $A_2$ $\mathcal{N} = (2,0)$ SCFT twisted by the $\mathbb{Z}_2$ outer-automorphism. The Class $\mathcal{S}$ construction of interest is represented in Figure 15 which involves a pair of twisted regular and irregular punctures (introduced in Section 4.8.2) on the UV curve $\mathcal{C}$ which is a sphere. The irregular puncture is specified by the Hitchin pole

$$\Phi_z \sim \frac{\begin{pmatrix} 0 & 0 & 0 \\ 0 & 1 & 0 \\ 0 & 0 & -1 \end{pmatrix}}{z^{3/2}} + \dots, \tag{4.33}$$

which exhibits a monodromy given by the $\mathbb{Z}_2$ outer-automorphism that acts as a $\mathbb{Z}_2$ per-

---

[50]For a general discussion on (generalized) symmetries as topological defects, see for example [259].

mutation of the bottom-right two-by-two block. The Coulomb branch data (i.e. CB chiral primaries, chiral couplings and mass parameters) of the resulting theory follows from the twisted Hitchin system. The Higgs branch and its flavor symmetry can also be inferred by taking into account the regular twisted puncture. Together, they imply that the resulting theory is the $\mathcal{N} = 4$ SU(2) SYM [56].

Recall as reviewed before, the $\mathcal{N} = 4$ SU(2) SYM has a more familiar Class $\mathcal{S}$ construction by considering the $A_1$ $\mathcal{N} = (2,0)$ on a torus with a single regular puncture as shown in Figure 13c. The two distinct Class $\mathcal{S}$ constructions are complementary. The one with regular puncture on a torus makes manifest the conformal manifold of the SCFT in terms of the complex structure moduli of the once punctured torus. Meanwhile, the construction with twisted irregular and regular punctures makes transparent fine structures of the SU(2) flavor symmetry. In particular, the $\mathbb{Z}_2$ twisted regular full punctures in $A_{2N}$ theories always carry the Witten's global anomaly for the USp(2$N$) flavor symmetry [213], which applies immediately to the $A_2$ case that engineers the $\mathcal{N} = 4$ SYM. Furthermore, twisted regular punctures (paired with an irregular puncture on a sphere) also have a natural correspondence with chiral algebras of the W-algebra type [56], which implies in particular that the chiral algebra corresponding to the $\mathcal{N} = 4$ SU(2) SYM is the $\widehat{\mathfrak{su}}(2)_{-\frac{3}{2}}$ Kac-Moody algebra, a fact easy to verify since the theory is Lagrangian.

Table 5: Relations between the algebra $\mathfrak{g}$, the twist $o$, the invariant subalgebra $\mathfrak{g}_0$ with respect to the action of $o$ and its Langlands dual $\mathfrak{g}_0^\vee$ (see for example [41]).

| $\mathfrak{g}$ | $o$ | $\mathfrak{g}_0$ | $\mathfrak{g}_0^\vee$ | $\mathfrak{g}$ | $o$ | $\mathfrak{g}_0$ | $\mathfrak{g}_0^\vee$ |
|---|---|---|---|---|---|---|---|
| $\mathfrak{a}_{2N-1}$ | $\mathbb{Z}_2$ | $\mathfrak{c}_N$ | $\mathfrak{b}_N$ | $\mathfrak{d}_4$ | $\mathbb{Z}_3$ | $\mathfrak{g}_2$ | $\mathfrak{g}_2$ |
| $\mathfrak{a}_{2N}$ | $\mathbb{Z}_2$ | $\mathfrak{b}_N$ | $\mathfrak{c}_N$ | $\mathfrak{e}_6$ | $\mathbb{Z}_2$ | $\mathfrak{f}_4$ | $\mathfrak{f}_4$ |
| $\mathfrak{d}_N$ | $\mathbb{Z}_2$ | $\mathfrak{b}_{N-1}$ | $\mathfrak{c}_{N-1}$ | | | | |

# 5 Geometric Engineering in IIB String Theory

In the last section, we have introduced the Class $\mathcal{S}$ constructions of 4d $\mathcal{N} = 2$ SCFTs, and we have seen that a SW geometry naturally appears in a Class $\mathcal{S}$ construction from the spectral curve of the Hitchin integrable system. More generally, we have also seen in Section 3 how the Coulomb branch of a generic 4d $\mathcal{N} = 2$ SCFT is described by a special Kähler geometry. Such geometric structures also arise naturally from the moduli space of a Calabi-Yau 3-fold [262], which leads us to ask whether one can directly engineer a 4d $\mathcal{N} = 2$ SCFT from a Calabi-Yau compactification of type II string theories. The main purpose of this section is to try to give an overview of the geometric engineering of 4d $\mathcal{N} = 2$ SCFTs from type IIB compactifications.

To orient ourselves, we first start by reviewing isolated 3-fold hypersurface singularities in Calabi-Yau 3-folds in Section 5.1. Following that, we list some typical examples of 4d $\mathcal{N} = 2$ SCFTs from the so-called $(G, G')$ singularities in Section 5.2. We briefly comment on the relation between this approach and the Class $\mathcal{S}$ construction in Section 5.3. We discuss generalizations of these isolated hypersurface singularities to other types of singularities in Section 5.4. In the end, we switch to F-theory to consider D3-branes probing the singularities of elliptically fibered Calabi-Yau manifolds, and review recent developments on the $\mathcal{N} = 2$ S-fold constructions.

## 5.1 4d $\mathcal{N} = 2$ SCFTs From 3-fold Singularities

In order to generate a 4d field theory limit in a type IIB compactification on a Calabi-Yau manifold, one typically needs to turn off the gravitational interactions by sending the 4d Planck scale $M_P$ to infinity, i.e., $M_P \to \infty$, which is proportional to the volume of the Calabi-Yau manifold $W$ by KK reduction. Thus, one should consider compactifications on non-compact Calabi-Yau manifold. Furthermore, one typically needs Calabi-Yau manifolds with singularities to generate interesting interacting field theories.

For concreteness, let us first consider a CY manifold $W$ with an isolated canonical 3-fold hypersurface singularity (IHS),[51] which is defined by a polynomial $F$ as follows:

$$W := \left\{ F(x_1, x_2, x_3, x_4) = 0 \right\} \subset \mathbb{C}^4, \tag{5.1}$$

where $F = dF = 0$ have a unique solution at the isolated point (without loss of generality, we assume it to be the origin $x_i = 0$). Note that some of these hypersurface singularities, e.g., the $A_k$ singularities in the upcoming Example 5.1 with large $k$ cannot be embedded in a compact Calabi-Yau. This is another reason to take $W$ to be a non-compact CY in the first place. The effective theory of type II string compactified on $W$ typically reduces to a 4d $\mathcal{N} = 2$ gauge field theory.[52] In order to obtain a 4d $\mathcal{N} = 2$ superconformal theory from the type IIB string on the IHS, the necessary and sufficient conditions on $W$ are [31, 268]:

1. The polynomial $F$ must be quasi-homogeneous, indicating that $F$ has a non-trivial $\mathbb{C}^*$ action such that all the coordinates $x_i$ have positive weights. Namely, this condition is

$$\mathbb{C}^* : F(\xi^{q_i} x_i) = \xi F(x_i), \, q_i > 0, \, i = 1, 2, 3, 4. \tag{5.2}$$

   This requirement comes from the fact that the 4d SCFT under consideration has a $U(1)_r$ R-symmetry, which descends from this $\mathbb{C}^*$ action and must be preserved.

2. The weights $q_i(x_i)$ under the $\mathbb{C}^*$ action have to satisfy the following condition to ensure that $W$ has a Calabi-Yau conic metric [31]:

$$\sum_{i=1}^{4} q_i > 1. \tag{5.3}$$

   This second condition can be understood from the perspective of a 2d $\mathcal{N} = (2, 2)$ Landau-Ginzburg (LG) model that describes the type IIB string theory background, where the polynomial $F$ is viewed as its worldsheet superpotential. To see that, one recall that by taking the type IIB string coupling $g_s \to 0$ while keeping the string scale $\ell_s$ fixed, the

---

[51]A singularity of a variety $X$ is called as a **canonical singularity** if it satisfies the following conditions:

- The canonical divisor (as a Weil divisor) $K_X$ is Q-Cartier, i.e. $rK_X$ is a Cartier divisor. Here $r \in \mathbb{Z}$ is known as the index of the singularity.
- For any resolution of singularities $f : Y \to X$, the rational coefficient $a_i$ in the new canonical divisor

$$K_Y = f^* K_X + \sum_i a_i E_i$$

are all non-negative, where $E_i$ denotes exceptional divisors in $Y$.

Especially, if all $a_i$ are zero, then the singularity admits a crepant resolution. If all $a_i$ are positive, such a singularity is known as a terminal singularity, which has been recently studied in SCFTs [236, 263, 264] and F-theory [265]. General background on these singularities and terminologies is provided e.g. in [266].

[52]In certain circumstances, in order to obtain a genuine 4d $\mathcal{N} = 2$ gauge theory, one needs to take further limits to decouple extra degrees of freedoms. For example, Type IIA on such a manifold gives rise to a 4d Kaluza-Klein (KK) theory [267] and one needs to take an additional scaling limit in Kähler moduli space to decouple the tower of KK modes and hence reach a 4d gauge theory, which is known as the geometric engineering limit [27].

decoupled dynamics at the IHS is described by a 4d little string theory (LST) (see e.g., [269]), which is holographically dual to type IIB string theory on the background [270]

$$\mathbb{R}^{1,3} \times \mathbb{R}_\phi \times (S^1 \times LG(F))/\Gamma, \tag{5.4}$$

with $\Gamma$ being a suitable Gliozzi-Scherk-Olive (GSO) orbifold projection in order to preserve the 4d $\mathcal{N} = 2$ spacetime supersymmetry. Here $\mathbb{R}_\phi$ is a linear dilaton direction with the dilaton profile $\Phi := -\frac{Q}{2}\phi$ and $LG(F)$ stands for a 2d $\mathcal{N} = (2,2)$ LG model with four chiral superfields $x_i$, with the holomorphic superpotential $F$. In particular, the above $\mathbb{C}^*$ action is interpreted as the U(1)$_r$ symmetry in the LG model. By further taking the low-energy limit $\ell_s \to 0$, one recovers a 4d $\mathcal{N} = 2$ SCFT from the 4d LST [270, 271]. Now, the $\mathcal{N} = 1$ linear dilaton theory has central charge $\frac{3}{2} + 3Q^2$, whereas the LG model has central charge $3\hat{c} = 3\sum_{i=1}^4 (1 - 2q_i)$, and the remaining $\mathbb{R}^{1,3} \times S^1$ contributes $c = \frac{15}{2}$. Consistency of the type IIB string theory on this background requires the worldsheet theory to have a total central charge of $26 - 11 = 15$, leading to the condition

$$\frac{1}{2}Q^2 = \sum_i q_i - 1 > 0, \tag{5.5}$$

as in (5.3). From the geometric viewpoint, the condition (5.3) imposes that the IHS is at finite distance on the Calabi-Yau moduli space when the IHS can be embedded in a compact Calabi-Yau 3-fold [268].[53]

As introduced in Section 2.3, an $\mathcal{N} = 2$ theory has a rich moduli space of vacua which can be characterized as a Coulomb branch, a Higgs branch, and/or a mixed branch depending on the R-symmetry subgroups that are spontaneously broken. In type IIB compactifications, the first two are related to the complex structure moduli and the Kähler moduli of the CY respectively. The low-energy physics on the Coulomb branch is particularly interesting and is determined by the Seiberg-Witten geometry [20, 21]. One of the most important tasks in studying an $\mathcal{N} = 2$ SCFT is to obtain its SW geometry, which in our cases can be found by identifying mini-versal deformations of the IHS.[54]

The mini-versal deformation of an IHS can be described as follows: given a quasi-homogeneous polynomial $F$ describing an IHS at the origin, the mini-versal deformation of the IHS takes the form:

$$\hat{F} = F(x_i) + \sum_{\alpha=1}^\mu g^\alpha(x_i)\lambda_\alpha. \tag{5.6}$$

Here $g^\alpha := x_1^{\alpha_1} x_2^{\alpha_2} x_3^{\alpha_3} x_4^{\alpha_4}$, with $\alpha = (\alpha_1, \alpha_2, \alpha_3, \alpha_4)$, denotes the monomial basis of the Jacobi algebra (Milnor ring) $J(F)$ defined as

$$J(F) = \mathbb{C}[x_1, x_2, x_3, x_4]/(dF), \tag{5.7}$$

where $\mathbb{C}[x_i]$ denotes the polynomial ring of $\mathbb{C}^4$ and $(dF)$ represents the Jacobi ideal generated by $\partial_i F = 0$ (which eliminates trivial deformations). The dimension of the Jacobi algebra $\mu$ is[55]

---

[53]Generally speaking, the condition for an isolated hypersurface singularity in a Calabi-Yau $d$-fold is at the finite distance in moduli space is $\hat{c} < d - 1$ [268].

[54]The deformation theory of isolated singularities is a well-studied subject in mathematics. We refer the interested readers to the classic references [272, 273] for the precise mathematical definitions and theorems. Here we simply note that, intuitively, versal deformations are deformations of the singularity that cannot be removed by a redefinition of the deformation parameters and minimimal versal or mini-versal deformations are versal deformations where the number of parameters is minimal. Physically, mini-versal deformations account for the independent parameters on the Coulomb branch of the corresponding the 4d $\mathcal{N} = 2$ theory.

[55]To derive this, one can for example construct the Poincaré polynomial of the algebra $J(F)$ and read off $\mu$. We refer to, e.g., [48] and references therein for details. Geometrically, $\mu$ is also known as the Milnor number, which counts the middle dimensional homology cycles of the deformed Calabi-Yau $\hat{W} := \{\hat{F} = 0\}$. From the 2d LG point of view, $\mu$ coincides with the Witten index $\mathrm{Tr}(-1)^F$.

$$\mu = \prod_{i=1}^{4} \left( q_i^{-1} - 1 \right) . \tag{5.8}$$

**Example 5.1:**

We provide some examples of 4d $\mathcal{N} = 2$ SCFTs[56] engineered by type IIB string theory probing ADE singularities, along with their mini-versal deformations:

$$
\begin{aligned}
F_{A_k}(x,y) &= x_1^2 + x_2^2 + x_3^2 + x_4^{k+1} , & \hat{F}_{A_k}(x,\lambda) &= F_{A_k} + \lambda_1 + \lambda_2 x_4 + \ldots + \lambda_k x_4^{k-1} , \\
F_{D_k}(x,y) &= x_1^2 + x_2^2 + x_3^{k-1} + x_3 x_4^2 , & \hat{F}_{D_k}(x,\lambda) &= F_{D_k} + \lambda_1 + \lambda_2 x_3 + \ldots + \lambda_k x_3^{k-1} , \\
F_{E_6}(x,y) &= x_1^2 + x_2^2 + x_3^3 + x_4^4 , & \hat{F}_{E_6}(x,\lambda) &= F_{E_6} + \lambda_1 + \lambda_2 x_3 + \lambda_3 x_4 + \lambda_4 x_4^2 \\
& & &\quad + \lambda_5 x_3 x_4 + \lambda_6 x_3 x_4^2 , \\
F_{E_7}(x,y) &= x_1^2 + x_2^2 + x_3^3 + x_3 x_4^3 , & \hat{F}_{E_7}(x,\lambda) &= F_{E_7} + \lambda_1 + \lambda_2 x_3 + \lambda_3 x_4 + \lambda_4 x_4^2 \\
& & &\quad + \lambda_5 x_4^3 + \lambda_6 x_3^2 x_4 + \lambda_7 x_3 x_4 , \\
F_{E_8}(x,y) &= x_1^2 + x_2^2 + x_3^3 + x_4^5 , & \hat{F}_{E_8}(x,\lambda) &= F_{E_8} + \lambda_1 + \lambda_2 x_3 + \lambda_3 x_4 + \lambda_4 x_4^2 \\
& & &\quad + \lambda_5 x_4^3 + \lambda_6 x_3 x_4 + \lambda_7 x_3 x_4^2 + \lambda_8 z_3 z_4^3 .
\end{aligned}
\tag{5.9}
$$

The above $\hat{F}$ defines a (generalized) Seiberg-Witten geometry with a deformed CY 3-fold fibered over the special Kähler moduli space parametrized by $\lambda_\alpha$ which are to be identified with the CB data in the 4d SCFT. This construction is different from what we have encountered in the previous sections: in Section 3.3, the SW geometry is given by a SW curve, i.e., $\hat{F} = \{x, z, \lambda\}$, fibered over the CB moduli space, which generally follows from the Hitchin system in Class $\mathcal{S}$ constructions. In the case of a type IIB geometric construction on an IHS, the SW geometry is described generally by a 3-fold instead of a curve, fibered over the moduli space [26, 31, 268, 274]. Moreover, the SW differential one-form is replaced by the Calabi-Yau holomorphic canonical 3-form $\Omega$ on $\hat{F}$,[57]

$$\Omega_3 = \frac{dx_1 \wedge dx_2 \wedge dx_3 \wedge dx_4}{d\hat{F}(x_i, \lambda_\alpha)} . \tag{5.10}$$

The BPS states in the CB of the resulting 4d SCFT arise from D3-branes wrapping around supersymmetric 3-cycles (called special Lagrangian submanifolds) on the deformed 3-fold $\hat{F}$ and their masses are determined by the integral of $\Omega_3$.[58] Only in special cases, the SW geometry in (5.6) can be reduced to a curve fibration and similarly for the SW differential (5.10). Such examples occur when the (non-compact) CY 3-fold is an ALE fibration over $\mathbb{P}^1$ [26, 164].

The set of $\lambda_\alpha$ comprises the moduli of the complex structure deformations of the singular Calabi-Yau 3-fold $W$, which correspond to the parameters on the CB of the resulting 4d SCFT preserving the $\mathcal{N} = 2$ SUSY including chiral couplings, masses, and Coulomb branch VEVs. To unpack the physical interpretation of each parameter $\lambda_\alpha$, we need to look at their scaling dimensions $\Delta(\lambda_\alpha)$ or equivalently their $U(1)_r$ charges. Recall from Section 2 that the scaling dimension $\Delta(\mathcal{O})$ of an operator $\mathcal{O}$ on a CB of a 4d SCFT, viewed as a superchiral primary, is proportional to its $U(1)_r$ charge, which is then proportional to the charge $q$ under the $\mathbb{C}^*$ action, i.e., $\Delta(\mathcal{O}) = aq(\mathcal{O})$ with $a$ the proportionality constant. To determine $a$, one can use the fact that the canonical 3-form $\Omega_3$, whose charge under $\mathbb{C}^*$ is $(\sum_{i=1}^{4} q_i - 1)$, has scaling

---

[56]Note that such 4d SCFTs have no marginal and irrelevant deformations.

[57]An interesting connection with the so-called primitive forms has been found in [275].

[58]Note that many 4d $\mathcal{N} = 2$ SCFTs, especially engineered from the type II string theories, share some common features with 2d $\mathcal{N} = (2,2)$ SCFTs including the BPS spectra and their wall-crossing phenomenon, whose details we do not explain in this review but rather refer to [72] on the so-called *4d-2d correspondences*.

dimension $\Delta(\Omega_3) = 1$, as the integration of $\Omega_3$ over a 3-cycle gives the mass of a BPS particle. Hence, we have

$$a = \frac{1}{\sum_{i=1}^4 q_i - 1}. \tag{5.11}$$

Now we want to determine the $\mathbb{C}^*$ charge $q(\lambda_\alpha)$ of the parameter $\lambda_\alpha$ for the deformation $\lambda_\alpha g^\alpha$ in (5.6). Because

$$q(\lambda_\alpha) = 1 - q(g^\alpha), \tag{5.12}$$

it suffices to know the charge $q(g^\alpha)$ of $g^\alpha := x_1^{\alpha_1} x_2^{\alpha_2} x_3^{\alpha_3} x_4^{\alpha_4}$, which is

$$q(g^\alpha) = \sum_{i=1}^4 q_i \alpha_i. \tag{5.13}$$

Hence we conclude that the scaling dimension $\Delta(\lambda_\alpha)$ of $\lambda_\alpha$ is given by

$$\Delta(\lambda_\alpha) = \frac{1 - \sum_{i=1}^4 q_i \alpha_i}{\sum_{i=1}^4 q_i - 1}. \tag{5.14}$$

**Example 5.2:**

Consider the singularity $F = x_1^{a_1} + x_2^{a_2} + x_3^{a_3} + x_4^{a_4}$ with the constraint $\sum_{i=1}^4 \frac{1}{a_i} > 1$. The Milnor number is then $\mu = \sum_{i=1}^4 (a_i - 1)$. The Jacobi ideal $(dF)$ is generated by $\partial_{x_i} F$ which gives $(x_1^{a_1-1}, x_2^{a_2-1}, x_3^{a_3-1}, x_4^{a_4-1})$. Thus the corresponding Jacobi algebra $J$ has the following monomial basis:

$$x_1^{\alpha_1} x_2^{\alpha_2} x_3^{\alpha_3} x_4^{\alpha_4}, \; 0 \leqslant \alpha_i \leqslant a_i - 2. \tag{5.15}$$

Consequently the scaling dimension of the coefficients for the above deformation is

$$\Delta(\lambda_\alpha) = \frac{1 - \sum_{i=1}^4 \frac{\alpha_i}{a_i}}{\sum_{i=1}^4 \frac{1}{a_i} - 1}. \tag{5.16}$$

The set of deformation parameters $\lambda_\alpha$ can be classified according to their scaling dimensions $\Delta(\lambda_\alpha)$. Among them, we have

1. A parameter $\lambda_{\alpha'}$ with scaling dimension $\Delta(\lambda_{\alpha'}) > 1$ is a Coulomb branch VEV parameter, i.e., the VEV $\langle \mathcal{O}_{\alpha'} \rangle := \lambda_{\alpha'}$ of a CB operator $\mathcal{O}_{\alpha'}$ of the same scaling dimension. We denote by $r$ the number of Coulomb branch parameters, i.e., the rank of the 4d $\mathcal{N} = 2$ SCFT. If there is a weakly coupled gauge theory description, this will coincide with the rank of the gauge group. Among this set of $\lambda_{\alpha'}$'s, each parameter $\lambda_{\alpha'}$ with $1 < \Delta(\lambda_{\alpha'}) < 2$ leads to a relevant CB operator, similarly if $\Delta(\lambda_{\alpha'}) = 2$ or $\Delta(\lambda_{\alpha'}) > 2$, the corresponding CB operator is marginal or irrelevant respectively.

2. A parameter $\lambda_\beta$ with scaling dimension $\Delta(\lambda_\beta) < 1$ is identified as a chiral coupling constant, which is accompanied by an $\mathcal{N} = 2$ supersymmetric F-term deformation $\int d^4\theta \mathcal{O}_\alpha \lambda_\beta$ of the SCFT. In particular, each parameter $\lambda_\beta$ with scaling dimension $\Delta(\lambda_\beta) = 0$ corresponds to an exactly marginal deformation.

3. A parameter $\lambda_\gamma$ with scaling dimension $\Delta(\lambda_\gamma) = 1$ is a mass parameter, which can be viewed as the VEV of the scalar in a background vector multiplet for the maximal Cartan torus of the flavor symmetry group $F$. We denote the number of the mass parameters as $f$, i.e., the rank of the flavor symmetry $F$ in the corresponding 4d $\mathcal{N} = 2$ SCFT.

From the second point above, one can conclude that the parameters $\lambda_\alpha$ come in pairs $(\lambda_{\alpha'}, \lambda_\beta)$ satisfying

$$\Delta(\lambda_{\alpha'}) + \Delta(\lambda_\beta) = 2, \tag{5.17}$$

with the exception of the mass parameters. Each such pair $(\lambda_{\alpha'}, \lambda_\beta)$ forms an $\mathcal{N} = 2$-preserving F-term deformation $\int d^4\theta \, \lambda_\beta \mathcal{O}_{\alpha'}$ with $\mathcal{O}_{\alpha'}$ a CB chiral primary operator whose VEV is $\lambda_{\alpha'}$. Thus we conclude $\mu = 2r + f$ due to the pairings, and the dimension $\mu$ can be identified as the rank of the electromagnetic charge lattice extended with the flavor charges.

Given the full spectrum of parameters $\lambda_\alpha$ of a 4d $\mathcal{N} = 2$ SCFT, one can proceed to compute the conformal central charges $(a, c)$ introduced in Section 2.6, which measure the degrees of freedom of such a theory. Considering Eq. (2.50), the Wess-Zumino contributions $R(A)$ and $R(B)$ are determined by the singularity deformation theory as follows [48]. Here $R(A)$ is determined by the CB operator scaling dimensions, namely $\lambda_\alpha$ with $\Delta[\lambda_\alpha] > 1$, and $R(B)$ receives contributions from codimension-1 singularities of $\hat{F}$ where there is an extra massless hypermultiplet and the local contributions are given by the $U(1)_r$-charge of the local coordinate transverse to such a singularity. For the 4d SCFT engineered by an IHS in type IIB string theory,[59] they are explicitly given by [48]

$$R(A) = \sum_{\Delta[\lambda_\alpha] > 1} \Delta[\lambda_\alpha] - r, \qquad R(B) = \frac{\hat{\alpha}_{max}}{4} \mu, \tag{5.18}$$

where $\hat{\alpha}_{max}$ refers to the maximal scaling dimension in the CB spectrum and is given by

$$\hat{\alpha}_{max} = \left( \sum_i^4 q_i - 1 \right)^{-1}. \tag{5.19}$$

## 5.2 Examples of $\mathcal{N} = 2$ SCFTs from $(G, G')$ Singularities

In this section, let us focus on a special type of isolated hypersurface singularities called $(G, G')$ singularities. In [72], the authors considered isolated 3-fold hypersurface singularities whose defining polynomials are given by the sums of two polynomials that define ADE surface singularities,

$$F : F_G(x_1, x_2) + F_{G'}(x_3, x_4) = 0, \tag{5.20}$$

where each $F_G$ belongs to one of the following types:

$$
\begin{aligned}
F_{A_k}(x, y) &= x^2 + y^{k+1}, \\
F_{D_k}(x, y) &= x^2 y + y^{k-1}, \\
F_{E_6}(x, y) &= x^3 + y^4, \\
F_{E_7}(x, y) &= x^3 + x y^3, \\
F_{E_8}(x, y) &= x^3 + y^5.
\end{aligned}
\tag{5.21}
$$

It turns out that such types of singularities generate a large class of Argyres-Douglas theories, dubbed as $(G, G')$ theories.[60] Note that the labeling $(G, G')$ is not unique: as one can easily see from (5.20) that there is an obvious isomorphism $(G, G') \sim (G', G)$. Furthermore, there are many additional equivalences [47, 72] such as

$$(A_1, D_4) \sim (A_2, A_2), (A_1, E_6) \sim (A_3, A_2), (D_4, A_3) \sim (E_6, A_2), (E_8, A_3) \sim (E_6, A_4), \ldots. \tag{5.22}$$

---

[59]In such a class of 4d SCFTs, there are also no free hypermultiplets, i.e., $h = 0$ in (2.50).

[60]Note that the label $(G, G')$ for such an AD theory means that the BPS quiver associated with the AD theory has the shape of the product of $G$ and $G'$ Dynkin diagrams. In particular, $\mu = 2r + f = r_G r_{G'}$.

**Exercise 5.1** Compute the central charge $(a, c)$ of the 4d $\mathcal{N} = 2$ SCFT engineered from the above $(A_1, A_2)$ singularity. What about the generic cases with $(A_n, A_m)$ singularities?

**Hint:** $R(B)$ in (2.50) reduces to $R(B) = \frac{r_G r_{G'}}{4} \frac{h_G^\vee h_{G'}^\vee}{h_G^\vee + h_{G'}^\vee}$ in the class of the $(G, G')$ theories, with $h_G^\vee$ and $r_G$ the dual Coxeter number and the rank of the group $G$, respectively (see e.g., [211, 229]).

**Example 5.3:**

Let us consider an explicit example of the $(A_1, A_2)$ singularity for the sake of illustration. According to (5.21), the associated hypersurface is defined by the following polynomial,

$$F = x_1^2 + x_2^2 + x_3^2 + x_4^3. \tag{5.23}$$

One can then easily read off the $\mathbb{C}^*$ charges $q_i = \{\frac{1}{2}, \frac{1}{2}, \frac{1}{2}, \frac{1}{3}\}$ and the proportionality constant $a = \frac{1}{\sum_{i=1}^4 q_i - 1} = \frac{6}{5}$. The ideal generated by $dF$ is $(x_1, x_2, x_3, x_4^2)$ and thus the corresponding Jacobi algebra $J(F)$ has the following monomial basis:

$$x_4^{\alpha_4}, \ 0 \leqslant \alpha_4 \leqslant 1. \tag{5.24}$$

Then one can easily write down the corresponding SW geometry as the deformation (5.6)

$$\hat{F} = x_1^2 + x_2^2 + x_3^2 + x_4^3 + \lambda_1 + \lambda_2 x_4, \tag{5.25}$$

where $\lambda_1$ is identified with a CB VEV with scaling dimension $\Delta(\lambda_1) = \frac{6}{5}$ and $\lambda_2$ the chiral coupling with scaling dimension $\Delta(\lambda_2) = \frac{4}{5}$, according to (5.16). This is the same as the original Argyres-Douglas theory that arises from the IR of the $\mathcal{N} = 2$ SU(3) pure SYM.

## 5.3 Relation with Class $\mathcal{S}$ Constructions

So far, we have introduced the general aspects of geometric engineering of 4d $\mathcal{N} = 2$ SCFT from isolated hypersurface singularities. One may ask if there are any relations between the type IIB engineering and the Class $\mathcal{S}$ constructions in Section 4. The answer is yes, and this is expected to be a consequence of the M-theory/IIB duality with five-branes and nontrivial internal geometries.

We can be more specific when the non-compact Calabi-Yau $W$ is a fibration of the ALE space over a curve. In this case, one can see explicitly using a T-duality that the CB of the resulting 4d $\mathcal{N} = 2$ SCFT is equivalently described by a five-brane wrapping a Riemann surface that coincides with the SW curve for the SCFT. This T-duality was studied in [276] and maps type IIB theory near an $A_{n-1}$ singularity of the ALE space to a stack of $n$ coinciding NS5-branes in type IIA theory whose worldvolume is described the $A_{n-1}$ type 6d $\mathcal{N} = (2, 0)$ SCFT in the low energy limit, and the 6d self-dual strings from wrapped D3-branes in type IIB correspond to the boundaries of D2-branes ending on the type IIA NS5-branes. For more details of the T-duality, we refer to the review [277].

For example, let us consider type IIB compactified on a Calabi-Yau manifold with the $(A_{m-1}, A_{n-1})$ singularity

$$x^m + y^n + z^2 + w^2 = 0, \tag{5.26}$$

which can be further viewed as an $A_{m-1}$ singularity of the ALE space fibered over the $y$-plane

locally:

$$\prod_{i=1}^{m}(x - a_i(y)) + z^2 + w^2 = 0\,, \tag{5.27}$$

where $a_i(y)$ can be viewed as a multivalued function on the $y$-plane.[61] By T-duality [276], this type IIB setup maps to a stack of $m$ NS5-branes in type IIA fibered over the $y$-plane. Lifting it to M-theory, we have $m$ M5-branes at $x = a_i(y)$, $z = w = 0$. Note that at $y = 0$, the $A_{n-1}$ singularity is developed and the $m$ M5-branes join together which host the $\mathcal{N} = (2,0)$ $A_{m-1}$ type SCFT. While at a generic point $y \neq 0$, the stack of the $m$ M5-branes splits and corresponds to a point on the Coulomb branch, and the 4d effective theory can be described by the low energy limit of a single M5-brane wrapping a Riemann surface $\Sigma := \prod_{i=1}^{m}(x - a_i(y)) = 0$ [26], which can be seen as an $m$-sheeted cover of the $y$-plane. This reproduces the Class $\mathcal{S}$ description of Section 4.3 where the Gaiotto curve $\mathcal{C}$ here is identified with the sphere $\mathbb{P}^1$, parameterized locally by $y$, with corresponding irregular punctures, as in Sections 4.8.2 and 4.8.3.

Moreover, these connections between the type IIB and Class $\mathcal{S}$ constructions via T-duality can be extended to all $(A, G)$ theories [47], where $G$ can be an arbitrary singularity of the above ADE types. To see that, one can take three coordinates (say $(x, z, w)$) giving rise to the type $G$ singularity $F_G(x, z, w) = 0$ as in (5.21) (and hence by T-duality mapped to the $(2, 0)$ SCFT of the $G$ type in 6d), and the other coordinate (say $y$) as defining how the type $G$ singularity fibers over the $y$-plane. We thus have a Class $\mathcal{S}$ description for the $(A, G)$ theories from the type $G$ $\mathcal{N} = (2, 0)$ SCFT compactified on a sphere with an irregular puncture (see [47] for details and more examples of these IIB and Class $\mathcal{S}$ correspondences). The SCFTs engineered by 3-fold singularities of the other types $(D, D)$, $(D, E)$ and $(E, E)$, however, do not have any known Class $\mathcal{S}$ descriptions.[62]

Before closing this subsection, we would like to stress that with such equivalence, all the data of a class $\mathcal{S}$ construction can be exploited from the local ALE fibration in the dual IIB geometric engineering including the global structures, which have been studied recently for examples in [278, 279].

## 5.4 Further Generalizations

### 5.4.1 General Singularities

In the previous sections, we mainly considered non-compact Calabi-Yau 3-folds described by certain isolated hypersurface singularities and saw that such geometric engineering in type IIB can provide many interesting 4d $\mathcal{N} = 2$ SCFTs. A complete list of hypersurface singularities satisfying the conditions to define 4d SCFTs has been obtained in [32, 48] (see also [52, 60]). The natural follow-up question would be whether one can go beyond the IHS to engineer more 4d $\mathcal{N} = 2$ SCFTs.

Indeed, one can also consider a more general isolated canonical singularity such as an isolated complete intersection singularity (ICIS), which is defined by several polynomials as

$$W := F_1(x_1, x_2, \ldots, x_{n+3}) = F_2(x_1, x_2, \ldots, x_{n+3}) = \ldots = F_n(x_1, x_2, \ldots, x_{n+3}) = 0 \subset \mathbb{C}^{n+3}\,. \tag{5.28}$$

To engineer a 4d SCFT, there are certain (sufficient) conditions on the structure of the singularity similar to the ones we have discussed in the case of IHS:

---

[61]The non-trivial two-cycles in the ALE fiber associated with the pairs $(a_i(y), a_j(y))$ correspond to the weights of $A_{m-1}$.

[62]Conversely, any Class $\mathcal{S}$ construction has a dual geometric description in the type IIB by a non-compact Calabi-Yau 3-fold. However, in general, the singularity in this 3-fold will not be isolated. See the next section and Footnote 64 for related comments. See also [278, 279] for recent discussions.

1. The singularity must be isolated. Namely, there is a unique solution for the equations $F_1 = F_2 = \ldots = F_n = 0$ and $dF_a = 0, a = 1, 2, \ldots, n$.

2. $W$ is a complete intersection, i.e., the Jacobi matrix $\frac{\partial F_a}{\partial x_i}, a = 1, \ldots, n, \alpha = 1, \ldots, n+3$ has rank $n$ everywhere except at the singular point.

3. Each polynomial $F_i$ in $W$ is homogeneous with degree $d_i$, and the weights of the coordinates $x_i$ are $w_i$.

4. The weights $w_i$, in the same spirit as $\sum_i q_i > 1$ for the IHS introduced before, need to satisfy the condition $\sum_i w_i > \sum_i d_i$.

It was recently proved in [50] that the above conditions require that $n = 2$, i.e., the ICIS must be defined by at most two polynomials $(F_1, F_2)$, leading to 303 classes of singularities in total. The constructions of ICIS are in many ways similar to the IHS. For example, the mini-versal deformation of the ICIS is defined as

$$F = F + \sum_{\alpha=1}^{\mu} \lambda_\alpha \phi_\alpha. \tag{5.29}$$

Here $F := (F_1, F_2)$ is a $2 \times 1$ column vector and $\phi_\alpha$, viewed as $2 \times 1$ column vectors with only one non-zero entry, give a monomial basis of the Milnor ring,

$$J(F_i) = \mathbb{C}^5 / (dF). \tag{5.30}$$

The canonical holomorphic 3-form on the 3-fold ICIS here is defined by

$$\Omega = \frac{dx_1 \wedge dx_2 \wedge \ldots \wedge dx_5}{dF_1 \wedge dF_2}, \tag{5.31}$$

which descends to the 4d $\mathcal{N} = 2$ SW differential in the resulting 4d SCFT. Similarly, one can read off various physical quantities such as the scaling dimensions of the parameters $\lambda$'s, central charges $(a, c)$ from the geometry of the ICIS as the ones from the IHS elaborated in the previous subsections.

Moreover, it was conjectured in [48] that the most general isolated singularities that generate 4d $\mathcal{N} = 2$ SCFTs are **rational graded Gorenstein** singularities.[63] The graded condition here implies that the singularities have a $\mathbb{C}^*$ action required by the $U(1)_r$ symmetry of a 4d $\mathcal{N} = 2$ SCFT, and the Gorenstein condition implies that there is a distinguished top form $\Omega$ which descends to a 4d $\mathcal{N} = 2$ SW differential. The rationality condition ensures that the top form $\Omega$ has a positive charge under the $U(1)_r$ symmetry.

One can even go beyond the isolated singularities by considering non-isolated ones in the geometric engineering. In such cases, there can be a complex one-dimensional singular locus. Indeed, as we have encountered in Section 4, the Class $\mathcal{S}$ constructions can be viewed as this type since they are partly classified by ADE singularities of ALE spaces, rather than Calabi-Yau 3-folds, hence reducing the dimension by one. A generic Class $\mathcal{S}$ construction can then be rephrased in terms of a hypersurface singularity defined by the following

$$F(x_1, x_2, x_3, x_4) = F_{ADE}(w_1(x_i), w_2(x_i), w_3(x_i)), \tag{5.32}$$

where $F_{ADE}(w_1(z_i), w_2(z_i), w_3(z_i))$ refers to an ADE singularity in terms of the three variables $w_i$. Thus the loci $\{w_i = 0\}$ gives a curve singularity in the ambient space parametrized by $x_i$.[64]

---

[63]A rational graded Gorenstein is also known as an index one canonical singularity.

[64]A curious reader may want to compare this to the T-duality discussed in Section 5.3 that relates certain IHS to a Class $\mathcal{S}$ construction. Indeed, one can find that the $\{w_i = 0\}$ loci for an isolated singularity of the $(G, G')$ type written in the form of (5.32) is a point, rather than a curve. Said differently, when mapped to the type IIB side by T-duality, a generic Class $\mathcal{S}$ construction leads to a curve singularity in the hypersurface 3-fold in type IIB, whereas a special subset of Class $\mathcal{S}$ constructions produces an isolated singularity (such as those that engineer $(A, G)$ theories).

However, the deformation theory of such non-isolated singularities is more complicated, and a detailed discussion is beyond the scope of this review.

### 5.4.2 F-theory Constructions and S-folds

Given that F-theory[65] is a non-perturbative completion of type IIB string theory and that it provides powerful geometrization of QFTs, one might wonder how to embed 4d $\mathcal{N} = 2$ SCFTs into F-theory constructions directly which may be useful to understand subtle features of known theories and also to produce previously unknown SCFTs. In this section, we would like to briefly survey these constructions, and learn what new 4d theories F-theory can bring to the table.

In F-theory constructions, a 4d $\mathcal{N} = 2$ SCFT is typically realized as the worldvolume theory of a stack of D3-branes probing a stack of $(p, q)$ seven-branes with a constant axion-dilaton $\tau$. Here $(p, q)$ transforms as a doublet under the type IIB SL(2, $\mathbb{Z}$) duality and these seven-branes are strong coupling cousins of the familiar D7-brane which corresponds to the $(1, 0)$ type.

Such a stack of seven-branes leads to a local elliptic K3 surface[66] with fixed type of Kodaira's singularity and gauge group $F$, which we list in Table 6 together with their values of $\tau$ [174, 176]. The stack of the D3-branes extending in $\mathbb{R}^{1,3} \subset \mathbb{R}^{1,7}$ can be viewed as non-Abelian instantons of the 8d SYM theories on the seven-branes, as a consequence of the Wess-Zumino coupling $\int \mathrm{tr}(F \wedge F) \wedge C_4$ on the seven-brane worldvolume. In such settings, the 4d worldvolume of the stack of D3-branes hosts an $\mathcal{N} = 2$ gauge theory, and the axion-dilaton $\tau$, i.e., the complex structure of the elliptic fiber of $K3$, can be identified with the complexified 4d gauge coupling. Since the seven-brane is non-compact in the transverse directions to the D3-branes, the 7-7 strings are frozen and one can focus on the 3-3 strings and the 3-7 strings in the 4d limit, where the former realizes the gauge bosons and the latter realizes the charged matter in the gauge theory, which further carries a global symmetry $F$ from the seven-brane. The number of probe D3-branes corresponds to the rank of the gauge group $G$ of the 4d effective gauge theory realized on their worldvolume. When the D3-branes are placed on top of one of the Kodaira singularities of the K3 surface listed in Table 6, the 4d effective gauge theory on the D3-branes reaches a conformal point, so a 4d $\mathcal{N} = 2$ SCFT with flavor symmetry $F$ emerges. Indeed, the conformality of the theory is also reflected by the fact that the axion-dilaton $\tau$ is constant near these singularities [176].

Table 6: Types of Kodaira's singularities with constant values of the axion-dilaton $\tau$, as well as the corresponding gauge groups $F$ on the seven-brane. Note that for $F = \mathrm{SO}(8)$, $\tau$ is an unfixed constant, and the background is nothing but the $\mathbb{Z}_2$ involution of a perturbative type IIB orientifold described by 4 D7-branes on top of an O7-plane. To better engage with later discussion, for each seven-brane, we also list its corresponding $\Delta_7$ which determines the deficit angle $\frac{2\pi}{\Delta_7}$, which measures the change in the angular coordinate of the transverse geometry when going around the seven-branes.

| Kodaira type | $II$ | $III$ | $IV$ | $I_0^*$ | $IV^*$ | $III^*$ | $II^*$ |
|---|---|---|---|---|---|---|---|
| $F$ | $\emptyset$ | SU(2) | SU(3) | SO(8) | $E_6$ | $E_7$ | $E_8$ |
| $\Delta_7$ | 6/5 | 4/3 | 3/2 | 2 | 3 | 4 | 6 |
| $\tau$ | $e^{\pi i/3}$ | $e^{\pi i/2}$ | $e^{\pi i/3}$ | free | $e^{\pi i/3}$ | $e^{\pi i/2}$ | $e^{\pi i/3}$ |

Let us consider a single D3-brane probing the $I_0^*$ singularity in F-theory listed in Table 6

---

[65]For a nice comprehensive review on F-theory, we refer to the TASI lecture review [280].

[66]Namely, the 10d background geometry locally is $\mathbb{R}^{1,7} \times \mathbb{P}^1$ with $\mathbb{P}^1$ being the base of the K3 surface.

as an example (the generalization to other types of singularity is straightforward). Such a setup realizes the 4d $\mathcal{N} = 2$ U(1) gauge theory with eight charged hypermultiplets from the strings between the D3 and the 4 D7-branes and their mirror images under the orientifold action [174, 175, 281] (or see [282] for a more systematical description). When the single D3-brane is on the top of the $I_0^*$ singularity, the gauge symmetry U(1) is enhanced to SU(2) by the $\mathbb{Z}_2$ monodromy from the 3-3 strings encircling the singularity, and the flavor symmetry $F$ is also enhanced to SO(8) (from the U(4) on the D7-branes).[67] The resulting SCFT is the $\mathcal{N} = 2$ SU(2) conformal SQCD with four fundamental hypermultiplets. By moving the D3-brane away from the $I_0^*$ singularity along the transverse space normal to the seven-branes (i.e., the base $B_1 := \mathbb{P}^1$ of the K3), we probe the Coulomb branch of this SCFT and indeed the worldvolume theory on the D3-brane reduces to a U(1) gauge theory and the holomorphic variation of $\tau$ over $B_1$ [282] quantitatively matches the behavior of the effective gauge coupling on the Coulomb branch obtained from the Seiberg-Witten geometry [21]. Furthermore, the gauge instantons in the 4d field theory which play an important role in the EFT are identified with the D($-1$) instantons in type IIB string theory, whose non-perturbative effects are automatically encoded in the profile of the elliptic fibration of K3 [282]. In other words, one can directly identify the SW curve of this rank-1 4d SCFT as the elliptic fiber of the K3 with the $I_0^*$ singularity resolved. The Higgs branch, on the other hand, corresponds to dissolving the D3-brane into a gauge flux on the D7-branes, which agrees with the reduced moduli space of one SO(8) gauge instanton.

The generalization to rank-$r$ is also straightforward by considering a stack of $r$ D3-branes probing one of the above seven-branes (equivalently Kodaira singularities). The $r$-dimensional Coulomb branch is then simply the symmetric orbifold of the "seed" Coulomb branch of the corresponding rank-1 case,[68] and the Higgs branch is given by the reduced moduli space of $r$ $F$-instantons of complex dimension $2(rh_F^\vee - 1)$ where $h_F^\vee$ denotes the dual Coxeter number for $F$.

Note that the above Kodaira singularities probed by D3-branes are all crepant. It is then natural to ask whether 4d $\mathcal{N} = 2$ SCFTs can be constructed by D3-branes in F-theory, from probing more exotic (but allowed) singularities such as terminal singularities. Indeed, a new construction of 4d $\mathcal{N} = 2$ SCFTs through the so-called S-folds in the presence of D7-branes has recently been worked out in [57, 58], followed by [59, 283, 284], which argued that it can reproduce all rank-1 4d SCFTs classified pure-theoretically in [118, 163, 165, 285].

Slightly later, a generalization with discrete torsion for S-folds has been discussed in [59, 283], which generates 4d SCFTs that can also be obtained by Higgsing from the torsionless S-folds in [57, 58]. All in all, such novel methods provide a large class of constructions for 4d SCFTs and are still under investigation in a bid to shed new light on the classification of 4d $\mathcal{N} = 2$ SCFTs, at least for the cases with low ranks. Following [57], let us also briefly summarize the underlying ideas of these types of constructions.

An S-fold in F-theory, firstly introduced in [263, 286] to study 4d $\mathcal{N} = 3$ SCFTs,[69] is a (non-perturbative) generalization of an orientifold in type IIB string theory such that the geometric orbifold action is accompanied by a non-trivial S-duality action on the axion-dilaton $\tau = C_0 + ie^{-\phi}$, thereby fixing $\tau$ to a specific value. Roughly speaking, it is equivalent to F-theory on the transverse geometry $(\mathbb{C}^3 \times T^2)/\mathbb{Z}_k$ with $k = 2, 3, 4, 6$.[70]

---

[67]This enhancement can also be seen by T-dualizing IIB on $T^2/\mathbb{Z}_2$ with one O7-plane and four D7-branes at each of the four fixed points to type I string theory on $T^2$. The corresponding Wilson lines on $T^2$ break the type I gauge group SO(32) to SO(8)$^4$ and each SO(8) factor is interpreted as the enhanced gauge symmetry at one of the four fixed points of $T^2/\mathbb{Z}_2$ on the type IIB side. Furthermore, under this T-duality, the type IIB D3-brane is mapped to a D5-brane wrapping the $T^2$ in the type I background and consequently hosts the enhanced USp(2) gauge symmetry because of the spacetime filling orientifold plane.

[68]However, the SW geometry of the higher rank theory can not be identified with the elliptic fibration of the K3.

[69]These theories have SU(3)$_r$ × U(1)$_r$ R-symmetry and the massless fields on the vacuum moduli space are coincident with those in the $\mathcal{N} = 4$ theories.

[70]One can immediately see that the $k = 2$ S-fold reduces to the usual O3$^-$ plane in type IIB orientifold compact-

In order to see how an S-fold engineers a 4d $\mathcal{N} = 3$ theory, let us first look at $\mathbb{C}^3$ and denote the coordinates in $\mathbb{C}^3$ by $z_i$ with $i = 1, 2, 3$ and $z_4 \equiv x + \tau y$ for $T^2$ with complex structure $\tau$. The action of $\mathbb{Z}_k$ is then defined by [57]

$$z_i \to e^{i\Phi_i} z_i, \quad e^{i\Phi_i} \in \mathbb{Z}_k, \tag{5.33}$$

which is expected to reduce the 16 supercharges carried by D3-branes probing the 6d transverse space $\mathbb{C}^3$ by one fourth.[71] To see that, note that the isometry of $\mathbb{C}^3$ gives rise to the R-symmetry $SU(4)_r$ of the 4d $\mathcal{N} = 4$ theory on the worldvolume of D3-branes, under which the 16 supercharges $Q_i$ with $i = 1, 2, 3, 4$ transform in the four-dimensional fundamental representation. The quotient $\mathbb{Z}_k$ acting on $\mathbb{C}^3$ thereby naturally induces an R-symmetry rotation on the supercharges $Q_i$, which is generated [57, 287]

$$
\begin{aligned}
r_k : Q_i \to & M_j^i Q_j, \\
M_j^i = & \operatorname{diag}\left(e^{i(\Phi_1 + \Phi_2 + \Phi_3)/2}, e^{i(\Phi_1 - \Phi_2 - \Phi_3)/2}, e^{i(-\Phi_1 + \Phi_2 - \Phi_3)/2}, e^{i(-\Phi_1 - \Phi_2 + \Phi_3)/2}\right).
\end{aligned}
\tag{5.34}
$$

Furthermore, the $\mathbb{Z}_k$ action on $T^2$ also has certain effects on the supercharges. To be an isometry of $T^2$, the group $\mathbb{Z}_k$ must take $k = 2, 3, 4, 6$. As a result, the complex structure $\tau$ of $T^2$ is restricted, except for the $k = 2$ case, to the values listed in Table 7. Due to the

Table 7: The fixed values of the axion-dilaton $\tau$ for the $\mathbb{Z}_k$ quotient, $k = 2, 3, 4, 6$. The corresponding $SL(2, \mathbb{Z})$ monodromy matrices $\rho$ are also fixed.

| | $k = 2$ | $k = 3$ | $k = 4$ | $k = 6$ |
|---|---|---|---|---|
| $\tau$ | free | $\tau = e^{\pi i/3}$ | $i$ | $\tau = e^{\pi i/3}$ |
| $\rho$ | $\begin{pmatrix} -1 & 0 \\ 0 & -1 \end{pmatrix}$ | $\begin{pmatrix} -1 & -1 \\ 1 & 0 \end{pmatrix}$ | $\begin{pmatrix} 0 & -1 \\ 1 & 0 \end{pmatrix}$ | $\begin{pmatrix} 0 & -1 \\ 1 & 1 \end{pmatrix}$ |

special value of $\tau$, the S-duality element of the $SL(2, \mathbb{Z})$ induces an additional phase on the supercharges $Q_i$, which in this setting acts as [99, 263, 286, 288]

$$s_k : Q_i \to e^{-\pi i/k} Q_i. \tag{5.35}$$

Now by choosing suitable phases in (5.34) such that $\Phi_1 = \Phi_2 = -\Phi_3 = \frac{2\pi}{k}$, 12 out of the 16 supercharges are preserved by the quotient $\mathbb{Z}_k$ generated by $r_k s_k$, leading to a 4d $\mathcal{N} = 3$ SCFT on the probe D3-branes [263, 286]. Note that all these (rigid) $\mathcal{N} = 3$ theories generated through these S-folds are non-Lagrangian and hence inherently strongly coupled.[72] For more details on these $\mathcal{N} = 3$ SCFTs, we refer to [263, 286, 290–293].

The reader may find that the two transformations $s_k$ and $r_k$ on the supercharges induced from the S-duality twist and the R-symmetry $SO(6)_r$ are not completely canceled out in general, and hence with different choices of the phases $\Phi_i$, it is possible to attain less supersymmetry on the probe D3-branes. For example, by choosing

$$\Phi_1 = \frac{2\pi}{k}, \ \Phi_2 = -\Phi_3 \mod 2\pi\mathbb{Z}, \tag{5.36}$$

the worldvolume of the stack of D3-branes preserves $\mathcal{N} = 2$ supersymmetry. From now on, we refer to this type of S-folds as $\mathcal{N} = 2$ S-folds, in order to differentiate them from the above $\mathcal{N} = 3$ S-folds.

---

ifications, which has a perturbative description.

[71]Note that the fiber $T^2$ in F-theory is not a bona fide part of the physical spacetime, and thus are not probed by the D3-branes.

[72]Note that a perturbative 4d $\mathcal{N} = 3$ theory with a Lagrangian description necessarily enhances to $\mathcal{N} = 4$, see e.g., [289].

The main point in [57, 58] is that one can further combine $\mathcal{N} = 2$ S-folds with 7-branes to generate more 4d $\mathcal{N} = 2$ SCFTs. Indeed, when 7-branes are placed at the locus $z_1 = 0$ inside $\mathbb{C}^3$, the supercharges preserved by the 7-branes are the ones with eigenvalue $+1$ with respect to the generator of the rotation SO(2) on the complex plane $z_1$, which is compatible with the action of the S-folds (5.36). However, this is too quick since we have not taken into account the back-reaction from the 7-branes on the geometry. A stack of 7-branes creates a deficit angle of $\frac{2\pi}{\Delta_7}$ in the flat transverse plane where $\Delta_7$ and $\tau$ fixed accordingly in Table 6. We denote the flat coordinate on this conical singularity by $u$ which is related to the $\mathbb{C}$ coordinate $z_1$ before inserting the 7-branes by $z_1 = u^{\Delta_7}$. Thus the $\mathbb{Z}_k$ rotation of $z_1$ induces a rotation of $u$ by an angle of $\frac{2\pi}{k\Delta_7}$, which induces a rotation of the supercharges by a phase of $e^{-\frac{\pi i}{k\Delta_7}}$.[73] As already mentioned, in order to preserve supersymmetry, one should offset this phase by a transformation $\mathbb{Z}_{k\Delta_7} \subset \mathrm{SL}(2,\mathbb{Z})$. Thus, one must have $k\Delta_7 = 2, 3, 4, 6$ and arrives at the following six possibilities [57, 58]:

- $k = 2$ and $\Delta_7 = \frac{3}{2}, 2, 3$, with 7-branes of type $IV, I_0^*$ and $IV^*$.

- $k = 3$ and $\Delta_7 = \frac{4}{3}, 2$, with 7-branes of type $III, I_0^*$.

- $k = 4$ and $\Delta_7 = \frac{3}{2}$, with a 7-brane of type $III$.

As suggested above, the powerful aspect of these F-theory constructions is that one can read off various properties of these 4d $\mathcal{N} = 2$ SCFTs from the elliptic geometries including the CB geometry (for rank 1 cases), conformal anomaly coefficients $(a, c)$, the global symmetries, etc.

Let us take the global symmetries as an example to illustrate this point. One can easily see that there are two distinct contributions to the global symmetries which are manifested from the geometric descriptions: one from the isometries of the background and the other from the gauge symmetries $F$ residing on 7-branes. The obvious U(1) isometry of the complex plane transverse to the 7-branes is identified as the $\mathrm{U}(1)_r$ symmetry of the 4d superconformal algebra. The directions of the 7-branes transverse to the D3-branes give $\mathbb{C}^2/\mathbb{Z}_k$, which has isometry U(2) for $k > 2$ and SO(4) for $k = 2$. Among them, the SU(2) subgroup is identified with the $\mathrm{SU}(2)_R$ symmetry in the 4d superconformal algebra, and the commutant becomes the flavor symmetry. Hence, we have a U(1) flavor symmetry for $k > 2$ and a SU(2) flavor symmetry for $k = 2$. Regarding the gauge symmetry $F$ on the 7-branes, one needs to take into account the $\mathbb{Z}_k$ quotient and only the $\mathbb{Z}_k$ invariant subgroup $H \subset F$ survives as a global symmetry in the 4d SCFT. Combining with the cases without S-folds, we summarize the constructions of rank 1 4d $\mathcal{N} = 2$ SCFTs in the Table 8.

Similar to the $\mathcal{N} = 3$ S-folds discussed in [263, 286], one can further extend the above $\mathcal{N} = 2$ S-folds with discrete torsions, i.e., adding trapped 3-form fluxes at the orbifolding fixed points. In such scenarios, the global symmetries of the resulting SCFTs are slightly different from the above $\mathcal{N} = 2$ S-folds, but can also be read off from the elliptic geometries. We do not cover this aspect here, but rather refer to [59, 283] for further details.

## 6 Conclusion and Open Questions

In the main text, we have presented an introduction to 4d $\mathcal{N} = 2$ superconformal field theories based on a set of three lectures at the Quantum Field Theories and Geometry School in 2020.

---

[73]The monomial $z_1^k$, which is $\mathbb{Z}_k$ invariant, is identified the VEV of a CB operator in the $\mathcal{N} = 2$ theory on the D3-brane, with scaling dimension $k\Delta_7$ which is also its $\mathrm{U}(1)_R$ charge. From this, we also see that the $\mathbb{Z}_k$ rotation of $z_1$ induces a $\mathrm{U}(1)_R$ phase $\omega$ for the supercharges that must satisfy $\omega^{k\Delta_7} = 1$.

Table 8: The rank one SCFTs (also in Table 3) labelled as in [285] with realizations in F-theory, adapted from [57]. The $k = 1$ column denotes the cases in the absence of S-folds, which have been introduced at the beginning of this section. The last row with $\Delta_7 = 1$ corresponds to the cases without 7-branes, and hence the supersymmetry enhances to $\mathcal{N} \geqslant 3$.

| $G$ | $\Delta_7$ | $k = 1$ | $k = 2$ | $k = 3$ | $k = 4$ |
|---|---|---|---|---|---|
| $E_8$ | 6 | $[II^*, E_8]$ | | | |
| $E_7$ | 4 | $[III^*, E_7]$ | | | |
| $E_6$ | 3 | $[IV^*, E_6]$ | $[II^*, C_5]$ | | |
| $D_4$ | 2 | $[I_0^*, D_4]$ | $[III^*, C_3 C_1]$ | $[II^*, A_3 \rtimes \mathbb{Z}_2]$ | |
| $H_2$ | 3/2 | $[IV, H_2]$ | $[IV^*, C_2 U_1]$ | | $[II^*, A_2 \rtimes \mathbb{Z}_2]$ |
| $H_1$ | 4/3 | $[III, H_1]$ | | $[III^*, A_1 U_1 \rtimes \mathbb{Z}_2]$ | |
| $H_0$ | 6/5 | $[II^*, H_0]$ | | | |
| $\emptyset$ | 1 | | $[I_0^*, C_1 \chi_0]$ | $[IV^*, U_1]$ | $[III^*, U_1 \rtimes \mathbb{Z}_2]$ |

As was already noted in the introduction, the subject of 4d $\mathcal{N} = 2$ SCFTs has been an active and vibrant research field over the last thirty years, and it is impossible to encapsulate all interesting aspects in this short review. Instead, we attempt to provide the reader with an essential guidebook to the fundamental features of these SCFTs and basic tools to construct them in string/M-/F-theory. We refer the readers to other existing reviews for complementary perspectives and the references thereof for further details. Finally, despite the substantial progress in understanding these SCFTs, there are numerous open questions. Below, we discuss a number of them briefly.

**Discrete symmetries and anomalies**
We have only kept track of continuous symmetries and their 't Hooft anomalies so far in the $\mathcal{N} = 2$ SCFTs. In recent years, there have been a lot of development in understanding discrete symmetries and their anomalies in QFTs, which prove to be powerful in constraining RG flows and delineating the IR phase diagram (see for example [259, 294]). It is then clearly important to identify how such symmetries are realized in the $\mathcal{N} = 2$ SCFTs, in particular from the geometric constructions in string/M-/F-theory constructions where the anomalies are determined by the inflow mechanism [295] (generalized to discrete symmetries). Recently, some exciting progress has been made along this line, concerning discrete higher form symmetries from the geometric constructions [278, 296–300].

**Complete data to specify a 4d $\mathcal{N} = 2$ SCFT**
A CFT is fully specified by its operator spectrum together with the OPE that satisfy consistency conditions such as unitarity, conformal invariance and crossing symmetry. However, this is not very efficient when the CFT has additional structures, such as 4d $\mathcal{N} = 2$ superconformal symmetry. In this case, one may expect the theory to be fully specified by much fewer parameters, such as the conformal (and flavor) central charges and protected local operator spectrum, up to discrete classes that are sensitive to the defect operators spectrum. The similar problem was analyzed in gauge theories in [7] and it would be interesting to understand such discrete parameters in $\mathcal{N} = 2$ SCFTs especially those that are non-Lagrangian.

**Rationality of 4d $\mathcal{N} = 2$ SCFTs**
A curious feature of the 4d $\mathcal{N} = 2$ SCFTs is the degree of *rationality* in a number of physical

observables which do not yet have an explanation. For example, in known $\mathcal{N} = 2$ SCFTs the half-BPS Coulomb branch operators all have rational scaling dimensions and relatedly the conformal central charges are rational numbers, in contrast to the cases in $\mathcal{N} = 1$ SCFTs where these quantities are generally irrational.

**Minimal 4d $\mathcal{N} = 2$ SCFT**

All known interacting 4d $\mathcal{N} = 2$ SCFTs have rank $\geq 1$, namely, there exists at least one half-BPS Coulomb branch operator. However, there is a priori no reason that there cannot be an interaction SCFT of rank 0 that does not contain any half-BPS Coulomb branch operator. Such an SCFT with the same amount of SUSY is known to exist in 3d, which has neither Coulomb nor Higgs branch half-BPS operators [301]. It would be interesting to rule out or confirm such a possibility in 4d using superconformal bootstrap, by studying the four-point function of the stress tensor multiplet.

**Classification of Coulomb branch EFTs**

The Coulomb branch EFT has been a fruitful playground to study the 4d $\mathcal{N} = 2$ SCFTs, as it provides a window to many physical observables at the strongly coupled fixed point. There have been recent attempts to classify $\mathcal{N} = 2$ SCFTs by analyzing the *admissible* Coulomb branch EFTs [118, 145, 163, 165, 166, 302]. Indeed, for the rank 1 case, this program successfully provides an exhaustive list of candidate theories that by now all have UV complete constructions. However, there are certain assumptions and subtleties that need to be understood. For example, the Coulomb branch chiral ring is assumed to be freely generated (some examples where this is not the case were found in [162, 303, 304]), and furthermore discrete symmetries and their anomalies are yet to be incorporated in the Coulomb branch EFT.

**Study of the Higgs branch and 3d Mirror Lagrangians**

For many of the strongly coupled 4d SCFTs that arise from type IIB geometric engineering, it is harder to access the Higgs branch, which encodes resolutions rather than complex structure deformations of the three-fold singularity. As mentioned in Section 4.8.2, one way to study the Higgs branch of such non-Lagrangian theories is to consider their $S^1$ compactification and look for Lagrangian descriptions of the resulting 3d $\mathcal{N} = 4$ SCFTs in the mirror frame [199]. This has been done systematically in [234, 237, 238, 240, 305] for the class of AD theories of the $(A,A)$, $(A,D)$ and $(D,D)$ type.[74] It would be interesting to pursue the studies of the 3d mirror theories for the remaining $(G,G')$ theories, in particular for those class of theories that do not admit a Class $\mathcal{S}$ description.

A generalization/weaker version of this construction are the so-called *magnetic quivers*. These are quivers whose Coulomb branch, when understood as a 3d theory, coincides with the Higgs branch of our 4d theory of interest (the same notion applies to 5 or 6 dimensional theories) [306–311].[75] They can be computed from the brane construction of the theory [311–318][76] or from its geometric engineering [60, 236, 264, 284, 325, 326], and can be

---

[74]In [237, 238, 305] the authors study also the conformal manifolds of such theories. They also provide the 3d mirror theories of $D_p(\mathrm{SU})$ and $D_p(\mathrm{SO})$ theories [43, 229].

[75]Note that we are not requiring that the Higgs branch of the magnetic quiver coincides with the Coulomb branch of the original theory, as opposed to 3d mirror symmetry. As a result, the magnetic quiver construction is not so much a physical connection between two theories as a bookkeeping device for the geometry. For example, it can happen that the Higgs branch of the theory of interest is the union of several hyperKähler cones, each of which will have its own associated magnetic quiver.

[76]The work [318] also studies different but IR dual magnetic quivers for the same Higgs branch. This is further supported by investigating how the supersymmetric line defects transform under the duality. Previous works on how 3d duality acts on supersymmetric line defects include [319, 320] in abelian gauge theories and [321–324] in non-abelian gauge theories.

used to obtain information about the Higgs branch of the theory, such as its symmetries, the stratification of the symplectic singularity, etc. [317, 327–333].

**Chiral algebra from geometry**

Any 4d $\mathcal{N} = 2$ SCFT contains an important and rich subsector in its full operator algebra described by a 2d chiral algebra [86], which encodes many features of the SCFT, especially those pertaining to the Higgs branch. This leads to a natural question to understand which 2d chiral algebras can arise in 4d $\mathcal{N} = 2$ SCFTs. Since most of the known SCFTs come from geometric constructions in string/M-/F-theory, it is desirable to identify the chiral algebra sector directly from the geometry.

# Acknowledgments

The authors thank Ibou Bah, Jonathan Heckman, Ken Intriligator, Sara Pasquetti, Shlomo Razamat, Sakura Schafer-Nameki and Alessandro Tomasiello for organizing the 2020 summer school "QFT and Geometry" and for their comments to the draft. We also thank Fabio Apruzzi, Ibou Bah, Federico Carta, Jacques Distler, Ori Ganor, Simone Giacomelli, Prem Kumar, Noppadol Mekareeya, Carlos Núñez, Diego Rodríguez-Gómez, Lucas Schepers and Dan Xie for comments on the draft and interesting discussions. Finally, we thank Rajeev Singh for the collaboration at the beginning of this project. M.A. is partially supported by an STFC consolidated grant ST/S505778/1. G.A.-T. is supported by the Spanish government scholarship MCIU-19-FPU18/02221. A.M. received funding from "la Caixa" Foundation (ID 100010434) with fellowship code LCF/BQ/IN18/11660045 and from the European Union's Horizon 2020 research and innovation program under the Marie Skłodowska-Curie grant agreement No. 713673 until September 2021. The work of A.M. is supported in part by Deutsche Forschungsgemeinschaft under Germany's Excellence Strategy EXC 2121 Quantum Universe 390833306. H.-Y.S. is supported in part by the Simons Collaborations on Ultra-Quantum Matter, a grant No. 651440 (A.K.) from the Simons Foundation. Z.S. acknowledges the support from the US Department of Energy (DOE) under cooperative research agreement DE-SC0009919 and Simons Foundation award #568420. Y.W. is supported in part by the Center for Mathematical Sciences and Applications and the Center for the Fundamental Laws of Nature at Harvard University.

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
