# Peer review of "The Hitchhiker's Guide to 4d $\mathcal{N}=2$ Superconformal Field Theories"

_SciPost Physics Lecture Notes, doi:SciPost Phys. Lect. Notes 64 (2022)_

## Round 2 · Referee Report · Mario Martone (Referee 1) · 2022-3-24

Report

The manuscript is well-written and reviews important developments in a very active area in high energy theoretical physics. Overall the choice of topics and their interconnection is excellent, thus I would recommend its publication in SciPost Physics Lecture Notes.

Requested changes

In order of appearance.

Introduction:

  1. Second paragraph at page 3, please integrate the first phrase also including a references to the original Argyres-Douglas theories work. They clearly represent a departure from Lagrangian theories which pre-dates many of the string theoretic developments.

  2. Please elaborate, in the caption, on how the external circle of figure 1 connects with the rest of the diagram.

Section 2:

  1. Page 24: it might be helpful to stress that the b-function is exact at one-loop both in N=1 and N=2, but in the former case the corrections to the Kahler potential make the physical coupling renormalize (this is mentioned in passing at the beginning of the section).

  2. The discussion around (2.50) misses a crucial development explained in [2006.16255], namely that R(A) and R(B) are related in a very simple form to the degrees of freedom becoming massless on the complex co-dimension 1 singular locus. The sentence "but in practice it still remains a challenge for general SCFT" reads particularly weird given that the central charge formulae - eq (1.1a)-(1.1c) of the aforementioned reference - provide an explicit and general expression for the a and c from which R(A) and R(B) can be readily obtained.

Section 3.

  1. Page 40 "an example can be found at the origin u=0 in the rank-1 cases, where a=\sqrt{u/2}", as it is the sentence is misleading. u=0 is only a singularity for rank-1 SCFT/IR-free theories and a=\sqrt{u/2} is specifically only true for SU(2) SCFT theories. It is up to the authors what to omit/add to make this sentence correct.

  2. Last paragraph in 47 misses the same aforementioned developments. While it is true that [75] initiated this set of ideas, (3.42) should be substituted by the central charge formulae, eq (1.1a)-(1.1c) of [2006.16255] perhaps preceded with a short explanation of the main content of these equations.

Section 4:

  1. Eq. (4.19) misses J_2.

  2. Footnote 42, Page 70 add reference to [2112.10227]

  3. Footnote 44, Page 71 perhaps add reference to [2005.12282] which shows that CB operators with fractional dimensions do appear in the regular twisted setup.

Section 5:

Typos:

Page 4: not uncommon that a same -> not uncommon that the same. Page 9: in the flat space -> in flat space. Page 27: calculated in UV -> calculated in the UV Page 61: being semi-infinite makes ->being semi-infinite make. Page 76: Thus, on should consider non-compact Calabi-Yau manifold compactifications -> Thus, one should consider compactifications on non-compact Calabi-Yau manifold.

  • validity: good
  • significance: good
  • originality: ok
  • clarity: good
  • formatting: good
  • grammar: good

Author:  Alessandro Mininno  on 2022-05-29  [id 2535]

(in reply to Report 1 by Mario Martone on 2022-03-24)

Dear Editor, We thank the referee for the detailed report. Following their bullet points, we have done the following modifications that will be integrated in a new version of our manuscript:

Introduction: 1. We modified the sentence to highlight AD work prior to the string constructions 2. We added a caption in Figure 1 explaining the relation between the outer circle and the internal graphs. 3. Page 24. We added a footnote 16 and NSVZ references. 4. The discussion around (2.50) has been improved Section 3. 1. Page 40 “an example can be found at the origin u=0 in the rank-1 cases, where a=\sqrt{u/2}“, has been explained better 2. Last paragraph in 47 has now the new developments as asked.

Section 4:

  1. Explanation added on the meaning of the subscripts for J
  2. Footnote 42, Page 70 added reference to [2112.10227]
  3. Footnote 44, Page 71 perhaps added reference to [2005.12282] with comment.

All recommended typos have been corrected.

We hope that such modifications are sufficient to approve the manuscript here.

Thank you Best regards The authors

---

## Round 2 · Referee Report · Anonymous (Referee 2) · 2022-7-7

Strengths

1-Clearly written lectures on 4d N=2 SCFTs.
2-Well-balanced mixed of introductory material with more "modern" QFT and stringy point of views.
3-List of references is very thorough.

Weaknesses

-A few imprecisions or confusions, see below.

Report

This is a thorough and clearly written set of lectures on 4d N=2 SCFTs, a very important and active research area. After a basic intro to N=2 SUSY, the lectures focus on three aspects which are all topical: IR approaches to SCFT classification; class-S constructions; geometric engineering. This will be a very useful reference for anyone who want to enter this field of research.

I only have a few basic suggestions for improvement, listed below. Other than that, I strongly recommend these lecture notes for publication in SciPost Physics Lecture Notes.

(I apologize to the authors for the unseemly delay in sending this report!)

Requested changes

1-On top of page 35, it says that "there always exists a duality frame in which is EFT is manifestly weakly coupled". That is false, since the the effective gauge coupling at a generic point will be finite in any duality frame, not weakly coupled. Please correct.

2-I think the discussion on p38 is imprecise for the same reason.

3-In (3.25), it might be worth pointing out that this is an asymptotic expansion of an underlying analytic function F. (Given F, for instance from the SW solution, you can get (3.25), but not the other way around.)

4-Second paragraph of p40: it says "the effective gauge coupling diverges at these singularities". That is only true at cusps, not at elliptic points (at rank one, and similarly at higher rank). Precisely at SCFT points (AD or MN points), the coupling is fixed and finite, not weak, but (a,aD) are not single-valued.

5-In the discussion before (4.1), it is not clear why we get 5 real scalars for every resolution parameter of the ADE singularity.

6-p54,55: you mention that the two compactifications of Fig.7 are related by 3d mirror symmetry. Is there an explanation why this is the case?

List of typos:
-p40: "due to that certain..."--> "due to the fact that certain..."
-p40: "lead to divergences"-->"leads to divergences"
-p44: "As alluded," --> "As alluded to,"
-p67, footnote 40: one reference missing at the end "[?,...]"
-p68: "Alday-Gaiotto-Tachikawa (AGT) correspondence..."--> "The Alday-Gaiotto-Tachikawa (AGT) correspondence...". Also in the title of the subsection: "AGT"-> "the AGT".

  • validity: high
  • significance: top
  • originality: good
  • clarity: high
  • formatting: excellent
  • grammar: good

Author:  Alessandro Mininno  on 2022-08-23  [id 2746]

(in reply to Report 2 on 2022-07-07)
Category:
answer to question
correction

Dear Referee and Editor,
We have uploaded a new version of the manuscript on ArXiv with all the modifications that have been asked. I hope that such changes are sufficient to approve the lecture notes

---

## Editorial Decision

published